# Deciphering cross-cohort metabolic signatures of immune responses and their implications for disease pathogenesis

Jianbo Fu [1,2], Nienke van Unen[1,2], Andrei Sarlea[3], Nhan Nguyen [1,2], Martin Jaeger[3], Javier Botey Bataller[1,2,3], Valerie A C M Koeken[3], L Charlotte de Bree[3], Vera P Mourits[3], Simone J C F M Moorlag[3], Godfrey Temba[3,4], Vesla I Kullaya[4,5], Quirijn de Mast[3], Leo A B Joosten[3,6], Cheng-Jian Xu [1,2], Mihai G Netea[3,7,9] & Yang Li [1,2,3,8,9 ✉]

## Abstract

The complex interplay between circulating metabolites and immune responses, which is pivotal to disease pathophysiology, remains poorly understood and understudied in systematic research. Here, we performed a comprehensive analysis of the immune response and circulating metabolome in two Western European cohorts (534 and 324 healthy individuals) and one from sub-Saharan Africa (323 healthy donors). At the metabolic level, our analysis revealed sex-specific differences in the correlation between phosphatidylcholine and cytokine responses following ex vivo stimulation. Notably, sphingomyelin exhibited a significant negative correlation with monocyte-derived cytokine production in response to *Staphylococcus aureus* stimulation, a finding that was validated through functional experiments. Subsequently, using Mendelian randomization analysis, we established a link between sphingomyelin and COVID-19 severity, providing compelling evidence for its modulatory role in immune responses during human infection. Collectively, our results represent a unique resource (https://lab-li.ciim-hannover.de/apps/imetabomap/) for exploring metabolic signatures associated with immune function in different populations, highlighting sphingomyelin metabolism as a potential target in treating inflammatory and infectious diseases.

Keywords Immune Response; Metabolomics; Multi-cohort; Multi-omics; Mendelian Randomization
Subject Categories Immunology; Metabolism; Microbiology, Virology & Host Pathogen Interaction

## Introduction

Human metabolism and immune response are closely linked, playing an important role in maintaining human health (Zmora et al, 2017). While the immune system protects the body against pathogens and maintains tissue homeostasis, these functions come at a significant bioenergetic cost, requiring precise control of cellular metabolic pathways (Ganeshan and Chawla, 2014). In turn, various metabolites serve not only as energy sources or building blocks of cellular function, but also as modulators of immune responses (O'Neill et al, 2016). An abnormal interaction between cellular metabolism and immune responses contributes to auto-immune, metabolic, and infectious diseases (Blanco and Kaplan, 2023; Palmer, 2022). Pro-inflammatory cytokines are involved in promoting inflammation and modulating adaptive immune responses, which are fundamental components of the immune response. In addition, many studies have shown that pro-inflammatory cytokines, such as tumor necrosis factor (TNF) (Sethi and Hotamisligil, 2021), interleukin 6 (IL-6) (Han et al, 2020; Wedell-Neergaard et al, 2019), IL-1β (de Baat et al, 2023), and interferon-gamma (IFN-γ) (Bradley et al, 2022), can influence insulin resistance, adipose tissue inflammation and regulate obesity-related metabolism. On the other hand, metabolite reprogramming can modulate inflammatory states. As just a few examples, uric acid promotes IL-1β production in peripheral blood mononuclear cells (Crisan et al, 2016), palmitate induces the secretion of IL-1β and IL-18 by macrophages (Wen et al, 2011), while TCA cycle metabolites such as fumarate, mevalonate, and itaconate modulate trained immunity responses (Arts et al, 2016).

Since immunity and metabolism play crucial roles and interact in health and disease, research in the field of immunometabolism has been steadily increasing in recent years (Bartel et al, 2015; Chu et al, 2021; Nath et al, 2017). However, a systematic assessment of the

[1]Centre for Individualised Infection Medicine (CiiM), a joint venture between the Helmholtz Centre for Infection Research (HZI) and Hannover Medical School (MHH), Hannover, Germany. [2]TWINCORE Centre for Experimental and Clinical Infection Research, a joint venture between the Helmholtz Centre for Infection Research (HZI) and the Hannover Medical School (MHH), Hannover, Germany. [3]Department of Internal Medicine and Radboud Center for Infectious Diseases, Radboud University Medical Center, Nijmegen, The Netherlands. [4]Department of Medical Biochemistry and Molecular Biology, Kilimanjaro Christian Medical University College (KCMUCo), Moshi, Tanzania. [5]Kilimanjaro Clinical Research Institute (KCRI), Kilimanjaro Christian Medical Center, Moshi, Tanzania. [6]Department of Medical Genetics, Iuliu Hatieganu University of Medicine and Pharmacy, Cluj-Napoca, Romania. [7]Department of Immunology and Metabolism, Life and Medical Sciences Institute (LIMES), University of Bonn, Bonn, Germany. [8]Cluster of Excellence RESIST (EXC 2155), Hannover Medical School, Carl-Neuberg-Straße 1, Hannover 30625, Germany. [9]These authors contributed equally: Mihai G Netea, Yang Li. ✉E-mail: yang.li@helmholtz-hzi.de

interplay between circulating metabolites and cytokine responses is missing, due to the difficulty to measure both immune response and metabolomic data from the same biological sources, such as identical samples and cell systems, across multiple cohorts. Furthermore, metabolic and immune interactions can vary considerably across diverse ethnic and geographical backgrounds due to genetic, environmental, dietary, and lifestyle factors (Christ et al, 2018; Temba et al, 2021; Ter Horst et al, 2016; Thorburn et al, 2014).

In this study, we analyzed plasma metabolite concentrations and cytokine responses in three distinct cohorts, totaling 1181 individuals, including two from Western Europe and one from Sub-Saharan Africa. In total, we investigated the relationships between 4361 metabolite features and immune cytokine responses, encompassing 172 distinct cytokine production responses to various stimuli. Our study aimed to understand the interaction between cytokine responses and metabolic interactions across different populations, in both men and women. To achieve this, we first analyzed the correlations between metabolite features and immune cytokine responses in each cohort. Subsequently, we conducted enrichment analyses to identify metabolites significantly associated with specific cytokine responses. We experimentally validated the relationships between specific metabolites and cytokine responses in in vitro models of cytokine production stimulation assays. Subsequently, we employed Mendelian randomization (MR) analysis to explore the causal relationships between the metabolites and infectious diseases such as COVID-19, to demonstrate the importance of these interactions in actual human infections. Finally, we integrated all metabolite–cytokine association results into a publicly available database, offering insights into immunity and metabolism interactions, and supporting new treatment development.

# Results

## Circulating metabolome and innate/adaptive immune response profiling across cohorts

To comprehensively understand the relationship between metabolites and immune cytokine responses, we integrated data from two different European healthy populations: Cohort_EU1 and Cohort_EU2 from Western Europe (Netherlands) with 534 and 324 participants, respectively, and one African cohort: Cohort_AF from sub-Saharan Africa (Tanzania) with 323 participants. We quantified plasma metabolite features across these cohorts using flow-injection time-of-flight mass spectrometry (TOF-M), identifying 1377 metabolite features in Cohort_EU1, 1373 in Cohort_EU2, and 1611 in Cohort_AF. In parallel, we assessed innate and adaptive immune response by measuring cytokine production in response to various stimuli: Cohort_EU1 was assessed for 6 cytokines across 18 stimulations, Cohort_EU2 for 4 cytokines across 2 stimulations, and Cohort_AF for 5 cytokines across 10 stimulations (Fig. 1A). Figure 1B provides an overview of the study design, selecting Cohort_EU1 and Cohort_AF as discovery cohorts and Cohort_EU2 as the replication cohort.

## Robust plasma metabolic pathways for immune functions across European and African populations

We initiated our analysis by examining metabolic networks through weighted co-expression network analysis of metabolite features using the WGCNA framework, to reveal interactions between metabolism

and immune responses. Specifically, we assessed metabolite co-expression networks associated with immune phenotypes (IL-1β, IL-6, TNF, and IFN-γ) following S. aureus stimulation, as these measurements were shared across the three cohorts. Of note, the peripheral blood mononuclear cells (PBMCs) in Cohort_EU2, the whole blood (WB) in Cohort_AF, and both WB and PBMCs in Cohort_EU1 were examined. We identified 11, 11, 10, and 10 modules of highly correlated metabolites in Cohort_EU1 (PBMCs), Cohort_EU1 (WB), Cohort_AF, and Cohort_EU2, respectively (as shown in Appendix Fig. S1 and Datasets EV1–3). Subsequently, we correlated the metabolites modules with the both monocyte-derived cytokines (IL-1β, IL-6, and TNF) and T-cell-derived cytokine (IFN-γ) profiles following S. aureus stimulation. We identified 7, 5, 8, and 3 metabolite modules that were found to be significantly associated with cytokine responses induced by S. aureus in Cohort_EU1 (PBMCs), Cohort_EU1 (WB), Cohort_AF, and Cohort_EU2, respectively, as shown in Fig. 2A–D.

In total, four metabolite modules were associated with monocyte-derived cytokines (IL-1β, IL-6, TNF) in the PBMCs of Cohort_EU1 following stimulation with S. aureus. The brown and blue modules exhibited negative correlations with cytokine production, while the pink and red modules showed positive correlations (Fig. 2A). The metabolites of the brown module were enriched in the glycerophospholipid metabolism ($P = 0.00028$) and sphingolipid metabolism pathways ($P = 0.00088$) (Fig. 2E; Table EV1); whereas the metabolites of the blue module were enriched in the primary bile acid biosynthesis and steroid biosynthesis pathways (Appendix Fig. S2A; Table EV1). On the other hand, the metabolites of the pink and red modules were enriched in the pathways of nicotinate and nicotinamide metabolism, alanine, aspartate and glutamate metabolism, and aminoacyl-tRNA biosynthesis, as well as vitamin B6 metabolism, tyrosine metabolism pathways (Appendix Fig. S2B,C; Table EV1).

We also identified a gray module correlated with cytokine production (TNF and IFN-γ) in the WB of Cohort_EU1 stimulated by S. aureus (Fig. 2B). Metabolites of this module were enriched in histidine metabolism ($P = 0.00846$), purine metabolism ($P = 0.01840$), and glycerophospholipid metabolism ($P = 0.04006$, Fig. 2F; Table EV1). In Cohort_AF, we observed two metabolite modules (magenta and red) negatively correlated with both monocyte-derived cytokines (IL-1β, IL-6, TNF) and T-cell-derived cytokine (IFN-γ) production after S. aureus stimulation (Fig. 2C). Metabolites of the magenta module were enriched in glycerophospholipid metabolism ($P = 0.00502$) and purine metabolism ($P = 0.01596$, Fig. 2G; Table EV2), while metabolites of the red module were enriched in pyrimidine metabolism, nicotinate and nicotinamide metabolism, and lysine degradation (Appendix Fig. S2D; Table EV2). It is worth noting that the glycerophospholipid metabolism pathway was consistently identified in both Cohort_EU1 (PBMCs and WB) and Cohort_AF, underscoring its robustness (as shown in Fig. 2E–G). This result was further validated in the replication cohort (Corhort_EU2). The magenta module, negatively correlated with IL-6, showed enrichment in the glycerophospholipid metabolism pathway (Fig. 2H; Table EV3). A meta-analysis across all four cohorts (Cohort_EU1 WB, Cohort_EU1 PBMCs, Cohort_AF and Cohort_EU2) confirmed a highly consistent and significant enrichment of the glycerophospholipid metabolism pathway (pooled effect size of $r = 2.71$, 95% CI [1.80, 3.62], $P < 0.0001$), with low heterogeneity observed ($I^2 = 0\%$, 95% CI: 0–84.8%, Q = 1.22, $P = 0.748$) (Appendix Fig. S3). This finding

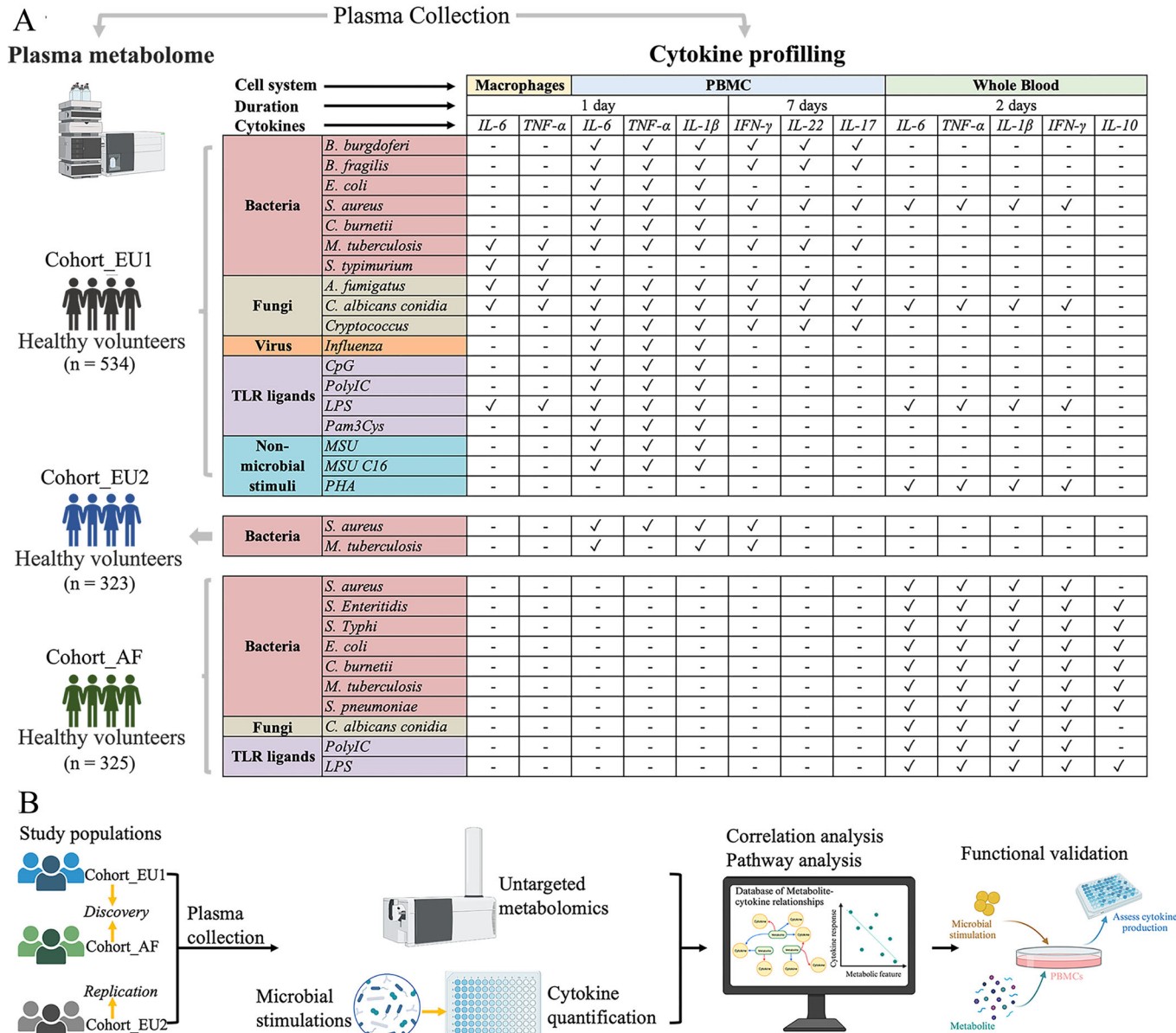

## A — Plasma Collection

**Plasma metabolome** — Cohort_EU1, Healthy volunteers (n = 534); Cohort_EU2, Healthy volunteers (n = 323); Cohort_AF, Healthy volunteers (n = 325)

**Cytokine profiling**

| Cell system | | Macrophages | | PBMC | | | | | | Whole Blood | | | | |
|---|---|---|---|---|---|---|---|---|---|---|---|---|---|---|
| Duration | | | | 1 day | | | | 7 days | | 2 days | | | | |
| Cytokines | | IL-6 | TNF-α | IL-6 | TNF-α | IL-1β | IFN-γ | IL-22 | IL-17 | IL-6 | TNF-α | IL-1β | IFN-γ | IL-10 |
| Bacteria | *B. burgdoferi* | - | - | ✓ | ✓ | ✓ | ✓ | ✓ | ✓ | - | - | - | - | - |
| | *B. fragilis* | - | - | ✓ | ✓ | ✓ | ✓ | ✓ | ✓ | - | - | - | - | - |
| | *E. coli* | - | - | ✓ | ✓ | ✓ | - | - | - | - | - | - | - | - |
| | *S. aureus* | - | - | ✓ | ✓ | ✓ | ✓ | ✓ | ✓ | ✓ | ✓ | ✓ | ✓ | - |
| | *C. burnetii* | - | - | ✓ | ✓ | ✓ | - | - | - | - | - | - | - | - |
| | *M. tuberculosis* | ✓ | ✓ | ✓ | ✓ | ✓ | ✓ | ✓ | ✓ | - | - | - | - | - |
| | *S. typimurium* | ✓ | ✓ | - | - | - | - | - | - | - | - | - | - | - |
| Fungi | *A. fumigatus* | ✓ | ✓ | ✓ | ✓ | ✓ | ✓ | ✓ | ✓ | - | - | - | - | - |
| | *C. albicans conidia* | ✓ | ✓ | ✓ | ✓ | ✓ | ✓ | ✓ | ✓ | ✓ | ✓ | ✓ | ✓ | - |
| | *Cryptococcus* | - | - | ✓ | ✓ | ✓ | ✓ | ✓ | ✓ | - | - | - | - | - |
| Virus | *Influenza* | - | - | ✓ | ✓ | ✓ | - | - | - | - | - | - | - | - |
| TLR ligands | *CpG* | - | - | ✓ | ✓ | ✓ | - | - | - | - | - | - | - | - |
| | *PolyIC* | - | - | ✓ | ✓ | ✓ | - | - | - | - | - | - | - | - |
| | *LPS* | ✓ | ✓ | ✓ | ✓ | ✓ | - | - | - | ✓ | ✓ | ✓ | ✓ | - |
| | *Pam3Cys* | - | - | ✓ | ✓ | ✓ | - | - | - | - | - | - | - | - |
| Non-microbial stimuli | *MSU* | - | - | ✓ | ✓ | ✓ | - | - | - | - | - | - | - | - |
| | *MSU C16* | - | - | ✓ | ✓ | ✓ | - | - | - | - | - | - | - | - |
| | *PHA* | - | - | - | - | - | - | - | - | ✓ | ✓ | ✓ | ✓ | - |
| Bacteria | *S. aureus* | - | - | ✓ | ✓ | ✓ | ✓ | - | - | - | - | - | - | - |
| | *M. tuberculosis* | - | - | ✓ | - | ✓ | ✓ | - | - | - | - | - | - | - |
| Bacteria | *S. aureus* | - | - | - | - | - | - | - | - | ✓ | ✓ | ✓ | ✓ | - |
| | *S. Enteritidis* | - | - | - | - | - | - | - | - | ✓ | ✓ | ✓ | ✓ | ✓ |
| | *S. Typhi* | - | - | - | - | - | - | - | - | ✓ | ✓ | ✓ | ✓ | ✓ |
| | *E. coli* | - | - | - | - | - | - | - | - | ✓ | ✓ | ✓ | ✓ | ✓ |
| | *C. burnetii* | - | - | - | - | - | - | - | - | ✓ | ✓ | ✓ | ✓ | ✓ |
| | *M. tuberculosis* | - | - | - | - | - | - | - | - | ✓ | ✓ | ✓ | ✓ | ✓ |
| | *S. pneumoniae* | - | - | - | - | - | - | - | - | ✓ | ✓ | ✓ | ✓ | ✓ |
| Fungi | *C. albicans conidia* | - | - | - | - | - | - | - | - | ✓ | ✓ | ✓ | ✓ | - |
| TLR ligands | *PolyIC* | - | - | - | - | - | - | - | - | ✓ | ✓ | ✓ | ✓ | - |
| | *LPS* | - | - | - | - | - | - | - | - | ✓ | ✓ | ✓ | ✓ | ✓ |

## B

Study populations: Cohort_EU1 / *Discovery* ; Cohort_AF / *Replication* ; Cohort_EU2

Plasma collection → Microbial stimulations → Cytokine quantification → Untargeted metabolomics → Correlation analysis / Pathway analysis (Database of Metabolite-cytokine relationships) → Functional validation (Microbial stimulation, Metabolite, PBMCs, Assess cytokine production)

**Figure 1. Study overview and analytical workflow.**

(A) Cohorts, sample sizes, and cytokine profiling summary. Three independent human cohorts were studied: Cohort_EU1 (*n* = 534, Western Europe, both PBMC and whole-blood assays), Cohort_EU2 (*n* = 323, Western Europe, PBMC only) and Cohort_AF (*n* = 325, Sub-Saharan Africa, whole-blood only). For each cohort, plasma was collected for untargeted metabolomics and a panel of ex vivo stimulations. In the matrix, rows list each bacterial, fungal, viral, TLR-ligand, or non-microbial stimulus, and columns are grouped by cell system (macrophages, PBMC or whole blood), stimulation duration (1 day, 2 days or 7 days) and cytokine measured (IL-6, TNF, IL-1β, IFN-γ, IL-22, IL-17, or IL-10). A ✓ indicates that the named cytokine was quantified by ELISA under that exact combination of stimulus, cell system and timepoint. For example, in Cohort_EU1 whole blood stimulated for 2 days with *S. aureus*, IL-6, TNF, IL-1β, and IFN-γ were measured (✓), whereas IL-10 was not (–). Brackets on the left annotate which rows apply to each cohort; only stimuli within a given bracket were assayed in that cohort. This layout allows direct comparison of which stimulus–cell–timepoint–cytokine combinations were profiled across cohorts. (B) Discovery and validation pipeline. Cohort_EU1 (WB + PBMCs) and Cohort_AF (WB) served as discovery sets, with Cohort_EU2 (PBMC) as an independent replication cohort. Spearman's rank correlations were computed between each metabolite and cytokine response (FDR-corrected), followed by WGCNA-based module detection and pathway enrichment. Functionally relevant hits (e.g., sphingomyelin) were validated by adding exogenous lipid to PBMC cultures and measuring dose-dependent inhibition of TNF, IL-1β, and IL-6. Finally, causal inference was assessed by two-sample Mendelian randomization, and all associations were integrated into the iMetaboMap web resource for community access.

aligns with the known role of glycerophospholipid metabolism is known to play an important role in regulating immune responses (Fuchs et al, 2019; Yan et al, 2022).

In summary, our study identified a common link between metabolic pathways and immune responses across different cohorts. Notably, glycerophospholipid metabolism consistently

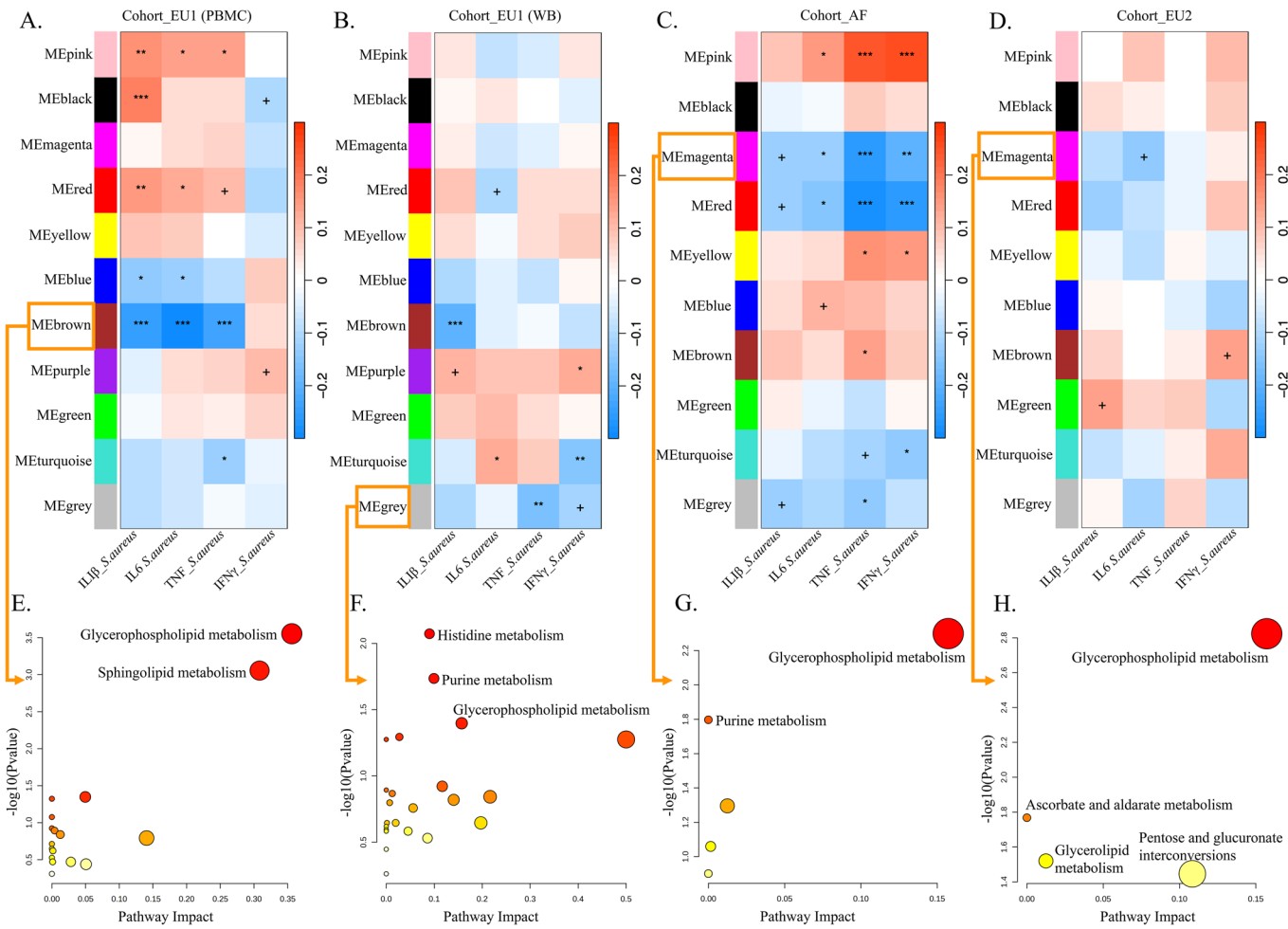

**Figure 2. Metabolite modules associated with cytokine responses induced by *S. aureus* in African and European populations.**

The specific metabolites within each WGCNA-defined module are listed in Table EV1 (**A–D**). In the three cohorts (Cohort_EU1, Cohort_AF, and Cohort_EU2), 11, 11, 10, and 10 modules, respectively, were found to correlate with cytokine responses (IL-1β, IL-6, TNF, and IFN-γ) induced by *S. aureus*. Significance thresholds: ***FDR < 0.001; **FDR < 0.01; *FDR < 0.05; +FDR < 0.1. (**E–H**) Pathway enrichment analysis for the MEbrown (**A**), MEgrey (**B**), MEmagenta (**C**), and MEmagenta (**D**) modules. Larger and darker bubbles represent higher −log(*P* value), indicating higher significance. Pathway enrichment was performed using a hypergeometric test. Cytokine responses were measured using PBMCs in Cohort_EU1 (**A**), whole blood in Cohort_EU1 (**B**), whole blood in Cohort_AF (**C**), and PBMCs in Cohort_EU2 (**D**). Source data are available online for this figure.

emerged as a pathway linked to immune regulation, highlighting its importance in infection-induced immune responses.

## Metabolic markers with potential sex-specific differences for immune functions

Since sex impacts both cytokine production (Aulock et al, 2006) and metabolic regulation (Tramunt et al, 2020), we investigated its impact on the relationship between metabolome and *S. aureus*-induced cytokine responses (IL-1β, IL-6, TNF, and IFN-γ). We had a balanced sex distribution among participants, with 50.77%, 56.33%, and 56.70% females, in Cohort_AF, Cohort_EU1, and Cohort_EU2, respectively (Appendix Fig. S4). In Cohort_EU1 (PBMCs), we identified 95 metabolites in males and 302 metabolites in females that were significantly correlated with at least one cytokine response, after adjusting for age and body mass index (BMI). In Cohort_EU1 (WB), 7 metabolites in males and 52

metabolites in females showed significant correlations with at least one cytokine response. However, no metabolites in Cohort_EU2 exhibited such correlations, possibly due to the sample size and large variability within the metabolite and cytokine data, limiting the statistical power to detect such correlations. In Cohort_AF, we identified 59 metabolites in males and 10 in females with significant correlations (Spearman correlation, FDR < 0.05 and Datasets EV4–6).

Interestingly, metabolites associated with cytokine response in males were prominently linked to glycerophospholipid metabolism and sphingolipid metabolism (Fig. 3A,B). In contrast, metabolites associated with cytokine response in females (Fig. 3C,D) were involved in a diverse range of metabolic pathways, including cysteine and methionine metabolism, linoleic acid metabolism, and aminoacyl-tRNA biosynthesis, with arachidonic acid metabolism having particularly notable effects. Pathway analysis is constrained by the limited number of significant associations. Despite these

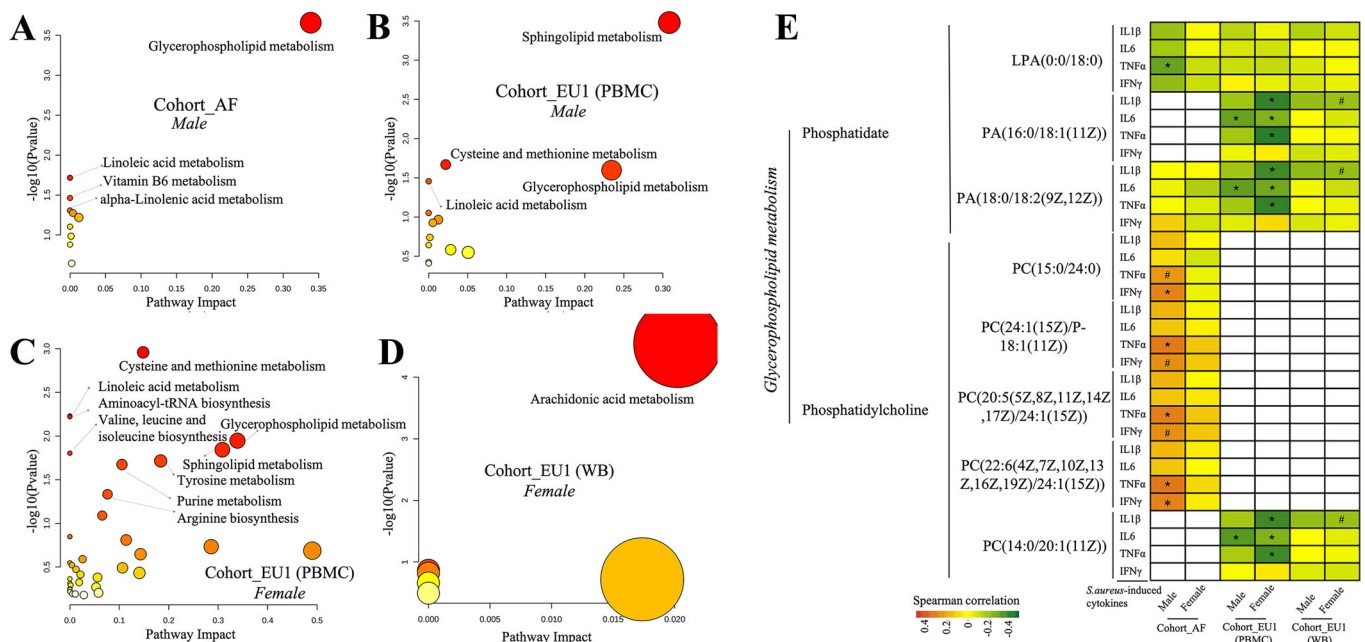

**Figure 3. Sex-specific metabolite–cytokine associations and pathway differences.**

Scatter plots illustrate pathway analysis results for significant correlations (FDR < 0.05) between metabolites and cytokine responses, with distinctions based on gender in varied cohorts: (A) Cohort_AF, male; (B) Cohort_EU1 (PBMCs), male; (C) Cohort_EU1 (PBMCs), female; (D) Cohort_EU1 (WB), female. The darker the bubble color, the larger the −log(Pvalue), indicating higher significance. Pathway enrichment was performed using a hypergeometric test. (E) Differences in the Glycerphospholipid metabolism between males and females. Red signifies a significant positive correlation between the metabolite and cytokine response, while green indicates a significant negative correlation. Cells shaded in light red and light green represent 0.05 < FDR < 0.10, with light red suggesting a trend toward positive correlation and light green pointing to a trend toward negative correlation. Asterisk *(FDR < 0.05) and hash # (0.05 < FDR < 0.10). Source data are available online for this figure.

limitations, sex differences in the major metabolic pathways were observed, with some shared pathways, such as sphingolipid metabolism, associated with cytokine response in both male and female subjects.

In addition, the metabolites from the glycerophospholipid metabolism pathway were associated with cytokine response in both males and females across different cohorts (Fig. 3A–C; Appendix Fig. S5). Within the glycerophospholipid metabolism pathway, phosphatidate metabolites were generally negatively correlated with monocyte-derived cytokine responses, while phosphatidylcholine metabolites displayed variable correlations with cytokine responses across different populations (Fig. 3E).

Among phosphatidate metabolites, lysophosphatidic acid (LPA) (0:0/18:0) showed a significant negative correlation with TNF response in males of the Cohort_AF, whereas both phosphatidic acid (PA)(16:0/18:1(11Z)) and PA(18:0/18:2(9Z,12Z)) displayed significant negative correlations with *IL-6* response in both males and females of Cohort_EU1 (PBMCs). In addition, PA(16:0/18:1(11Z)) and PA(18:0/18:2(9Z,12Z)) also exhibited significant negative correlations with IL-1β and TNF response in females of the Cohort_EU1(PBMCs) and Cohort_EU1(WB). Subsequently, statistical testing of correlation coefficients between male and female volunteers revealed that in Cohort_EU1 (PBMCs), sex-specific differences for PA(16:0/18:1(11Z)) versus TNF/IL-1β and for PA(18:0/18:2(9Z,12Z)) versus TNF/IL-1β showed suggestive evidence ($P < 0.05$; see Table EV4).

In the phosphatidylcholine group of the glycerophospholipid metabolism pathway, a positive correlation trend with TNF and

IFN-γ response was observed solely in males of the Cohort_AF. In contrast, this phosphatidylcholine group showed negative correlations with cytokine responses for both males and females in the European cohorts. For instance, phosphatidylcholine (PC)(14:0/20:1(11Z)) displayed a significant negative correlation with IL-6 in both males and females in Cohort_EU1 (PBMCs). In addition, this metabolite showed negative correlations with IL-1β and TNF response in females in Cohort_EU1 (PBMCs) and a negative correlation trend with IL-1β in females in Cohort_EU1 (WB). Subsequently, statistical testing of correlation coefficients between male and female volunteers revealed that in Cohort_EU1 (PBMCs), the sex-specific difference in correlation coefficients between PC(14:0/20:1(11Z)) and TNF/IL-1β showed suggestive evidence ($P < 0.05$; see Table EV4). Similarly, in Cohort_AF, sex-specific differences in correlations for PC(15:0/24:0) versus TNF/IFN-γ and for PC(22:6(4Z,7Z,10Z,13Z,16Z,19Z)/24:1(15Z)) versus TNF/IFN-γ showed suggestive evidence ($P < 0.05$; see Table EV4).

To explore sex-specific metabolite–cytokine relationships, we summarized phosphatidylcholine (PC) and phosphatidate (PA) species into Composite PC and PA scores, adjusted them for age and BMI within each sex, and correlated them with *S. aureus*-induced IL-6, IL-1β, TNF, and IFN-γ (Spearman's ρ, FDR < 0.05). We then compared male versus female ρ values via Fisher's Z and pooled these sex-difference Z's across cohorts by meta-analysis (Appendix Fig. S6). This revealed a pronounced male bias for Composite PC–TNF correlations (fixed-effect r_diff ≈0.743, 95% CI [0.691, 0.787], $P ≈ 2 × 10^{-68}$; random-effects r_diff ≈ 0.745, 95% CI [0.537, 0.867], $P ≈ 1.7 × 10^{-7}$; $I^2 = 91.1\%$), indicating consistently

stronger PC–TNF associations in males (Appendix Fig. S6A). In contrast, although fixed-effect models for Composite PC–IL-1β, –IFN-γ, and –IL-6 suggested positive male biases (all $P < 0.001$), their random-effects 95% CIs all spanned zero, reflecting high heterogeneity and no consistent sex effect (Appendix Fig. S6B–D). Composite PA–cytokine analyses showed no reliable male–female differences for any cytokine (all pooled r_diff CIs included zero), indicating similar PA–cytokine associations in both sexes (Appendix Fig. S6E–H).

In summary, metabolites such as phosphatidylcholine (PC) can serve as potential sex-specific metabolic markers for innate immune responses, consistent with previous studies that highlight sex disparities in these metabolites (Rauschert et al, 2017). This underscores the importance of considering sex disparities in the contribution of the glycerophospholipid metabolites to immune function, essential for both research and therapeutic interventions.

In addition to identifying the glycerophospholipid metabolism pathway, we also discovered the regulatory role of the sphingomyelin metabolism pathway on immune responses in both males and females of cohort_EU1 (PBMC) (Fig. 3B,C), a pathway known for its key role in immune regulation (Maceyka and Spiegel, 2014). Moreover, in females, metabolites associated with cytokine response were significantly enriched in arachidonic acid metabolism (Fig. 3D). Previous studies have highlighted the relevance of arachidonic acid metabolism to the presence of sex differences (Perez-Torres et al, 2010; Zhuang et al, 2017).

Altogether, these findings underscore the intricate interplay between metabolite and cytokine response across sex and various ethnicities.

## Sphingolipid metabolism consistently correlates with the capacity of monocyte-derived cytokine production

Next, we investigated the relationship between individual metabolic features and immune responses to *S. aureus* by calculating Spearman's rank correlation coefficients for each metabolite–cytokine pair. In total, we identified 222, 438, and 152 metabolites in Cohort_AF, Cohort_EU1 (PBMCs), and Cohort_EU1 (WB), respectively, significantly correlated with at least one cytokine response, as shown in Datasets EV7–9 (FDR < 0.05). In the EU2 cohort, no metabolites showed significant associations after FDR correction for multiple comparisons. Based on pathway analysis, vitamin B6 metabolism and sphingolipid metabolism stood out as the most statistically significant pathways within the Cohort_AF (Fig. 4A). While there was a more diverse set of pathways in Cohort_EU1 using PBMC samples (Fig. 4C) compared to WB samples (Fig. 4B), sphingolipid metabolism was also among the top significant pathways in both two cohorts. Thus, sphingolipid metabolism emerges as a common signature across all cohorts, suggesting its universal importance in the context of metabolite–cytokine correlations. This finding aligns with the known role of sphingolipid metabolism and its derived metabolites in immune responses (Hannun and Obeid, 2018; Lee et al, 2023; Maceyka and Spiegel, 2014).

Specifically, in Cohort_EU1 (PBMCs) SM displayed a consistent negative correlation with monocyte-derived cytokines like IL-6, IL-1β, TNF, but not with IFN-γ response (Appendix Figs. S7 and 8). Similarly, in Cohort_EU1 (WB), we found a significant negative correlation between the monocyte cytokine IL-1β and SM, but not

for IFN-γ response (Appendix Figs. S7 and 8). In Cohort_AF, such as TNF exhibited a negative correlation with SM, whereas IL-6 and IFN-γ showed minimal correlation (Appendix Fig. S7). In Cohort_EU2, IL-6 was negatively correlated with SM (Appendix Fig. S7).

Given the above indications that SM may be more closely related to monocyte-derived cytokines rather than IFN-γ, we then performed the meta-analysis of the correlation coefficients between SM and *S. aureus*-induced cytokine production in three cohorts. Here, we have used both the fixed-effect model and the random-effect model for computation. The results (Fig. 4D) showed that circulating SM is significantly negatively correlated with monocyte-derived cytokines (IL-6, TNF, and IL-1β) (Random-effect model: pooled $r = -0.1659$, 95% CI $-0.2118$ to $-0.1200$, $P < 0.0001$; Fixed-effect model: pooled $r = -0.1708$, 95% CI $-0.1881$ to $-0.1535$, $P < 0.0001$). In addition, there is no correlation between circulating SM and T-cell-derived cytokine (IFN-γ) response (Random-effect model: pooled $r = -0.0158$ 95% CI $-0.0471$ to 0.0155, $P = 0.3232$; Fixed-effect model: pooled $r = -0.0156$, 95% CI $-0.0456$ to 0.0144, $P = 0.3074$) (Fig. 4E).

To complement these single-molecule findings, we constructed a Composite SM score by averaging the raw abundance values of all SM species that were individually associated with IL-6, IL-1β, TNF, and IFN-γ responses in the three cohorts. As shown in Appendix Fig. S9, we calculated Spearman correlation coefficients (Fisher's Z) between the Composite SM and *S. aureus*-induced IL-6, IL-1β, TNF, and IFN-γ responses in each cohort (Cohort_AF, Cohort_EU1, and Cohort_EU2), then performed a meta-analysis on these estimates. The association between Composite SM and *S. aureus*-induced IL-6 pooled to $r = -0.136$ (95% CI $[-0.185, -0.086]$, $P = 9.96e-8$) under the fixed-effect model and $r = -0.129$ (95% CI $[-0.242, -0.013]$, $P = 0.0293$) under the random-effects model, indicating that SM consistently suppresses IL-6 secretion in all cohorts with a highly significant negative effect (Appendix Fig. S9A). Furthermore, we conducted a combined meta-analysis of the Spearman correlations between Composite SM and monocyte-derived cytokines (IL-1β, IL-6, TNF) across the three cohorts (Cohort_AF, Cohort_EU1, and Cohort_EU2). As shown in Appendix Fig. S9E, the fixed-effect model pooled $r = -0.128$ (95% CI $[-0.156, -0.099]$, $P = 4.7e-18$), and the random-effects model pooled $r = -0.117$ (95% CI $[-0.189, -0.043]$, $P = 0.00191$). These results demonstrate that higher Composite SM levels are consistently associated with lower monocyte-derived cytokine production, regardless of whether samples were obtained from African or European cohorts.

In summary, SM showed consistent negative correlations with monocyte-derived cytokine responses in the multiple cohorts, suggesting a potential inhibitory effect of SM on the innate immune function. To validate this funding, we stimulated peripheral blood mononuclear cells (PBMCs) with either *S. aureus* or lipopolysaccharide (LPS) for 24 h, followed by measurement of pro-inflammatory cytokines in the supernatant of the stimulated cells (Fig. 5). Pre-treatment of human PBMCs with porcine brain-derived SM significantly inhibited IL-1β and TNF release in response to *S. aureus* stimulation, but had minimal impact on IL-6 (Fig. 5A–F). Chicken yolk-derived SM (10 μM–10 ng/mL) prior to 24 h stimulation with heat-killed *S. aureus* or LPS led to a significant reduction in IL-1β and IL-6 secretion at the highest dose, while TNF exhibited a non-significant downward trend (Fig. 5G–L). No effects of the various conditions on cell viability were observed.

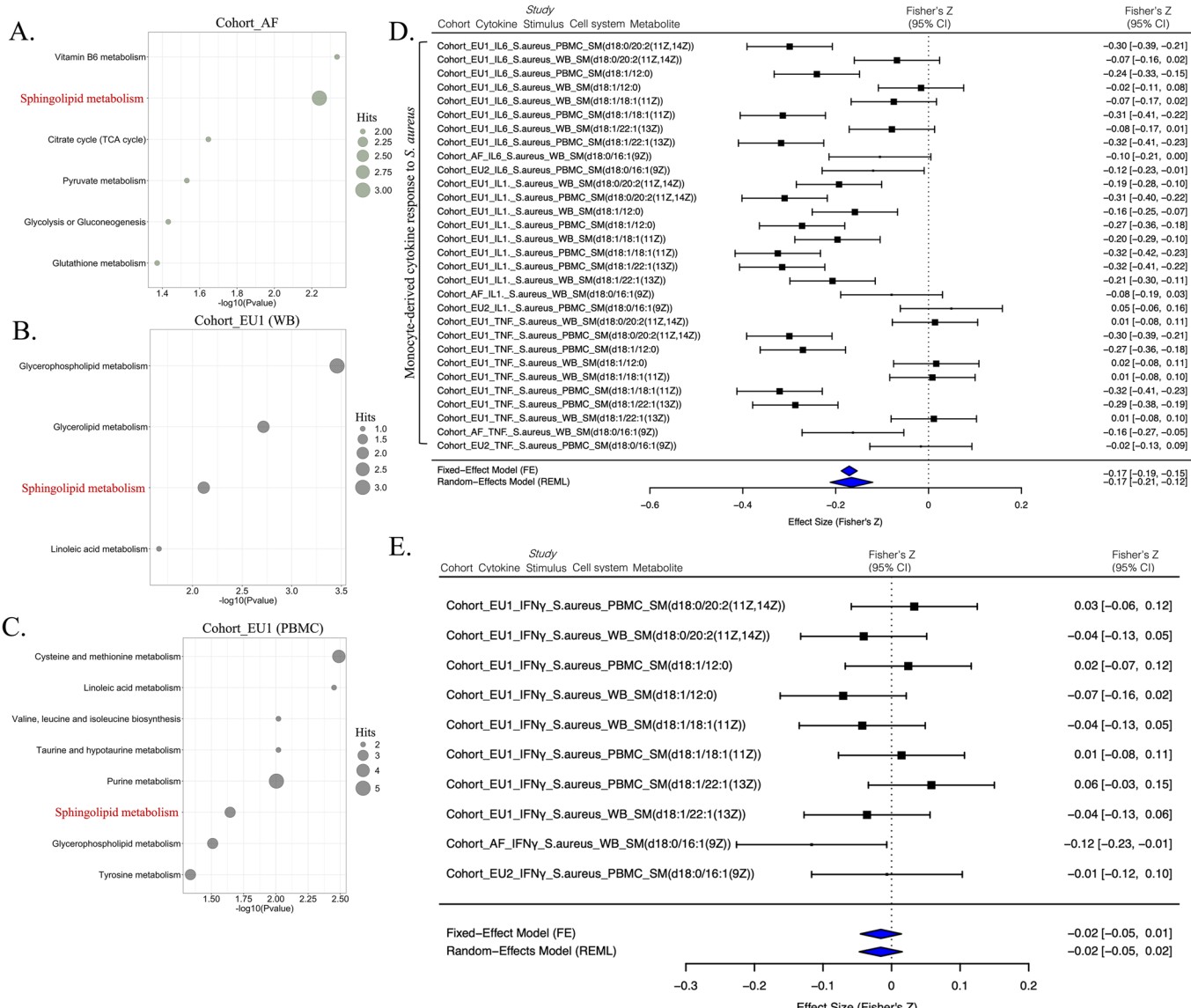

**Figure 4. Immune response-related metabolites across multiple cohorts.**

Bubble plots showing the results of pathway analysis of metabolites significantly correlated (FDR < 0.01) with *S. aureus*-induced cytokine responses (IL-1β, IL-6, TNF, and IFN-γ) in Cohort_AF (**A**), Cohort_EU1 (**B**), and Cohort_EU1 (PBMCs) (**C**). Pathway enrichment was performed using a hypergeometric test. (**D, E**) A "forest" plot displaying different mean values (center of symbols), confidence limits (95% confidence intervals), and precision levels (denoted by the size or "weight" of the symbols, where larger symbols signify higher precision). These are shown for the effect sizes derived from individual studies (in black), as well as the aggregate mean values (center of symbols) and 95% confidence intervals (width of symbols) calculated through meta-analysis using both a fixed-effect model and a random-effects model (both in blue). Panel (**D**) is based on n = 30 cytokine–metabolite associations, and panel (**E**) is based on n = 10 cytokine–metabolite associations. Source data are available online for this figure.

## Genetic analysis shows the causal role of Sphingomyelins in COVID-19

The results described above suggest an inhibitory role of SM metabolite on innate immune function. We next examined whether SM could be a potential modulator for cytokine responses and may hold potential therapeutic significance in diseases, such as managing the cytokine overproduction in COVID-19 patients that can lead to intense inflammation, organ damage, and even death (Del Valle et al, 2020). Firstly, we used a public metabolomics dataset comprising 198 individuals with COVID-19 of varying

severities (Lee et al, 2022) to examine the association between all sphingomyelin metabolites and the disease severity. The concentrations of sphingomyelins varied among healthy donors and different severity groups of COVID-19 patients: mild, moderate, and severe (Appendix Fig. S10). In general, the concentration of the majority of SM metabolites (Appendix Fig. S10) was significantly higher in the healthy donors compared to the COVID-19 patients and was negatively associated with COVID-19 severity. For several SM metabolites, such as SM (d18:0/18:0, d19:0/17:0), SM (d18:1/20:2, d18:2/20:1, d16:1/22:2), SM (d18:1/22:2, d18:1/22:1, d16:1/24:2), and SM (d18:2/24:2) (Appendix Fig. S10I,L,Q,U,

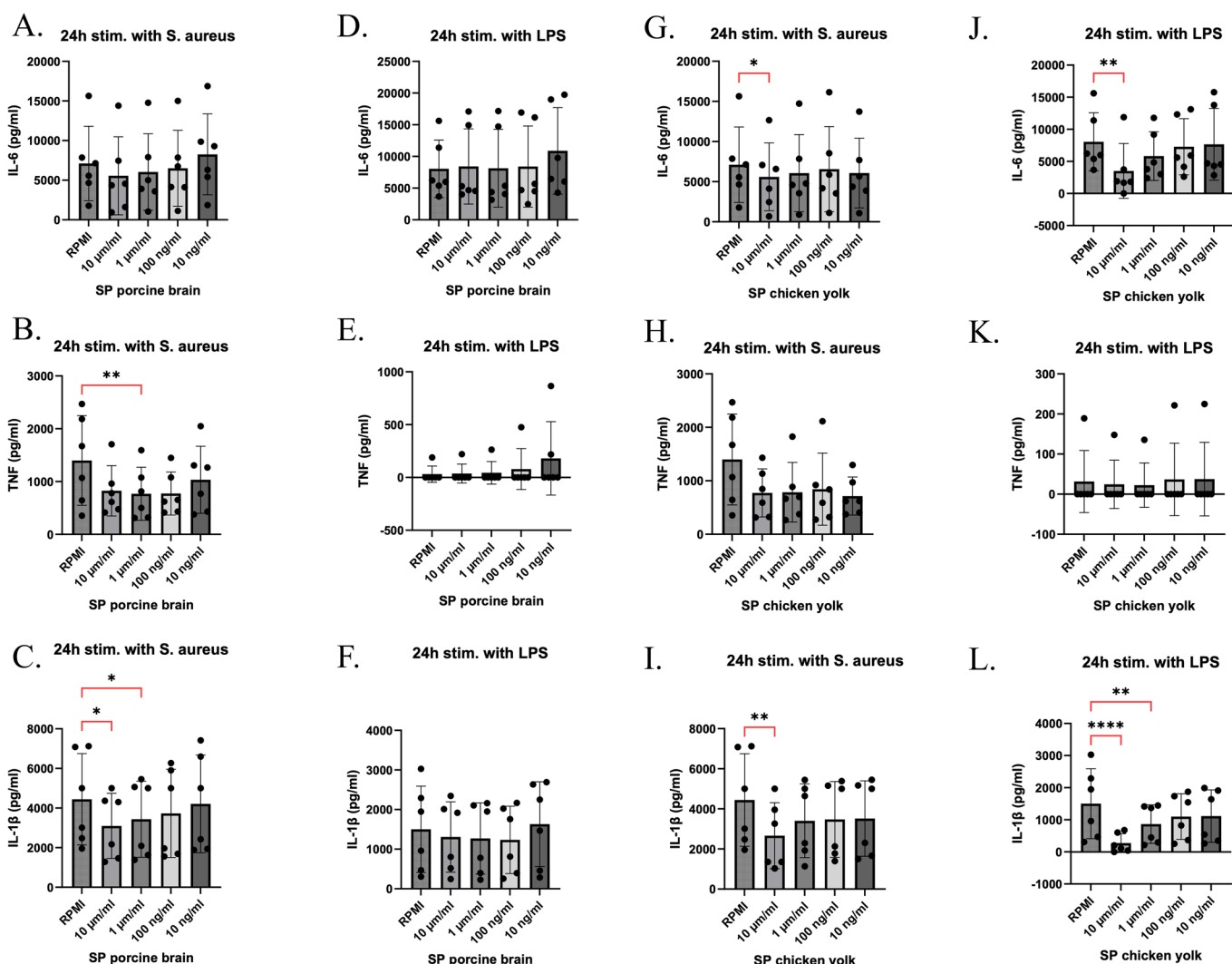

**Figure 5.  Human PBMCs were stimulated with LPS or heat-killed *S. aureus* in the absence of presence of chicken yolk or porcine brain sphingomyelin.**

Human PBMCs from six healthy donors (two independent experiments, $n = 6$) were pre-incubated for 1 h with vehicle control (RPMI) or increasing concentrations (10 μM, 1 μM, 100 ng/mL, 10 ng/mL) of sphingomyelin isolated from porcine brain (A–F) or chicken egg yolk (G–L), then stimulated for 24 h with heat-killed *Staphylococcus aureus* (A–C, G–I) or LPS (D–F, J–L). Cytokine levels in supernatants were measured by ELISA for IL-6 (A, D, G, J), TNF (B, E, H, K), and IL-1β (C, F, I, L). Bars represent mean ± SEM with individual donor values overlaid as dots. For (K), boxplots represent the median (center line), the 25th and 75th percentiles (bounds of the box), whiskers indicate the minimum and maximum values. Statistical analysis was performed using Friedman nonparametric ANOVA followed by Dunn's multiple comparisons test versus RPMI control. Statistical significance versus RPMI control is indicated by brackets: *$P < 0.05$, **$P < 0.01$, ***$P < 0.001$, ****$P < 0.0001$. Exact adjusted $P$ values (Dunn's test): (B) RPMI vs. 1 μM/mL, $P = 0.0076$; (C) RPMI vs. 10 μM/mL, $P = 0.0139$; RPMI vs. 1 μM/mL, $P = 0.0423$; (G) RPMI vs. 10 μM/mL, $P = 0.0423$; (I) RPMI vs. 10 μM/mL, $P = 0.0010$; (J) RPMI vs. 10 μM/mL, $P = 0.0041$; (L) RPMI vs. 10 μM/mL, $P < 0.0001$; RPMI vs. 1 μM/mL, $P = 0.0041$. Source data are available online for this figure.

respectively), patients with moderate symptoms showed significantly higher in SM concentrations compared to the healthy control group.

We next examined the causal relationship between sphingomyelin metabolites and COVID-19 using the previously reported SM-associated variants (mQTLs, or metabolic quantitative trait loci) (Long et al, 2017) and the public GWAS summary statistics of COVID-19 (Initiative, 2020), using the Mendelian randomization (Smith and Ebrahim, 2003) (MR) method. Using 10 independent SNPs ($P < 5.0 \times 10^{-7}$ and clumping variants with linkage disequilibrium $r^2 < 0.001$) as instruments, the results of two commonly used MR methods, i.e., weighted median estimator and inverse-variance

weighted (Bowden et al, 2017; Burgess et al, 2015), consistently showed that a decrease in circulating sphingomyelin concentration had a causal effect on COVID-19 severity ($P = 4.79 \times 10^{-2}$, and $7.92 \times 10^{-3}$, respectively; effect sizes = −0.12, and −0.13, respectively; Fig. 6A,B; Table EV5). A forest plot of the 10 plasma SM SNPs associated with the risk of COVID-19 is shown in Fig. 6C. The plot shows each SNP's effect size and 95% Confidence Interval (CI) with points and lines; the negative estimates suggest that higher SM concentrations decrease the risk of developing severe COVID-19 symptoms.

To further assess robustness, we extracted instruments at two additional thresholds: a more stringent $P < 5 \times 10^{-8}$ (6 SM-related

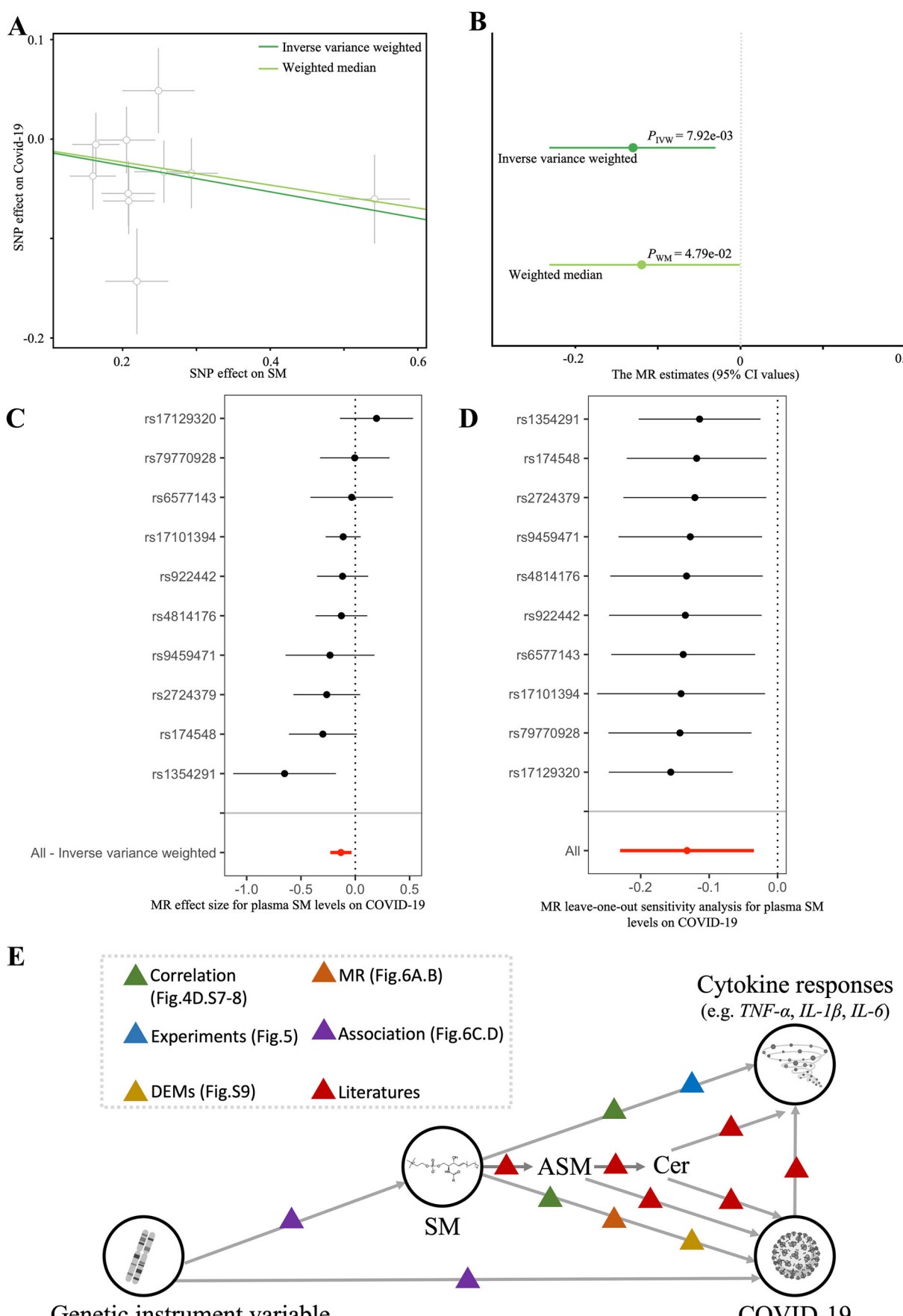

**Figure 6. Causal relationships between sphingomyelin and COVID-19 as assessed by MR analysis.**

(A) The regression lines for the inverse-variance weighted (IVW), weighted median (WM) method is shown ($n = 10$ SNPs). Error bars indicate 95% confidence intervals. (B) Forest plot of the MR estimates of the effects of SM on COVID-19 ($n = 10$ SNPs), where points represent MR effect estimates and error bars represent 95% confidence intervals. (C) Forest plot of the individual effects of the 10 plasma SM-associated SNPs on COVID-19 risk, with points representing per-SNP MR effect estimates and error bars indicating 95% confidence intervals. (D) Leave-one-out sensitivity analysis based on the 10 SNP instruments, where points represent MR estimates excluding each SNP in turn and error bars indicate 95% confidence intervals. (E) A graphic summary of the regulation network of sphingomyelin, cytokine responses, and COVID-19. Source data are available online for this figure.

mQTLs) and a more relaxed $P < 1 \times 10^{-6}$ (21 mQTLs). At $P < 1 \times 10^{-6}$, both methods remained significant (weighted median $P = 3.85 \times 10^{-2}$, $\beta = -0.11$; IVW $P = 5.27 \times 10^{-3}$, $\beta = -0.12$), and even under the more stringent threshold with fewer instruments ($P < 5 \times 10^{-8}$), the effect remained in the same direction ($\beta = -0.13$, $P = 7.16 \times 10^{-3}$). These analyses are summarized in Supplementary Appendix Fig. S11A–C,E–G and detailed in Table EV5.

Considering the heterogeneity of SNP effects, we went further to apply an aggregated approach, the Inverse Variance Weighted (IVW) method. As the red line indicates, there is a collective trend where higher SM levels correspond to a lower risk of developing severe COVID-19, providing a more reliable overall estimate. To validate the adherence of our data to MR assumptions, we conducted a series of sensitivity analyses. These included tests for horizontal pleiotropy, as indicated by a non-significant MR-Egger intercept ($P > 0.05$, see Table EV6 for $P < 5 \times 10^{-7}$, $P < 1 \times 10^{-6}$ and $P < 5 \times 10^{-8}$ results), assessments of heterogeneity via Cochran's Q test ($P > 0.05$, shown in Table EV7), and a leave-one-out analysis (illustrated in Fig. 6D; Appendix Fig. S11D,H). We refrained from using causal estimates obtained through the MR-Egger method as a filtering criterion due to its relatively low power in detecting causality (Bowden et al, 2015). To sum up, the robustness of the MR causal estimates was further substantiated by these sensitivity analyses.

In line with our observation of the negative correlation between SM and COVID-19 severity, previous research indicated a significant decrease in sphingomyelin (SM) concentrations and a notable increase in ceramide (Cer) concentrations in severe COVID-19 patients (Petrache et al, 2023). Previous studies have also reported that the SARS-CoV-2 virus can activate acid sphingomyelinase (ASM) and the Sphingomyelinase-Ceramide pathway (Abusukhun et al, 2021), and Cer can promote increased cytokine secretion (Boon et al, 2013; Giltiay et al, 2005; Norris and Blesso, 2017). We therefore propose that the interactions between sphingomyelin (SM), enzymes of the Sphingomyelinase-Ceramide pathway, and Cer, modulate cytokine secretion and disease severity in COVID-19 patients (Fig. 6E). This regulatory framework connects genomic variations to COVID-19 diseases by modulating gene expression, metabolite profiles, and immune responses. This framework is constructed using multi-omics data sourced from various cohorts, public repositories, and existing literature.

### An online tool for exploring metabolic features for immune functions in human

To aid in exploring the intricate connections between metabolite features and immune function, we established the IMetaboMap online tool (https://lab-li.ciim-hannover.de/apps/imetabomap/), a pioneering resource that catalogs interactions between plasma metabolites and cytokine responses to various stimuli in humans. This tool will prove instrumental in examining how these interactions differ among various populations, between sexes, and across different tissue types. Our analysis showed that Cohort_EU1 had 125,307 metabolite–cytokine connections, while Cohort_EU2 and Cohort_AF yielded 28,833 and 80,550 connections, respectively, totaling 234,690 unique connections recorded in IMetaboMap. This tool stands as a testament to our comprehensive approach, enabling detailed exploration of the dynamic relationships between metabolites and cytokines, and shedding light on specific associations that may vary by sex and population. This extensive mapping effort is visualized in Fig. 7, which serves as a foundational reference for ongoing and future research into the mechanisms and functions of immunometabolism, advancing our understanding of immune-metabolic interplay and paving the way for effective therapeutic interventions.

## Discussion

In this study, we conducted an in-depth analysis of the interplay between plasma metabolite features and cytokine response to various stimuli across multiple cohorts with diverse ethnicities including two Western European cohorts and one East-African cohort from Tanzania. An online tool, IMetaboMap, has been developed to offer insights into the interrelationships between metabolites and cytokine responses across different populations and sexes. Furthermore, we particularly highlighted the pivotal role of baseline metabolites in immune modulation and COVID-19.

Benefiting from the uniqueness of the data from multiple cohorts with different populations, we systematically explored the relationship between metabolites and immune phenotypes. First, we identified glycerophospholipid metabolism, a common key metabolic pathway, across different populations, to play an important role in regulating immune responses and cytokines production by human circulating immune cells (Mendes-Frias et al, 2020; Wu et al, 2021). Further, in the analysis of sex differences, we found that phosphatidylcholine, an important component of the glycerophospholipid metabolic pathway, can serve as a sex-specific metabolic marker, which is consistent with previous studies on sex differences in these metabolites (Rauschert et al, 2017). These findings not only provide valuable evidence for the study of the glycerophospholipid metabolic pathway, but also indicate potential targets for the study of immune-related diseases.

An important finding highlights the importance of the sphingolipid metabolism pathway in the regulation of immune responses, as well as the potential of sphingolipids in modulating various immune responses and their therapeutic implications. In particular, the concentration of sphingomyelins significantly

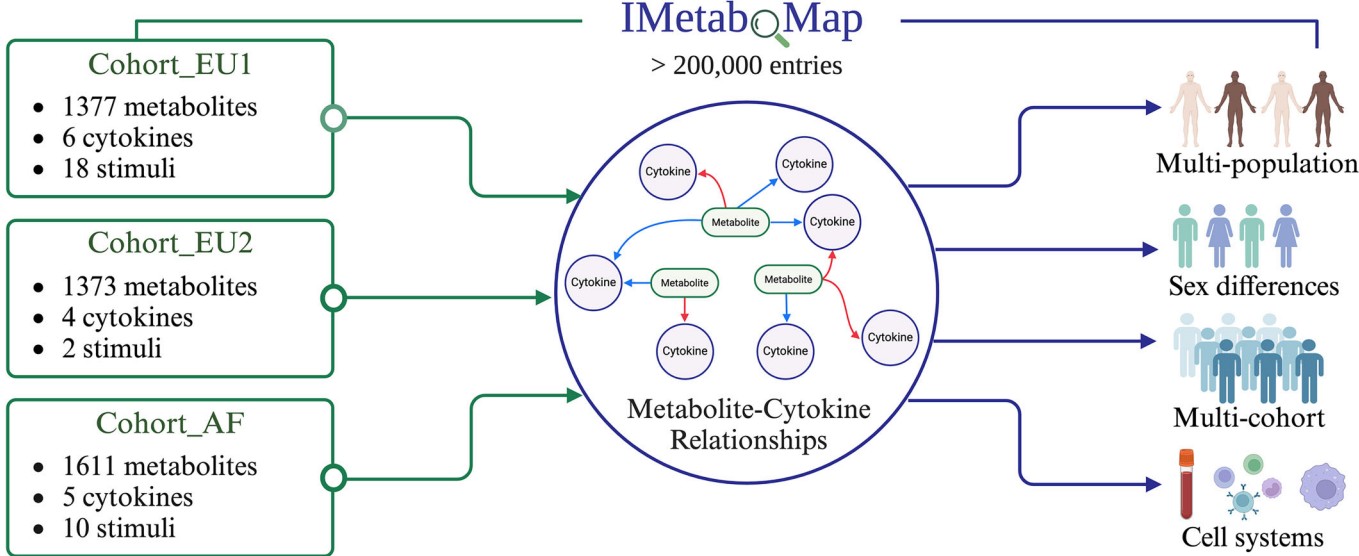

**Figure 7. IMetaboMap is a web-based platform designed for users to explore correlations between metabolites and cytokine responses interactions.**

This pioneering tool offers insights into the interplay between plasma metabolites and cytokine responses to various stimuli in humans, with the capability to analyze variations across population, sex, and cell system.

correlates with monocyte-derived cytokines response (TNF, IL-1β, IL-6), but not with T-cell-derived cytokines response (IFN-γ) to *S. aureus* stimulation (Fig. 4D,E). This implies a stronger role for sphingomyelins in the modulation of innate immune responses rather than in the adaptive responses. The mechanism by which sphingomyelin influences immune responses are likely complex, involving cellular processes such ascell signaling (Olivera and Rivera, 2005), membrane fluidity (Shiao and Vance, 1993), and intercellular interactions (Mathias et al, 1993). The role of sphingomyelin in the modulation of innate immunity, as observed in our study, might be due to its localization within cell membranes and its participation in the immediate cellular response upon recognition of pathogen- and danger-associated molecular patterns. It also suggests that sphingomyelin predominantly participates in immediate and ubiquitous defense mechanisms, such as inflammation or rapid responses to pathogens, rather than specialized, long-term defense strategies involving memory lymphocytes and antibodies. The sphingomyelin–cytokines response interaction described here supports previous studies which suggested that cytokines released by innate immune cells are more strongly influenced by circulating metabolites (Chu et al, 2021; Garand et al, 2017). Thus, sphingomyelin metabolites could be potential targets for therapy in inflammation-mediated diseases.

Our study has several important strengths. First, the complementary, but independent, identification of metabolite-immune interaction in omics studies, functional experiments, and genetic Mendelian randomization provides very strong arguments for the validity of our conclusions. Second, our study proved the relevance of these findings by the identification of an association between sphingomyelin concentrations and COVID-19 severity. Whereas the significant mortality associated with COVID-19 is closely linked to the "cytokine storm" (Del Valle et al, 2020), our findings indicate an inverse relationship between elevated SM concentrations and the severity of COVID-19 infection. This suggests that modulating SM levels might regulate cytokine production, attenuate inflammatory responses, and potentially reduce mortality. In line with our findings, the presence of sphingolipids in the milk fat globule membrane has demonstrated anti-bacterial properties against several microorganisms (Sprong et al, 2001), while milk sphingomyelin could mitigate LPS-induced inflammation in RAW264.7 macrophages (Norris et al, 2017). Finally, our findings have been validated in cohorts of individuals with both European and African ancestry. The vast majority of studies to date investigate solely individuals of European ancestry, while knowledge in non-European populations is very limited. Our study aimed to improve to knowledge on immune response regulation in both European and African populations. To achieve this aim, we have also built an online tool that can be used to efficiently explore our findings as an important research and development resource.

While our study provides extensive data on the cohorts and offers valuable insights, it still has some potential limitations. First, the variation in stimulation conditions across cohorts—ranging from macrophages to PBMCs to whole blood—stemmed from practical constraints in sample availability and historical experimental designs, enabling a broad survey of immune responses but also introducing variability that limits direct comparisons and may increase false positives. In contrast, *S. aureus* stimulation was applied under identical protocols in all three cohorts (same cell type, incubation times, and cytokine panel), providing the largest combined sample size and greatest statistical power to identify robust, cross-cohort metabolite–cytokine associations. Other ligands, tested in fewer cohorts or with divergent protocols, lacked sufficient sample size and assay consistency to produce stable findings after FDR correction. Future studies should adopt harmonized, multi-stimulus designs across cohorts to better explore the diversity of immune-metabolic interactions. In addition, our study design included both whole blood (WB) and peripheral blood mononuclear cell (PBMC) assays, each with distinct advantages and

limitations. The WB stimulation system, containing a mix of immune and non-immune cells, is subject to greater inter-individual variability and reflects the complexity of in vivo immune responses. In contrast, the PBMC assay, with standardized cell numbers and exclusion of non-PBMC elements, yields more reproducible cytokine measurements. Differences in cell composition and abundance, particularly for monocyte and T-cell subsets, are likely to influence cytokine outputs in both systems. Although our findings suggest specific associations between sphingolipid metabolism and monocyte-derived cytokine production, we did not quantify immune-cell subpopulations in this study. Future work including detailed immune-cell phenotyping will be required to distinguish effects of cell abundance versus activation state. Second, confounding factors such as age, lifestyle, and genetic background, dietary habits (Gonzalez-Granda et al, 2018; Playdon et al, 2016) can influence individual blood metabolite profiles. However, the consistent negative association between the sphingolipid metabolism pathway and cytokine responses across different cohorts and populations emphasizes its role in immune function and reassures on the validity of our conclusions. Third, while we employed the same mass spectrometry platform (flow-injection TOF-M) to measure metabolites across cohorts, this technique cannot precisely identify the isomers of metabolites. Since very long-chain SM and long-chain SM in plasma can play different roles in modulating inflammation (Sakamoto et al, 2017), we also observed that the concentration of some SMs did not show a similar declining trend with the severity of COVID-19 as the majority of SMs (Appendix Fig. S10). Thus, future research might delve deeper into the roles of different SM isomers in infectious diseases. Fourth, although our findings demonstrate that sphingomyelin modulates immune responses following *S. aureus* stimulation, we acknowledge that extending these observations to broader immune contexts requires further validation. Future experiments involving a wider variety of pathogens and immunostimulants are essential to confirm and generalize the role of sphingomyelin metabolism across diverse immune challenges and disease conditions. Five, although we identified sex-specific metabolic markers potentially linked to immune regulation, the current study does not experimentally validate their functional, sex-specific impacts. Future investigations incorporating targeted functional assays—such as sex-stratified immune-cell stimulation experiments and mechanistic analyses—are required to robustly confirm these markers' roles in immune regulation and assess their clinical implications. While our results highlight potential sex-specific differences in immune-metabolic associations, inconsistent replication across cohorts limits their immediate clinical applicability. Therefore, further experimental validation and larger-scale, sex-stratified analyses are essential to substantiate the robustness and clinical relevance of these findings. In addition, our in vitro validation was limited by PBMC availability, and cytokine measurements were only taken at 24 h—missing potential early secretion peaks. The small donor cohort further introduces inter-individual variability. The lack of a clear dose–response and the absence of a robust TNF response to LPS stimulation was surprising and is most likely technical. This limitation should be addressed in future studies. Further work should expand the functional assays to include expanded timeframes of stimulation. Finally, while our MR analysis suggests a causal link between circulating sphingomyelin levels and COVID-19 severity, we acknowledge MR's inherent limitations—particularly the potential for pleiotropy and horizontal genetic effects—and therefore interpret these findings as suggestive rather than definitive. Additional confirmatory biological evidence, including targeted in vitro and in vivo experiments, will be essential to fully substantiate and clarify these

associations. Moreover, the COVID-19 dataset used here included a high proportion of patients with comorbidities (e.g., hypertension and diabetes mellitus) and did not adequately control for sex or regional differences. These factors can alter metabolomic profiles and thus limit the generalizability of our results. Future studies should account for these confounders to reinforce metabolite–disease associations. In summary, our study describes the interplay between metabolic signatures and immune functions across different cohorts and ethnic backgrounds, highlighting the potential of using metabolites as immunological modulators. The associations between metabolite–cytokine production revealed in this study are accessible for future research through an online tool known as IMetaboMap. In particular, we identified that sphingolipid metabolism was associated with cytokine responses and disease risk in severe infection (COVID-19). Sphingomyelins can mitigate inflammatory responses, providing tantalizing suggestions that diets rich in sphingomyelin may be useful for the modulation of inflammation in immune-based diseases. These findings provide valuable perspectives on the interplay between sphingolipid metabolism and immune responses in infectious diseases, although further studies are needed to validate and expand upon these observations.

# Methods

**Reagents and tools table**

| Reagent/resource | Reference or source | Identifier or catalog number |
| --- | --- | --- |
| **Experimental models** | | |
| Human PBMCs | Six healthy volunteers (ethics NL84281.091.23, Arnhem-Nijmegen) | N/A |
| Heparinized whole blood | Same donors; collected in EDTA Vacutainer tubes | N/A |
| **Recombinant DNA** | | |
| **Antibodies** | | |
| **Oligonucleotides and other sequence-based reagents** | | |
| **Chemicals, enzymes, and other reagents** | | |
| EDTA Vacutainer tubes (10 mL) | Becton Dickinson (Vacutainer system) | Cat# 367525 |
| Ficoll-Paque PLUS | Cytiva (formerly GE Healthcare) | Cat# 17-1440-02 |
| LPS (E. coli O55:B5) | Sigma-Aldrich | Cat# L2880 |
| Poly(I:C) | InvivoGen | Cat# tlrl-pic |
| RPMI 1640 Medium (Dutch modification) | Gibco | Cat# 21875034 |
| Sodium pyruvate | Gibco | Cat# 11360070 |
| GlutaMAX | Gibco | Cat# 35050061 |
| Gentamicin | Centraform | Cat# 2275 |
| Sphingomyelin (porcine brain-derived) | Sigma-Aldrich | Cat# 383907-91-3 |
| Sphingomyelin (chicken yolk-derived) | Sigma-Aldrich | Cat# 85187-10-6 |

| Reagent/resource | Reference or source | Identifier or catalog number |
|---|---|---|
| IL-1β DuoSet ELISA kit | R&D Systems | Cat# DY201-05 |
| IL-6 DuoSet ELISA kit | R&D Systems | Cat# DY206-05 |
| TNF-α DuoSet ELISA kit | R&D Systems | Cat# DY210-05 |
| IFN-γ ELISA kit | Sanquin | Cat# 75-0528-46 |
| **Software** | | |
| R | R Foundation for Statistical Computing | v4.3.2 |
| ggplot2 (R package) | CRAN | v3.4.3 |
| Shiny (R package) | CRAN | v0.13.1 |
| MetaboAnalyst | MetaboAnalyst.ca | v5.0 |
| WGCNA (R package) | Bioconductor | v1.71 |
| metafor (R package) | CRAN | v3.8.1 |
| TwoSampleMR (R package) | CRAN | v0.5.6 |
| corr.test (psych R package) | CRAN | v2.2.9 |
| **Other** | | |
| Agilent 6520 Q-TOF MS | Agilent Technologies | Model 6520 |
| Agilent 1100 LC pump | Agilent Technologies | Series 1100 |
| Gerstel MPS2 autosampler | Gerstel | MPS2 |
| Ella platform | ProteinSimple | Simple Plex cartridge |

## Cohorts' descriptions

The first cohort, designated as Cohort_EU1, is part of the Human Functional Genomics Project as 500FG cohort and comprises 534 healthy Caucasian individuals of Western European ancestry, with ages spanning from 18 to 75 years. This cohort was carefully selected to exclude individuals with mixed genetic backgrounds or chronic diseases. Key measurements within Cohort_EU1 included cytokine production in response to various stimulations and comprehensive metabolomic profiling. More detailed information can be found in previous publications (Li et al, 2016). The second cohort from Western Europe, Cohort_EU2, included 324 healthy volunteers of Western European descent, aged between 18 and 71 years. These participants were enrolled in the 300BCG cohort from April 2017 to June 2018, with further details documented in a previous publication (Koeken et al, 2020). The third cohort, Cohort_AF, encompasses 323 healthy Tanzanians between 18 and 65 years old from the Kilimanjaro region, who were recruited through the Kilimanjaro Christian Medical Center and Lucy Lameck Research Center from March to December 2017, as previously described (Temba et al, 2021).

## Plasma metabolome measurement and analysis

Untargeted metabolomics measurements from plasma were performed by high-throughput flow-injection time-of-flight mass spectrometry (Fuhrer et al, 2011). The platform employed an Agilent 6520 Series Quadrupole Time-of-flight mass spectrometer and Agilent Series 1100 LC pump coupled to a Gerstel MPS2 autosampler. The metabolites were matched and annotated with HMDB (www.hmdb.ca), KEGG (www.genome.jp/kegg/) and ChEBI (www.ebi.ac.uk/chebi/) identifiers.

The NOREVA platform (http://idrblab.cn/noreva/) was employed to perform comprehensive data analysis using the peak intensity table of annotated metabolites (Fu et al, 2022; Yang et al, 2020). Log transformation and Pareto scaling were applied before data analysis. Pathway enrichment analysis of identified metabolite lists was performed using the pathway analysis (Hypergeometric test) function of MetaboAnalyst V5.0 (https://www.metaboanalyst.ca/) (Pang et al, 2022). The KEGG library was selected as the reference pathway library.

## Whole blood (WB) or peripheral blood mononuclear cells (PBMCs) stimulations

In Cohort_AF, cytokine responses were measured using WB only; in Cohort_EU2, using PBMCs only; and in Cohort_EU1, both WB and PBMC stimulations were performed. For WB stimulations, 100 μl of heparin blood was diluted 1/5 with culture medium containing stimuli in 48-well plates and incubated for 48 h with 100 ng/ml *E. coli*-derived *LPS*, 50 μg/mL Poly(I:C), $10^6$/mL *C. albicans*, $10^6$/mL *S. aureus*, 5 μg/mL *M. tuberculosis* (MTB), $10^6$/mL *E. coli*, $10^7$/ml *C. burnetii*, $10^7$/ml *S. pneumonia*, $10^6$/mL *S. typhimurium* or $10^6$/mL *S. enteritidis*.

In all, $5 \times 10^5$ PBMCs per well were stimulated in round-bottom 96-well plates with 5 μg/mL *M. tuberculosis* (MTB) or $10^6$/mL *S. aureus* for 24 h or 7 days. RPMI 1640 Medium (Dutch modification, Gibco) supplemented with 1 mM sodium pyruvate (Gibco), 2 mM GlutaMAX (Gibco), and 5 μg/ml gentamicin (Centraform) was used in all cell culture experiments. Supernatants were collected and stored at −20 °C until cytokine quantification by ELISA. Cytokines were quantified using DuoSet ELISA Development Systems (R&D), except for the IFN-γ ELISA (Sanquin), according to the manufacturer's instructions.

## Measurement of circulating cytokine concentrations

In the African cohort, cytokine concentrations in plasma were measured with the Ella platform (ProteinSimple) using Simple Plex cartridges following the manufacturer's protocols. In the European cohort, measurement of biomarker concentrations was performed using the Olink Inflammation Panel consisting of 92 markers (Olink Biosciences). This method employs proximity extension assay and provides relative protein quantification expressed as normalized protein expression (NPX) values (Assarsson et al, 2014).

## Statistical analyses

To elucidate the metabolic networks underlying immune phenotypes, we assessed the metabolite co-expression networks of

immune responses (IL-1β, IL-6, TNF, IFN-γ) following *S. aureus* stimulation. Utilizing Weighted Correlation Network Analysis (WGCNA) (Langfelder and Horvath, 2008), we identified modules of metabolites exhibiting high correlations (Appendix Fig. S1). We proceeded to examine the relationship between the summary profile (eigengene) of each module and different immune responses (IL-1β, IL-6, TNF, and IFN-γ), as depicted in Fig. 2. Upon discovering any modules with significant correlations to immune responses, we isolated the metabolites within these modules for further exploration. Pathway analyses were conducted using MetaboAnalyst 5.0. We reported metabolic pathways achieving a threshold of $P < 0.05$, which are presented.

We first assessed the extent to which age, sex, and BMI confounded raw metabolite–cytokine correlations. In three cohorts (Cohort_AF, Cohort_EU1 and Cohort_EU2), we performed Spearman rank correlations between metabolite features and each demographic variable (sex, age, BMI), applying Benjamini–Hochberg FDR correction (FDR < 0.05). Before adjustment, the proportions of metabolites significantly associated with each covariate were as follows (Appendix Fig. S6): Age: 50.7% (Cohort_AF), 32.6% (Cohort_EU1), 22.1% (Cohort_EU2); Sex: 43.9% (Cohort_AF), 53.1% (Cohort_EU1), 39.9% (Cohort_EU2); BMI: 20.6% (Cohort_AF), 18.4% (Cohort_EU1), 0% (Cohort_EU2).

To remove these effects, we fitted a multivariable linear regression model for each metabolite in R (v4.x). In Cohort_AF and Cohort_EU1—where ≥15% of metabolites were BMI-associated—we included sex, age, and BMI as covariates. In EU2, where no metabolites showed BMI associations, we included only sex and age to avoid over-adjustment. After model fitting, we extracted the residuals and re-tested their Spearman correlations with sex, age and BMI (FDR < 0.05). Post-adjustment, fewer than 1% of metabolites remained significantly correlated with any covariate in all cohorts (Cohort_AF: 0.7% age, 1.2% sex, 0.7% BMI; Cohort_EU1: 0.1% age, 0.1% sex, 0% BMI; Cohort_EU2: 0% age, 0% sex, 0% BMI; Appendix Fig. S12), demonstrating effective confounder removal.

For downstream metabolite–cytokine association testing, all analyses were conducted on these de-confounded residuals. We again performed Spearman correlations against each immune cytokine and retained only those metabolite features with FDR < 0.05 in at least one cytokine comparison for further analysis. All model fitting and residual extraction used R's lm() function, correlations were computed with corr.test(), and multiple testing correction was applied via p.adjust (method = "fdr").

## Statistical evaluation of sex-specific metabolite–cytokine correlations

To evaluate whether metabolite–cytokine associations differ between males and females, we first stratified each cohort by sex. Within each sex-specific subgroup, individual metabolite abundances were regressed on age and BMI to remove confounding effects; the resulting residuals were then used in all downstream correlation analyses. For each stimulus–cell condition, Spearman correlation coefficients between residualized metabolite levels and cytokine concentrations were calculated separately in the male and female groups. Each pair of sex-specific Spearman r values (r_male and r_female) was then transformed to Fisher Z scores, and a two-sample Z test was performed to assess whether the

difference

$$Z_{diff} = \frac{Z_{male} - Z_{female}}{\sqrt{\frac{1}{n_{male}-3} + \frac{1}{n_{female}-3}}}$$

was statistically significant. Finally, all $P$ values from the $Z$ tests were adjusted for multiple comparisons using the Benjamini–Hochberg procedure to control the false discovery rate.

## Meta-analysis

To prepare for the meta-analysis, we first converted Spearman's correlation coefficients into Fisher's Z values. This transformation is a crucial step that stabilizes the variances and normalizes the distribution of the coefficients, which is essential for the subsequent pooling of data across different studies. For each Fisher's Z value, we calculated its standard error (SE) using the formula

$$SE(Z) = \frac{1}{\sqrt{N - 3}}$$

where $N$ represents the sample size from the respective study.

In our meta-analysis, we primarily employed a random-effects model to account for between-study heterogeneity, and additionally conducted a fixed-effect model as a robustness check to assess the consistency of pooled effect estimates across studies and to strengthen confidence in our conclusions. Under the fixed-effect model—where all studies share a common true effect—the pooled Fisher's Z estimate ẐFE is given by

$$\hat{Z}FE = \frac{\sum_{i=1}^{k} \omega_i Z_i}{\sum_{i=1}^{k} \omega_i}, \ \omega_i = \frac{1}{SE_i^2}, Var(\hat{Z}FE) = \frac{1}{\sum_{i=1}^{k} \omega_i},$$

where $Z_i$ is the Fisher's Z–transformed correlation from study $i$, and SE$i$ is its standard error. Here, $\omega_i$ represents the inverse-variance weight, so studies with smaller SE$i$ contribute more to the pooled estimate. When heterogeneity was detected, we employed a random-effects model using restricted maximum likelihood (REML) to estimate the between-study variance $\tau^2$. In that case, weights become

$$\omega_i^* = \frac{1}{SE_i^2 + \tau^2}, \ \hat{Z}FE = \frac{\sum_{i=1}^{k} \omega_i^* Z_i}{\sum_{i=1}^{k} \omega_i}, Var(\hat{Z}FE) = \frac{1}{\sum_{i=1}^{k} \omega_i}.$$

This acknowledges that individual studies may estimate different yet related effects by incorporating both within-study and between-study variation. The 'metafor' package (Viechtbauer, 2010) facilitated the computation of pooled effect sizes and their corresponding 95% confidence intervals. We further generated forest plots to visually represent the individual study effects, their confidence intervals, and the pooled estimate, providing a clear and concise graphical summary of the meta-analysis results.

## Composite SM score and meta-analysis of SM–cytokine associations

Within each cohort (Cohort_EU1, Cohort_EU2, and Cohort_AF), all detected sphingomyelin (SM)–related metabolites were first averaged to

create a single "Composite SM score" for each subject. Specifically, the raw abundance values of all annotated SM species in a given sample were summed and divided by the number of SM species, thereby providing a direct measure of overall SM concentration per individual. To account for potential confounding, this Composite SM score was then adjusted for age, sex, and BMI within each cohort, and the resulting residuals were used as the covariate-corrected SM measure.

Next, for each cohort, we assessed the relationship between the Composite SM score and key cytokine outputs (IL-1β, IL-6, TNF, and IFN-γ) under *S. aureus* stimulation. Spearman rank-order correlation coefficients ($r$) were calculated for each Composite SM–cytokine pair, and corresponding $P$ values were adjusted for multiple testing using the Benjamini–Hochberg FDR procedure. Associations with FDR < 0.05 were considered statistically significant.

After extracting the Spearman ρ (Fisher's Z–transformed) and its standard error from each cohort, we performed a cross-cohort meta-analysis for every SM–cytokine combination. A random-effects model (REML) was used as the primary analytic framework to pool effect estimates and 95% confidence intervals; a fixed-effect inverse-variance model was also computed as a robustness check. Heterogeneity was assessed via Cochran's Q and $I^2$ statistics.

## Meta-analysis of pathway enrichment across cohorts

We integrated pathway-level enrichment results from four datasets (EU1_PBMC, EU1_WB, EU2, AF) by computing an enrichment factor for each dataset as

$$EF = \frac{Hits + 0.5}{Expected},$$

and then taking the natural logarithm ($\log EF$) as the effect size. From each pathway's original $P$ value, we derived a two-sided z-score and calculated its standard error as

$$SE = \frac{|\log EF|}{\mathcal{Z}}$$

A random-effects meta-analysis of the cohort-specific $\log EF$ and SE inputs was performed using the metafor R package (rma(…, method = "REML")). We report the pooled $\log EF$, its 95% confidence interval, and heterogeneity statistics ($\tau^2$, $I^2$, and Cochran's Q test). Results are visualized in a forest plot (metafor's forest() function), displaying each cohort's effect size alongside the overall summary estimate. The imprecision and potential bias of the $I^2$ statistic in small meta-analyses have been discussed in detail by von Hippel (von Hippel, 2015), who recommends that confidence intervals for $I^2$ should always be reported and interpreted cautiously.

## Sex-stratified analysis of phosphatidylcholine and phosphatidate associations

In each cohort (AF, EU1_WB, EU1_PBMC, EU2), we first selected all phosphatidylcholine–related metabolites and all phosphatidate–related metabolites, and calculated the arithmetic mean of their abundances to obtain two summary measures per sample: mean_PC and mean_PA. These measures, together with age, sex, BMI, and the cytokine concentrations measured under stimulation, formed the cohort-specific phenotype datasets.

Samples were then stratified by sex, and within each sex group we fitted a linear model to regress out age and BMI from mean_PC (or mean_PA), using the resulting residuals as the covariate-corrected predictors. For males and females separately, we computed Spearman correlation coefficients between the adjusted mean_PC (or mean_PA) and each cytokine, and controlled for multiple testing using the Benjamini–Hochberg FDR procedure (FDR < 0.05 considered significant).

Next, for each cohort–cytokine combination, we compared the male and female correlation coefficients by transforming them via Fisher's Z, calculating their difference and associated $P$ value (with BH correction). Finally, the cohort-specific gender-difference $Z$ scores from all four sources were meta-analyzed using both fixed-effect and random-effects models, and we report the pooled Fisher Z difference, the corresponding Spearman ρ difference with 95% CI, and the $I^2$ heterogeneity statistic.

## Meta-analysis of metabolite–cytokine correlations

We combined correlation results for four key cytokines (IL-1β, IL-6, TNF, IFN-γ) across three cohorts (AF, EU1 and EU2). First, we loaded each cohort's metabolite–cytokine correlation table, renamed and standardized columns (Metabolite, Cytokine, $r$, cohort, $n$), and mapped all cytokine names to a common scheme. We then filtered to the four target cytokines and kept only metabolites present in all three cohorts. For each cohort–cytokine pair, we transformed Spearman's $r$ to Fisher's Z, estimated its variance ($1/(n–3)$) and back-transformed 95% confidence bounds to $r$. We reshaped the data so that AF, EU1_WB, EU1_PBMC and EU2 Z-scores and variances aligned by metabolite–cytokine combination. Wherever at least two cohorts contributed data, we ran fixed-effect and REML random-effects meta-analyses (meta-for::rma), extracting pooled $Z$ (and corresponding r), 95% CIs, $P$ values, Cochran's Q and $I^2$. The complete results are available on the IMetaboMap platform and have been compiled in Dataset EV10, detailing each metabolite–cytokine pairing.

## Mendelian randomization

MR analysis was done using the TwoSampleMR (version 0.5.6) R package. We leveraged GWAS summary statistics from the COVID-19 Host Genetics Initiative Release 5 study of hospitalized versus non-hospitalized COVID-19 cases (Initiative, 2020) (GWAS Catalog accession GCST011080, EBI dataset ebi-a-GCST011080). The main two-sample MR methods used in this study include IVW (Bowden et al, 2017) and weighted median (Minelli et al, 2021). Therefore, MR estimates were calculated using Wald ratios and these Wald ratios were meta-analyzed using the IVW method (Bowden et al, 2017). To ensure the validity of the results, several sensitivity analyses were performed. We excluded MR estimates potentially driven by horizontal pleiotropy (removing results with MR-Egger (Bowden et al, 2015) intercept $P < 0.05$) and heterogeneity (removing results with Cochran's Q test $P < 0.05$). In addition, we carried out leave-one-out analysis (Hemani et al, 2018) to check whether the MR estimates were possibly driven by a single SNP. Multiple testing correction was performed using the Benjamini–Hochberg approach based on IVW $P$ values. To avoid complex causality relationships, we excluded the results that showed a nominally significant MR estimate in the other direction

($P < 0.05$). For MR of plasma SM on COVID-19 severity, we selected SNPs at three $P$ value thresholds—$P < 5 \times 10^{-7}$ (10 SNPs) as our primary instrument set to balance stringency and SNP count, and $P < 1 \times 10^{-6}$ (21 SNPs; more relaxed) and $P < 5 \times 10^{-8}$ (6 SNPs; genome-wide significance) for sensitivity analyses—thereby demonstrating that effect estimates remain consistent regardless of the instrument threshold chosen. This approach is informed by similar strategies in genetic research, where such a threshold has been employed to balance the trade-off between sensitivity and specificity, particularly in contexts where maximizing the discovery of candidate SNPs is crucial (Arvanitis et al, 2020; Li et al, 2022).

### In vitro validation experiments

Venous blood was sampled in sterile 10 mL ethylenediaminetetraacetic acid (EDTA) tubes (Vacutainer system, Becton Dickinson) from six healthy volunteers (ethical approval NL84281.091.23 of the Arnhem-Nijmegen Ethical Committee). Peripheral blood was processed within 1–4 h after collection. Human PBMCs were isolated by differential centrifugation by Ficol-Paque, as previously described (Ajie et al, 2023). PBMCs were stimulated with either lipopolysaccharide (LPS from *E. coli* 055:B5, Sigma) or heat-killed *Staphylococcus aureus* for 24 h, in the absence or presence of different concentrations of chicken yolk or porcine brain sphingomyelin (10 ng/ml to 10 μg/ml). Culture supernatants were collected at the end of the incubation period and stored at −20 °C until cytokines were measured using enzyme-linked immunosorbent assay. The production of pro-inflammatory cytokines was measured using commercial ELISA kits (R&D Duoset ELISA Systems) according to the manufacturer's instructions. Data from repeated measures were analyzed using Friedman nonparametric ANOVA followed by Dunn's multiple comparisons test (GraphPad Prism 10). Adjusted $P$ values are reported for all comparisons.

### Visualization

R package ggplot2 was used to perform most visualizations, including correlation plots, bar charts, and boxplots.

### Online tool implementation details

IMetaboMap was developed by R v4.3.2 and Shiny v0.13.1 running on Shiny-server v1.7.5. Various R packages were utilized in the background processes. IMetaboMap can be readily accessed by all users with no login requirement, and by diverse and popular web browsers including Google Chrome, Mozilla Firefox, Safari and Internet Explorer 10 (or later).

## Data availability

Data from Cohort_EU1, including raw spectral files, are available in MetaboLights under accession number MTBLS2633. Metabolomics data from Cohort_AF have been deposited in the EMBL-EBI MetaboLights database under accession number MTBLS2267. The metadata on study participants, metabolomics, and cytokine measurements from the Cohort_EU2 study are publicly available at https://gitlab.com/xavier-lab-computation/public/bcg300. The processed cytokine and metabolomics data for Cohort_AF, Cohort_EU1, and Cohort_EU2 are available via GitHub (https://github.com/JianboFu0406/IMetaboMap/tree/main/data/Metabolomics%20%26%20Cytokines%20Data).

The source data of this paper are collected in the following database record: biostudies:S-SCDT-10_1038-S44320-025-00146-w.

## Peer review information

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

## Acknowledgements

The authors thank all volunteers from the Radboud University Medical Center and Tanzania for their participation in the study. This work was supported with funds provided by ERC starting grant 948207, I ASPASIA, and Radboud University Medical Center Hypatia grant to YL, ERC advanced grant 833247 and Spinoza grant of the Netherlands Organization for Scientific Research to MGN, and Helmholtz Initiative and Networking Fund to CJX (1800167). This study was also supported by the COFONI (COVID-19 Research Network of the State of Lower Saxony) with funding from the Ministry of Science and Culture of Lower Saxony, Germany (14-76403-184-3) and the Deutsche Forschungsgemeinschaft (DFG; German Research Foundation) under Germany's Excellence Strategy—EXC 2155 project number 390874280 to YL. The study received financial support from the Joint Programming Initiative, A Healthy Diet for a Healthy Life (JPI-HDHL), and ZonMw (the Netherlands Organization for Health Research and Development), within the framework of the 'TransMic' and 'TransInf' projects.

## Author contributions

**Jianbo Fu**: Conceptualization; Data curation; Software; Formal analysis; Visualization; Methodology; Writing—original draft; Writing—review and editing. **Nienke van Unen**: Software; Investigation; Writing—review and editing. **Andrei Sarlea**: Validation. **Nhan Nguyen**: Investigation; Writing—original draft. **Martin Jaeger**: Resources. **Javier Botey Bataller**: Validation; Writing—review and editing. **Valerie A C M Koeken**: Resources. **L Charlotte de Bree**: Resources. **Vera P Mourits**: Resources. **Simone J C F M Moorlag**: Resources. **Godfrey Temba**: Resources. **Vesla I Kullaya**: Resources. **Quirijn de Mast**: Resources. **Leo A B Joosten**: Resources. **Cheng-Jian Xu**: Writing—original draft; Writing—review and editing. **Mihai G Netea**: Supervision; Validation; Writing—original draft; Writing—review and editing. **Yang Li**: Conceptualization; Supervision; Funding acquisition; Writing—original draft; Writing—review and editing.

Source data underlying figure panels in this paper may have individual authorship assigned. Where available, figure panel/source data authorship is listed in the following database record: biostudies:S-SCDT-10_1038-S44320-025-00146-w.

## Funding

## Disclosure and competing interests statement

MGN and LABJ are scientific founders of TTxD and Lemba Therapeutics. MGN is a founder of BioTRIP.

