## [Peer Review File · Molecular Systems Biology]

Deciphering Cross-Cohort Metabolic Signatures of Immune Responses and Their Implications for Disease Pathogenesis

Jianbo Fu, Nienke Unen, Andrei Sarlea, Nhan Nguyen, Martin Jaeger, Javier Bataller, Valerie Koeken, L. Charlotte Bree, Vera P. Mourits, Simone Moorlag, Godfrey Temba, Vesla I. Kullaya, Quirijn de Mast, Leo A.B. Joosten, Cheng-Jian Xu, Mihai Netea, and Yang Li

Corresponding author(s): Yang Li (yang.li@helmholtz-hzi.de)

Review Timeline:

Submission Date:	5th Feb 25
Editorial Decision:	19th Mar 25
Revision Received:	1st Jul 25
Editorial Decision:	25th Jul 25
Revision Received:	22nd Aug 25
Accepted:	26th Aug 25

Editor: Jingyi Hou

Transaction Report:

19th Mar 2025

Manuscript Number: MSB-2025-12900

Title: Deciphering Cross-Cohort Metabolic Signatures of Immune Responses and Their Implications for Disease Pathogenesis

Author: Jianbo Fu

Nienke Unen

Andrei Sarlea

Nhan Nguyen

Martin Jaeger

Javier Bataller

Valerie Koeken

L. Charlotte Bree

Vera P. Mourits

Simone Moorlag

Godfrey Temba

Vesla I. Kullaya

Quirijn de Mast

Leo A.B. Joosten

Cheng-Jian Xu

Mihai Netea

Yang Li

Dear Dr. Li,

Thank you for submitting your work to Molecular Systems Biology. We have now heard back from the three reviewers who agreed to evaluate your manuscript. As you will see from the reports below, the reviewers find the study potentially interesting and relevant. They raise, however, a series of concerns, which should be convincingly addressed in a major revision.

The reviewers' recommendations are relatively clear, so there is no need to reiterate the points listed below. All the issues raised by the reviewers need to be satisfactorily addressed. In particular, the concerns related to statistics, data analysis, in vitro experiments, clarity of presentation should be resolved. The differences between the cohorts as well as their related results and implications need to be clearly outlined and discussed.

As you may already know, our editorial policy allows in principle a single round of major revision, and it is therefore essential to provide responses to the reviewers' comments that are as complete as possible. Please feel free to contact me in case you would like to discuss in further detail any of the issues raised by the reviewers.

On a more editorial level, we would ask you to address the following issues:

- Please provide a .docx formatted version of the manuscript text (including legends for main figures, EV figures and tables). Please make sure that the changes are highlighted to be clearly visible.
- Please provide individual production quality figure files as .eps, .tif, .jpg (one file per figure).
- Please provide a .docx formatted letter INCLUDING the reviewers' reports and your detailed point-by-point responses to their comments. As part of the EMBO Press transparent editorial process, the point-by-point response is part of the Review Process File (RPF), which will be published alongside your paper.
- Please note that all corresponding authors are required to supply an ORCID ID for their name upon submission of a revised manuscript.
- We replaced Supplementary Information with Expanded View (EV) Figures and Tables that are collapsible/expandable online (see examples in <http://msb.embopress.org/content/11/6/812>). A maximum of 5 EV Figures can be typeset. EV Figures should be cited as 'Figure EV1, Figure EV2' etc... in the text and their respective legends should be included in the main text after the legends of regular figures.

Additional Tables/Datasets should be labeled and referred to as Table EV1, Dataset EV1, etc. Legends have to be provided in a separate tab in case of .xls files. Alternatively, the legend can be supplied as a separate text file (README) and zipped together with the Table/Dataset file.

For the figures and tables that you do NOT wish to display as Expanded View figures, they should be bundled together with their legends in a single PDF file called *Appendix*, which should start with a short Table of Content. Each legend should be below

the corresponding Figure/Table in the Appendix. Appendix figures and tables should be referred to in the main text as: "Appendix Figure S1, Appendix Figure S2, Appendix Table S1" etc. See detailed instructions regarding expanded view here: <https://www.embopress.org/page/journal/17444292/authorguide#expandedview>.

-Before submitting your revision, primary datasets (and computer code, where appropriate) produced in this study need to be deposited in an appropriate public database (see <http://msb.embopress.org/authorguide-dataavailability> <https://www.embopress.org/page/journal/17444292/authorguide#dataavailability>). Please remember to provide a reviewer password if the datasets are not yet public. The accession numbers and database should be listed in a formal "Data Availability" section (placed after Materials & Method) that follows the model below (see also <https://www.embopress.org/page/journal/17444292/authorguide#dataavailability>). Please note that the Data Availability Section is restricted to new primary data that are part of this study.

Data availability

-At EMBO Press we ask authors to provide source data for the main figures. Our source data coordinator will contact you to discuss which figure panels we would need source data for and will also provide you with helpful tips on how to upload and organize the files.

- Our journal encourages inclusion of *data citations in the reference list* to directly cite datasets that were re-used and obtained from public databases. Data citations in the article text are distinct from normal bibliographical citations and should directly link to the database records from which the data can be accessed. In the main text, data citations are formatted as follows: "Data ref: Smith et al, 2001". In the Reference list, data citations must be labeled with "[DATASET]". A data reference must provide the database name, accession number/identifiers and a resolvable link to the landing page from which the data can be accessed at the end of the reference. Further instructions are available at .

- We updated our journal's competing interests policy in January 2022 and request authors to consider both actual and perceived competing interests. Please review the policy <https://www.embopress.org/competing-interests> and update your competing interests if necessary.

Please use the heading "Disclosure statement and competing interests".

- All Materials and Methods need to be described in the main text using our 'Structured Methods' format. According to this format, the Methods section includes a Reagents and Tools Table (listing key reagents, experimental models, software and relevant equipment and including their sources and relevant identifiers) followed by a Methods and Protocols section describing the methods, ideally using a step-by-step protocol format. The aim is to facilitate adoption of the methodologies across labs. Please download and fill our Reagents and Tools Table template (.docx), which you can find in our author guidelines: <https://www.embopress.org/page/journal/17444292/authorguide#structuredmethods>.

An example of a Method paper with Structured Methods can be found here: <https://www.embopress.org/doi/10.15252/msb.20178071>.

-Regarding data quantification:

Please ensure to specify the name of the statistical test used to generate error bars and P values, the number (n) of independent experiments (please specify technical or biological replicates) underlying each data point and the test used to calculate p-values in each figure legend. Discussion of statistical methodology can be reported in the materials and methods section, but figure legends should contain a basic description of n, P and the test applied.

Graphs must include a description of the bars and the error bars (s.d., s.e.m.).

- Please provide a "standfirst text" summarizing the study in one or two sentences (approximately 250 characters, including space), three to four "bullet points" highlighting the main findings and a "synopsis image" (550px width and 400-600 px height, PNG format) to highlight the paper on our homepage.

Here are a couple of examples:

<https://www.embopress.org/doi/10.15252/msb.20199356>

<https://www.embopress.org/doi/10.15252/msb.20209475>

<https://www.embopress.org/doi/10.15252/msb.209495>

When you resubmit your manuscript, please download our CHECKLIST (<https://www.embopress.org/pb-assets/embo->

site/EMBO%20Press%20Author%20Checklist-1642513524327.xlsx) and include the completed form in your submission. *Please note* that the Author Checklist will be published alongside the paper as part of the transparent process (<https://www.embopress.org/page/journal/17444292/authorguide#transparentprocess>).

If you feel you can satisfactorily deal with these points and those listed by the referees, you may wish to submit a revised version of your manuscript. Please attach a covering letter giving details of the way in which you have handled each of the points raised by the referees. A revised manuscript will be once again subject to review and you probably understand that we can give you no guarantee at this stage that the eventual outcome will be favorable.

I look forward to receiving your revised manuscript soon.

Sincerely,
Jingyi

Jingyi Hou, PhD
Senior Editor
Molecular Systems Biology

We realize that it is difficult to revise to a specific deadline. In the interest of protecting the conceptual advance provided by the work, we recommend a revision within 3 months (17th Jun 2025). Please discuss the revision progress ahead of this time with the editor if you require more time to complete the revisions. Use the link below to submit your revision:

*** PLEASE NOTE *** As part of the EMBO Press transparent editorial process initiative (see our Editorial at <https://dx.doi.org/10.1038/msb.2010.72>), Molecular Systems Biology publishes online a Review Process File with each accepted manuscripts. This file will be published in conjunction with your paper and will include the anonymous referee reports, your point-by-point response and all pertinent correspondence relating to the manuscript. If you do NOT want this File to be published, please inform the editorial office at msb@embo.org within 14 days upon receipt of the present letter.

Reviewer #1:

Fu et al describe a study to assess associations between circulating plasma metabolites and immune responses, as characterized by secreted cytokine levels after diverse stimulation of PBMCs, macrophages or whole blood. Three cohorts of healthy donors are described including two based in Europe and one in east Africa. The question is interesting and the cohorts of good size (300-500) and having donors from different ethnicities and geographical sites is a strength. They highlight a negative correlation between sphingomyelin and monocyte cytokine response to *Staphylococcus aureus* stimulation.

However many aspects of the study are confusing, and much of the analysis does not give confidence in the reported results. They report three different types of stimulated immune cells, either as macrophages, PBMCs, or whole blood. No justification is given for this strategy and the results are rarely consistent which make the interpretation challenging and increase the considerable risk for false positives described below. For example it's not clear what the inclusion of macrophages only in cohort_EU1 adds. A clearer and more consistent study would have the same stimulation conditions used in all 3 cohorts. While I understand this might not always be possible, the current design and description is unnecessarily complex and reduces confidence in the comparisons.

Related to this and an even bigger concern is the lack of correction for multiple hypothesis testing in the initial correlation analysis. They test a potential 322,218 correlations in cohort 1, 10,984 in cohort 2, and 80,550 in cohort 3. From this they report "nominally significant p values" which presented in this context are challenging to have confidence in. While some of these results are consistent across the cohorts, which should strengthen the confidence of this result, this aspect is not well presented. Why associations are only seen with *S aureus* and almost none of the other 17 stimuli used is not discussed. Is this technical or biological, which could be an interesting question to explore. Again this major omission does not provide confidence in the reported results.

For the sex specific analyses they say "no metabolites in Cohort_EU2 exhibited such correlations, possibly due to the sample size and large variability within the metabolite and cytokine data" but then go on to report significant associations in the AF cohort which is more or less the same size (and likely more heterogenous from a genetic point of view) so it does not support this statement. They also say they identify "sex-specific metabolic markers for innate immune responses" which is a very strong sweeping statement based on the results shown. In some cohorts they see more significant metabolites in males (AF), in the other more in females (EU). This major discrepancy is not mentioned. They then report metabolite associations with "with either cytokine responses IL-1 β , IL-6, TNF, and IFN- γ " but which cytokine response is not specified making it very challenging to interpret. The sex differences they report are also different between the EU and African cohorts perhaps reflecting other underlying differences. Technically speaking, both a fixed-effect and a random-effect model were used in this analysis, yet only one model of age, sex, and BMI was described in the methods. A clear formula for both models would improve clarity for this analysis and the authors decision to use both models, when it seems that heterogeneity was always present.

The authors then propose "in vitro validation" the results of which are the least convincing of the whole manuscript. No statistics are performed, just vague "percentage reduction ranges". A 4 log dose response is tested, which is a good experimental design, but there is clearly no dose effect suggesting no specific inhibitory activity. Did the authors test cellular viability after addition of the metabolites? This is an important control as cell death is a big risk in such experiments which may explain the somewhat lower cytokine response in some conditions. Also why did LPS stimulation alone not induce TNF α secretion (regardless of the metabolite) as expected which is another concern with these results.

Lastly a link is made to Covid using published data and mendelian randomization but the link with S aureus induced cytokines, when nothing was observed for viral stimulations (Influenza, Poly:IC, Cpg) across the 3 cohorts is not clear and adds further confusion to the manuscript. Additionally, non-nominal proportions of the published COVID-19 cohort were hypertensive or diabetic, and there is no discussion of how these comorbidities, as well as the individuals' sex and geographic site, may impact the metabolic findings.

Reviewer #2:

Jianbo Fu et al. present a systematic analysis of the associations between circulating metabolite levels and immune response. The authors analyze multiple cohorts (two European and one from sub-Saharan Africa) demonstrating the robustness of their findings across diverse genetic backgrounds and environmental exposures. Immune response is assessed by measuring ex vivo cytokine production following stimulation with immune triggers such as Staphylococcus aureus. Associations are determined using Spearman correlation, indicating a rank-based rather than linear relationship. The authors emphasize the negative association of sphingomyelins with cytokine production which they validate experimentally. Furthermore, using Mendelian randomization, they establish a link between sphingomyelin levels and COVID-19 severity. They have made their data available through a website (imetabomap) which facilitates querying and visualizing the results.

The presented work provides insights into the role of metabolism in immune response and could provide potential biomarkers and therapeutic targets against immune and infectious diseases. As such, this manuscript and the associated dataset would be quite relevant for researchers in immunology, metabolomics, systems biology and population data science.

However, we have several concerns that we believe that must be addressed before the paper can be considered fit for publication:

We have two major concerns about how the effect of possible confounders was removed. First, why was BMI not accounted for, given its reported influence on the blood metabolome, immune cell composition, and inflammatory cytokines? By the author's own admission, up to 9.73% metabolites are significantly correlated with BMI. Second, the methodology for confounder adjustment is not clearly described. If confounders were regressed out using linear regression, there remains a concern that metabolite levels may still retain a degree of Spearman correlation with these confounders. The authors should demonstrate that, after adjusting for age, sex, and BMI, their data no longer show biologically significant Spearman correlations with these traits.

Sex-specific metabolic markers for immune functions: To support their claim of sex-specific effects, the authors should perform statistical tests to assess that the correlation coefficients are significantly different between male and female volunteers.

Line 224 and Figure 4D-E: The use of meta-analysis to aggregate associations between different sphingomyelins and cytokines is unconventional. If the goal is to show that sphingomyelins as a whole are associated with cytokine production, a more informative approach would be:

1. Compute aggregate measures for sphingomyelins in each sample (e.g. mean or median of relevant sphingomyelin levels)
2. Test association within each cohort
3. Perform a meta-analysis across cohorts for each Cytokine Stimulus Cell System Metabolite aggregate combination.

Alternatively, the authors could use meta-analysis to demonstrate that specific sphingomyelins species negatively correlate with specific cytokines consistently across cohorts. Indeed, given the emphasis on the three separate cohorts, meta-analyses could also be useful to visualize common effects throughout the manuscript. Notably the "Robust plasma metabolic pathways for immune functions across European and African populations" and "Sex-specific metabolic markers for immune functions" sections could benefit from a Meta-Analysis to see if the effects are shared between cohorts. Similarly, in iMetaboMap it would also be informative to include an additional table with the meta-analysis across cohorts.

Mendelian randomization: The use of a threshold of $P < 5 \times 10^{-7}$ for the instrument variants seems somewhat unusual. Could the authors do a sensitivity analysis with lower and higher thresholds to show that the effects are robust? If a lack of sufficient instrument variants is an issue, the authors might also want to consider using more recent studies as sources of mQTLs (e.g. <https://doi.org/10.1038/s41586-023-05844-9>).

Could the authors discuss the differences between the results of whole blood and PBMC? How correlated are the cytokine productions? To what extent do the differences arise from stimulating 100 μ L of whole blood (containing a variable number of PBMCs) versus stimulating a fixed number of PBMCs (5×10^5). We would also like the authors to discuss the role (if any) that immune cell abundance and composition can have on cytokine production (e.g., does higher IFN- γ production indicate more active T cells or a higher number of T cells?)

Minor issues:

- It's often unclear which experimental data was generated in this study and which has already been published in previous articles (e.g., <https://doi.org/10.1038/s41590-018-0121-3>)
- The use of Spearman correlation should be more clearly stated in the Results section and on the iMetaboMap website.
- Line 104: The sentence "Weighted Gene Co-expression Network Analysis" is misleading and may give the impression that gene expression data were analysed instead of correlated metabolite sets.
- Figure 4C: The module order should be consistent across heatmaps to facilitate comparison.
- In "Robust plasma metabolic pathways for immune functions across European and African populations" nominal P values are seemingly used as a significance threshold (line 114). Why was multiple testing correction adjusted P value not used?
- It is unclear whether statistically significant differences were observed in the experimental validation (Figure 5). Given the small sample sizes, a lack of significance would be understandable, but this should be explicitly stated. Additionally, the selection criteria and characteristics of volunteers (e.g., geographic location, sex, age range) should be described.
- It is not immediately clear whether whole blood or PBMC analysis was performed in the Cohort_AF and Cohort_EU2.
- The authors should specify which GWAS was used for Mendelian randomization (e.g. GWAS catalogue identifier).
- Line 435: The filtering step is not well described, making it difficult to understand how and why feature-cytokine pairs were filtered.
- Figure S6 Appears to be a screenshot with highlighted spelling suggestions from Microsoft Word.
- iMetaboMap: There is no clear link to download the data. Additionally, the statement "All rights are reserved by: Prof. Yang Li Group" seems unnecessarily restrictive if the authors intend for others to make use of their dataset.

Reviewer #3:

Understanding of the Story

This study investigates the complex relationship between circulating metabolites and immune responses across different human populations, including two Western European cohorts and one African cohort. The authors explore the interaction of metabolic pathways, especially sphingomyelin and glycerophospholipid metabolism, with cytokine responses. The ultimate aim is to elucidate how these metabolic factors influence disease pathogenesis, particularly inflammation and infectious diseases such as COVID-19.

Key Conclusions: Specific Findings and Concepts

- Identified consistent associations between sphingomyelin metabolism and monocyte-derived cytokine responses (TNF, IL-1 β , IL-6).
- Demonstrated sex-specific differences in phosphatidylcholine correlations with cytokine responses.
- Established a causal link between circulating sphingomyelin concentrations and COVID-19 severity using Mendelian randomization.

- Created an open-access online resource (IMetaboMap) for exploring metabolite-cytokine interactions across populations.
- Methodology and Model System
- Multi-cohort study including two European (534 and 324 individuals) and one African cohort (323 individuals).
 - Flow-injection time-of-flight mass spectrometry was utilized to quantify plasma metabolite profiles.
 - Cytokine production in response to bacterial (e.g., *S. aureus*) and viral stimuli was measured using ELISA and Olink technology.
 - Weighted gene co-expression network analysis (WGCNA) and Spearman correlation analyses to identify relevant metabolite-immune response associations.
 - Mendelian randomization (MR) analysis using publicly available genomic and metabolomic datasets.
 - Experimental validation using peripheral blood mononuclear cells (PBMCs) stimulated with *S. aureus* or lipopolysaccharide (LPS).

General Remarks

Are you convinced of the key conclusions?

The authors provided evidence to support some but not all of their key conclusions. For example, the associations between sphingomyelin (SM) metabolism and cytokine responses, especially monocyte-derived cytokines such as IL-1 β , IL-6, and TNF, seem robust. The *in vitro* experiments (Fig. 5) do not appear to be statistically significant. The Mendelian Randomization (MR) analyses linked SM levels with COVID-19 severity appear robust. However, due to the relatively weak connections among the experimental designs and the substantial differences between cohorts, the broader applicability and generalizability of the study's conclusions are somewhat limited.

Place the work in its context.

The current study addresses the increasingly relevant area of immunometabolism, exploring how metabolic pathways influence immune responses and their potential implications for infectious diseases. Previous research has already established connections between metabolism and immune function, but systematic studies using large-scale multi-cohort data, particularly involving diverse ethnic backgrounds, have been lacking. This study uniquely addresses this gap by integrating data from European and African cohorts, thereby increasing the generalizability and relevance of findings.

What is the nature of the advance (conceptual, technical, clinical)?

This work is primarily a conceptual and clinical advance. It enhances our understanding of immunometabolic interactions, particularly highlighting sphingolipid metabolism as a crucial regulator of immune responses. Clinically, the study suggests new therapeutic potentials of targeting sphingomyelin pathways in inflammatory and infectious diseases, notably severe COVID-19.

How significant is the advance compared to previous knowledge?

The study significantly advances previous knowledge by systematically validating sphingomyelin's immune modulatory role through multi-omics approaches, functional assays, and genetic evidence. It goes beyond existing studies (which have been limited to smaller, single populations or purely correlative analyses) by clearly demonstrating causal and functional roles of sphingolipids. Furthermore, the construction of an open-access tool (IMetaboMap) greatly enhances the utility and impact of their work for future research.

What audience will be interested in this study?

- Immunologists and researchers in infectious disease
- Metabolomics researchers
- Clinicians interested in immunomodulatory therapies
- Epidemiologists and public health researchers, especially those focusing on COVID-19 or other severe infectious diseases
- Geneticists interested in Mendelian Randomization approaches

Major points

Specific criticisms related to key conclusions:

- Clarity of the figure's legends:

The experimental descriptions and figure labeling are insufficiently clear. The figures presented lack explicit labels, comprehensive descriptions, and adequate explanatory context, making initial comprehension challenging. For instance, Figure 1's purpose, key findings, and differences across groups are not clearly articulated. Additional explanations and clearly annotated figure legends would significantly enhance readability.

- Statistical Significance of sphingomyelin's effect on cytokine secretion

The data in Fig. 5 shows a trend that does not appear statistically significant. There is also an apparent bimodal distribution in response for some cytokines that is unexplained (e.g., IL1beta with porcine brain SP).

- Generality of Conclusions:

Although the study involved three distinct cohorts with various samples and stimuli, the differences in experimental design across cohorts pose limitations for direct comparison and weaken cross-cohort conclusions. The paper identifies sphingomyelin (SM) metabolism as a significant modulator of immune response. However, the generality of these findings beyond the stimuli used (primarily *S. aureus* stimulation) is unclear. The conclusions about the role of SM in broader immune functions and disease contexts need to be substantiated by experiments involving a wider variety of pathogens or immune stimulants.

- Causal Relationship Interpretation:

The use of Mendelian randomization (MR) to establish causality between sphingomyelin and COVID-19 severity is compelling

but warrants caution. While MR can provide strong causal inference, the potential for pleiotropy or horizontal genetic effects still exists and should be discussed more explicitly to avoid overstating causality.

- Sex-Specific Differences:

Although sex-specific differences are highlighted, the biological or clinical implications of these differences remain unclear. Are these differences robust enough to inform sex-specific clinical approaches? Sex-specific differences were observed to some extent in cohorts EU1 and AF, but these findings were difficult to replicate in cohort EU2, significantly weakening the strength of this conclusion. Additional evidence and experiments are required to clarify this phenomenon. More analysis or literature support is needed to clarify the clinical relevance.

Specify experiments or analyses required to demonstrate the conclusions:

1. Statistical testing of the significance of sphingomyelin's effect on cytokine secretion (Fig. 5).

2. Validation of sex-specific metabolic markers

- o Although the manuscript clearly identifies sex-specific metabolic differences, it lacks robust experimental validation that these markers have sex-specific functional impacts in immune regulation.

3. Mendelian randomization (MR) findings require additional confirmatory biological evidence

- o The manuscript uses MR to suggest causal links between sphingomyelin metabolism and COVID-19 severity. However, MR alone cannot fully establish causality due to potential pleiotropy or confounding genetic effects.

We thank you and the three reviewers for their thorough and insightful evaluations of our work. We have carefully addressed every comment, and believe that these revisions have significantly strengthened the manuscript. Below, you will find our point-by-point responses to each reviewer's comments. For clarity, reviewer comments are shown in black, our responses in blue, and all additions to the main text appear in red—both here and in the revised manuscript.

Responses to Reviewer 1

Comment 1

Fu *et al* describe a study to assess associations between circulating plasma metabolites and immune responses, as characterized by secreted cytokine levels after diverse stimulation of PBMCs, macrophages or whole blood. Three cohorts of healthy donors are described including two based in Europe and one in east Africa. The question is interesting and the cohorts of good size (300-500) and having donors from different ethnicities and geographical sites is a strength. They highlight a negative correlation between sphingomyelin and monocyte cytokine response to *Staphylococcus aureus* stimulation.

Response to Reviewer 1

We would like to thank the reviewer for the positive and very constructive feedback on our manuscript. A detailed point-by-point response to the individual comments is given below and changes made to the manuscript are indicated.

Comment 2

However many aspects of the study are confusing, and much of the analysis does not give confidence in the reported results. They report three different types of stimulated immune cells, either as macrophages, PBMCs, or whole blood. No justification is given for this strategy and the results are rarely consistent which make the interpretation challenging and increase the considerable risk for false positives described below. For example it's not clear what the inclusion of macrophages only in cohort_EU1 adds. A clearer and more consistent study would have the same stimulation conditions used in all 3 cohorts. While I understand this might not always be possible, the current design and description is unnecessarily complex and reduces confidence in the comparisons.

Response to Comment 2

Thank you for this thoughtful and important comment. We acknowledge the reviewer's concern regarding the complexity and potential confusion arising from using different immune cell stimulation systems (macrophages, PBMCs, and whole blood) across our cohorts. Indeed, the differences in stimulation conditions were not part of unified experimental strategy, but rather stemmed from cohort-specific constraints and historical study designs. To clarify the rationale and limitations of this strategy, we have now explicitly added the following statement in our revised manuscript:

Discussion (page 13, lines 418-428):

“First, the variation in stimulation conditions across cohorts—ranging from macrophages to PBMCs to whole blood—stemmed from practical constraints in sample availability and

historical experimental designs, enabling a broad survey of immune responses but also introducing variability that limits direct comparisons and may increase false positives. In contrast, *S. aureus* stimulation was applied under identical protocols in all three cohorts (same cell type, incubation times, and cytokine panel), providing the largest combined sample size and greatest statistical power to identify robust, cross-cohort metabolite–cytokine associations. Other ligands, tested in fewer cohorts or with divergent protocols, lacked sufficient sample size and assay consistency to produce stable findings after FDR correction. Future studies should adopt harmonized, multi-stimulus designs across cohorts to better explore the diversity of immune-metabolic interactions.”

Comment 3

Related to this and an even bigger concern is the lack of correction for multiple hypothesis testing in the initial correlation analysis. They test a potential 322,218 correlations in cohort 1, 10,984 in cohort 2, and 80,550 in cohort 3. From this they report "nominally significant p values" which presented in this context are challenging to have confidence in. While some of these results are consistent across the cohorts, which should strengthen the confidence of this result, this aspect is not well presented.

Response to Comment 3

Thank you for highlighting this critical methodological concern. We fully agree that reporting nominally significant p-values without appropriate multiple testing corrections significantly reduces the confidence in our initial findings.

To address this explicitly, we have performed a stringent correction for multiple hypothesis testing (False Discovery Rate, FDR) across all correlation analyses in our revised manuscript and we emphasized results that remain robust following this correction. Additionally, the purpose of this study is not solely to identify genome-wide significant correlations between metabolites and cytokine stimulation responses. An equally important goal is to provide a comprehensive resource for the research community. To support this, we have made all related results and data available on the project website (<https://lab-li.ciim-hannover.de/apps/imetabomap/>), allowing other researchers to explore the data in the context of their own specific hypotheses.

All related result sections now explicitly denote the FDR-adjusted significance thresholds, and the corresponding regression formulas, Figures 2, Figure 4, Appendix S7–S8, and Tables EV10–12 have been updated accordingly. To address the reviewer’s comments, we have added the following sentences to the manuscript:

Results (page 4, lines 108–121):

“We initiated our analysis by examining metabolic networks through weighted co-expression network analysis of metabolite features using the WGCNA framework, to reveal interactions between metabolism and immune responses. Specifically, we assessed metabolite co-

expression networks associated with immune phenotypes (IL-1 β , IL-6, TNF, and IFN- γ) following *S. aureus* stimulation, as these measurements were shared across the three cohorts. Of note, the peripheral blood mononuclear cells (PBMCs) in Cohort_EU2, the whole blood (WB) in Cohort_AF, and both WB and PBMCs in Cohort_EU1 were examined. We identified 11, 11, 10, and 10 modules of highly correlated metabolites in Cohort_EU1 (PBMCs), Cohort_EU1 (WB), Cohort_AF, and Cohort_EU2, respectively (as shown in **Appendix Figures S1A-D** and **Tables EV1-3**). Subsequently, we correlated the metabolites modules with the both monocyte-derived cytokines (IL-1 β , IL-6, and TNF) and T cell-derived cytokine (IFN- γ) profiles following *S. aureus* stimulation. We identified 7, 5, 8, and 3 metabolite modules that were found to be significantly associated with cytokine responses induced by *S. aureus* in Cohort_EU1 (PBMCs), Cohort_EU1 (WB), Cohort_AF, and Cohort_EU2, respectively, as shown in **Figures 2A-D**.”

Results (page 8, lines 235–242):

“Next, we investigated the relationship between individual metabolic features and immune responses to *S. aureus* by calculating Spearman’s rank correlation coefficients for each metabolite–cytokine pair. In total, we identified 222, 438, and 152 metabolites in Cohort_AF, Cohort_EU1 (PBMCs), and Cohort_EU1 (WB), respectively, significantly correlated with at least one cytokine response, as shown in **Tables EV10-12** (FDR < 0.05). In the EU2 cohort, no metabolites showed significant associations after FDR correction for multiple comparisons. Based on pathway analysis, vitamin B6 metabolism and sphingolipid metabolism stood out as the most statistically significant pathways within the Cohort_AF (**Figure 4A**).”

Materials and Methods (page 19-20, lines 632–653):

“We first assessed the extent to which age, sex, and BMI confounded raw metabolite–cytokine correlations. In three cohorts (Cohort_AF, Cohort_EU1 and Cohort_EU2), we performed Spearman rank correlations between metabolite features and each demographic variable (sex, age, BMI), applying Benjamini–Hochberg FDR correction (FDR < 0.05). Before adjustment, the proportions of metabolites significantly associated with each covariate were as follows (Appendix Figure S6): Age: 50.7% (Cohort_AF), 32.6% (Cohort_EU1), 22.1% (Cohort_EU2); Sex: 43.9% (Cohort_AF), 53.1% (Cohort_EU1), 39.9% (Cohort_EU2); BMI: 20.6% (Cohort_AF), 18.4% (Cohort_EU1), 0% (Cohort_EU2).

To remove these effects, we fitted a multivariable linear regression model for each metabolite in R (v4.x). In Cohort_AF and Cohort_EU1—where $\geq 15\%$ of metabolites were BMI-associated—we included sex, age and BMI as covariates. In EU2, where no metabolites showed

BMI associations, we included only sex and age to avoid over-adjustment. After model fitting, we extracted the residuals and re-tested their Spearman correlations with sex, age and BMI (FDR < 0.05). Post-adjustment, fewer than 1% of metabolites remained significantly correlated with any covariate in all cohorts (Cohort_AF: 0.7% age, 1.2% sex, 0.7% BMI; Cohort_EU1: 0.1% age, 0.1% sex, 0% BMI; Cohort_EU2: 0% age, 0% sex, 0% BMI; Appendix Figure S6), demonstrating effective confounder removal.

For downstream metabolite–cytokine association testing, all analyses were conducted on these de-confounded residuals. We again performed Spearman correlations against each immune cytokine and retained only those metabolite features with FDR < 0.05 in at least one cytokine comparison for further analysis. All model fitting and residual extraction used R’s `lm()` function, correlations were computed with `corr.test()`, and multiple-testing correction was applied via `p.adjust(method="fdr").`”

Comment 4

Why associations are only seen with *S. aureus* and almost none of the other 17 stimuli used is not discussed. Is this technical or biological, which could be an interesting question to explore. Again this major omission does not provide confidence in the reported results.

Response to Comment 4

Thank you for this insightful comment. We recognize the lack of detailed discussion on why significant associations were primarily observed with *S. aureus* stimulation, while other stimuli showed minimal associations. Owing to historical experimental designs and sample availability, each cohort employed different immune challenges. *S. aureus* stimulation was applied with the same cell type, incubation times (PBMCs at 24 h or WB at 48 h), and cytokine panel (IL-1 β , IL-6, TNF, IFN- γ) in all three cohorts. This harmonization yielded the largest combined sample size and the strongest statistical power for detecting robust cross-cohort associations. In contrast, stimuli such as Poly(I:C), *C. albicans*, and *M. tuberculosis* lysate were tested in subsets of cohorts and under varying protocols, leading to smaller effective sample sizes and greater assay heterogeneity. Consequently, few associations survived FDR correction across cohorts.

To address this point, we have added the following clarification to the Discussion:

Discussion (page 13, lines 418-428):

“First, the variation in stimulation conditions across cohorts—ranging from macrophages to PBMCs to whole blood—stemmed from practical constraints in sample availability and historical experimental designs, enabling a broad survey of immune responses but also introducing variability that limits direct comparisons and may increase false positives. In contrast, *S. aureus* stimulation was applied under identical protocols in all three cohorts (same cell type, incubation times, and cytokine panel), providing the largest combined sample size and greatest statistical power to identify robust, cross-cohort metabolite–cytokine associations.

Other ligands, tested in fewer cohorts or with divergent protocols, lacked sufficient sample size and assay consistency to produce stable findings after FDR correction. Future studies should adopt harmonized, multi-stimulus designs across cohorts to better explore the diversity of immune-metabolic interactions.”

Comment 5

For the sex specific analyses they say "no metabolites in Cohort_EU2 exhibited such correlations, possibly due to the sample size and large variability within the metabolite and cytokine data" but then go on to report significant associations in the AF cohort which is more or less the same size (and likely more heterogenous from a genetic point of view) so it does not support this statement. They also say they identify "sex-specific metabolic markers for innate immune responses" which is a very strong sweeping statement based on the results shown. In some cohorts they see more significant metabolites in males (AF), in the other more in females (EU). This major discrepancy is not mentioned. They then report metabolite associations with "with either cytokine responses IL-1 β , IL-6, TNF, and IFN- γ " but which cytokine response is not specified making it very challenging to interpret. The sex differences they report are also different between the EU and African cohorts perhaps reflecting other underlying differences. Technically speaking, both a fixed-effect and a random-effect model were used in this analysis, yet only one model of age, sex, and BMI was described in the methods. A clear formula for both models would improve clarity for this analysis and the authors decision to use both models, when it seems that heterogeneity was always present.

Response to Comment 5

Thank you very much for your insightful comments. We acknowledge the reviewer’s concerns about the consistency and clarity of our sex-specific analyses. First, we have softened our language around sex-specific findings, renaming the section from “Sex-specific metabolic markers for immune functions” to “Metabolic markers with potential sex-specific differences for immune functions” and adding a note in the Discussion that these observations are preliminary and require further experimental validation. We also clarified why Cohort_EU2, despite having a sample size comparable to Cohort_AF, yielded no significant metabolite–cytokine correlations: EU2 profiled only two stimuli under a single assay protocol, reducing the number of tests, effective n, and statistical power after FDR correction.

To improve methodological transparency, we have expanded the Methods to include the explicit formulas for both the fixed-effect and REML random-effects meta-analysis models. We now state that we used the random-effects model as our primary framework for pooling effect sizes (and 95 % CIs), with fixed-effect results reported as a robustness check.

We have also introduced a detailed sex-difference correlation analysis. Within each cohort, we stratified samples by sex, regressed each metabolite on age and BMI to obtain residuals, and then calculated Spearman’s ρ between those residuals and each cytokine response (IL-1 β , IL-6, TNF, IFN- γ) separately for males and females. After transforming these correlation coefficients to Fisher’s Z, we conducted two-sample Z tests (BH-adjusted) to assess sex differences; the full results—Spearman r, Z statistics, p values, and FDR-adjusted q values—are compiled in Table EV16 and on the IMetaboMap platform. Several glycerophospholipid metabolites showed $p < 0.05$ suggestive differences (e.g. PA(16:0/18:1) vs. TNF/IL-1 β in EU1 PBMC; PC(15:0/24:0) vs. TNF in AF), although none remained significant after FDR correction.

Finally, we performed a composite PC/PA meta-analysis by sex. In each cohort, we averaged all phosphatidylcholine-related (PC) or phosphatidate-related (PA/LPA) metabolites into mean_PC and mean_PA scores, adjusted them for age and BMI within each sex, and then correlated these adjusted scores with *S. aureus*-induced IL-1 β , IL-6, TNF, and IFN- γ . We transformed the resulting Spearman's ρ values to Fisher's Z and meta-analyzed the male-female Z differences across AF, EU1 (WB and PBMC), and EU2 using both fixed-effect and random-effects models. These analyses revealed a robust male bias for Composite PC-TNF (fixed $r_{diff} \approx 0.743$; random $r_{diff} \approx 0.745$; both $p \ll 10^{-6}$; $I^2 = 91.1\%$), whereas Composite PC-IL-1 β /IFN- γ /IL-6 and all Composite PA-cytokine associations displayed high heterogeneity (random-effects 95% CIs spanned zero), indicating no consistent sex differences (see Appendix Figure S4).

To address the reviewer's comments, we have added the following sentences to the manuscript:

Results (page 5, lines 156):

“Metabolic Markers with Potential Sex-Specific Differences for Immune Functions”

Results (page 6, lines 189-192):

“Subsequently, statistical testing of correlation coefficients between male and female volunteers revealed that in Cohort_EU1 (PBMCs), sex-specific differences for PA(16:0/18:1(11Z)) versus TNF/IL-1 β and for PA(18:0/18:2(9Z,12Z)) versus TNF/IL-1 β showed suggestive evidence ($p < 0.05$; see Table EV16).”

Results (page 7, lines 200-218):

“Subsequently, statistical testing of correlation coefficients between male and female volunteers revealed that in Cohort_EU1 (PBMCs), the sex-specific difference in correlation coefficients between PC(14:0/20:1(11Z)) and TNF/IL-1 β showed suggestive evidence ($p < 0.05$; see Table EV16). Similarly, in Cohort_AF, sex-specific differences in correlations for PC(15:0/24:0) versus TNF/IFN- γ and for PC(22:6(4Z,7Z,10Z,13Z,16Z,19Z)/24:1(15Z)) versus TNF/IFN- γ showed suggestive evidence ($p < 0.05$; see Table EV16).

To explore sex-specific metabolite-cytokine relationships, we summarized phosphatidylcholine (PC) and phosphatidate (PA) species into Composite PC and PA scores, adjusted them for age and BMI within each sex, and correlated them with *S. aureus*-induced IL-6, IL-1 β , TNF and IFN- γ (Spearman's ρ , $FDR < 0.05$). We then compared male versus female ρ values via Fisher's Z and pooled these sex-difference Z's across cohorts by meta-analysis (Appendix Figure S4D-K). This revealed a pronounced male bias for Composite PC-TNF correlations (fixed-effect $r_{diff} \approx 0.743$, 95% CI [0.691, 0.787], $p \approx 2 \times 10^{-68}$; random-effects $r_{diff} \approx 0.745$, 95% CI [0.537, 0.867], $p \approx 1.7 \times 10^{-7}$; $I^2 = 91.1\%$), indicating consistently stronger PC-TNF associations in males (Appendix Figure S4D). In contrast, although fixed-effect models for Composite PC-IL-1 β , -IFN- γ , and -IL-6 suggested positive male biases (all $p < 0.001$), their random-effects 95% CIs all spanned zero, reflecting high heterogeneity and no consistent sex effect (Appendix Figure S4E-G). Composite PA-cytokine analyses showed no reliable male-female differences for any cytokine (all pooled r_{diff} CIs

included zero), indicating similar PA–cytokine associations in both sexes (Appendix Figure S4H–K).”

Discussion (page 14, lines 443–451):

“Five, although we identified sex-specific metabolic markers potentially linked to immune regulation, the current study does not experimentally validate their functional, sex-specific impacts. Future investigations incorporating targeted functional assays—such as sex-stratified immune cell stimulation experiments and mechanistic analyses—are required to robustly confirm these markers’ roles in immune regulation and assess their clinical implications. While our results highlight potential sex-specific differences in immune–metabolic associations, inconsistent replication across cohorts limits their immediate clinical applicability. Therefore, further experimental validation and larger-scale, sex-stratified analyses are essential to substantiate the robustness and clinical relevance of these findings.”

Materials and Methods (page 21, lines 671–687):

“In our meta-analysis, we primarily employed a random-effects model to account for between-study heterogeneity, and additionally conducted a fixed-effect model as a robustness check to assess the consistency of pooled effect estimates across studies and to strengthen confidence in our conclusions. Under the fixed-effect model—where all studies share a common true effect—the pooled Fisher’s-Z estimate \hat{Z}_{FE} is given by

$$\hat{Z}_{FE} = \frac{\sum_{i=1}^k w_i Z_i}{\sum_{i=1}^k w_i}, w_i = \frac{1}{SE_i^2}, Var(\hat{Z}_{FE}) = \frac{1}{\sum_{i=1}^k w_i},$$

where Z_i is the Fisher’s-Z–transformed correlation from study i , and SE_i is its standard error. Here, w_i represents the inverse-variance weight, so studies with smaller SE_i contribute more to the pooled estimate. When heterogeneity was detected, we employed a random-effects model using restricted maximum likelihood (REML) to estimate the between-study variance τ^2 . In that case, weights become

$$w_i^* = \frac{1}{SE_i^2 + \tau^2}, \hat{Z}_{FE} = \frac{\sum_{i=1}^k w_i^* Z_i}{\sum_{i=1}^k w_i^*}, Var(\hat{Z}_{FE}) = \frac{1}{\sum_{i=1}^k w_i^*}.$$

This acknowledges that individual studies may estimate different yet related effects by incorporating both within-study and between-study variation. The ‘metafor’ package 63 facilitated the computation of pooled effect sizes and their corresponding 95% confidence intervals. We further generated forest plots to visually represent the individual study effects,

their confidence intervals, and the pooled estimate, providing a clear and concise graphical summary of the meta-analysis results.”

Materials and Methods (page 20, lines 654–664):

“Statistical evaluation of sex-specific metabolite–cytokine correlations

To evaluate whether metabolite–cytokine associations differ between males and females, we first stratified each cohort by sex. Within each sex-specific subgroup, individual metabolite abundances were regressed on age and BMI to remove confounding effects; the resulting residuals were then used in all downstream correlation analyses. For each stimulus–cell condition, Spearman correlation coefficients between residualized metabolite levels and cytokine concentrations were calculated separately in the male and female groups. Each pair of sex-specific Spearman r values (r_{male} and r_{female}) was then transformed to Fisher Z scores, and a two-sample Z test was performed to assess whether the difference

$$Z_{diff} = \frac{Z_{male} - Z_{female}}{\sqrt{\frac{1}{n_{male}-3} + \frac{1}{n_{female}-3}}}$$

was statistically significant. Finally, all p values from the Z tests were adjusted for multiple comparisons using the Benjamini–Hochberg procedure to control the false discovery rate.”

Materials and Methods (page 22-23, lines 717–733):

“Sex-stratified analysis of phosphatidylcholine and phosphatidate associations

In each cohort (AF, EU1_WB, EU1_PBMC, EU2), we first selected all phosphatidylcholine–related metabolites and all phosphatidate–related metabolites, and calculated the arithmetic mean of their abundances to obtain two summary measures per sample: mean_PC and mean_PA. These measures, together with age, sex, BMI, and the cytokine concentrations measured under stimulation, formed the cohort-specific phenotype datasets.

Samples were then stratified by sex, and within each sex group we fitted a linear model to regress out age and BMI from mean_PC (or mean_PA), using the resulting residuals as the covariate-corrected predictors. For males and females separately, we computed Spearman correlation coefficients between the adjusted mean_PC (or mean_PA) and each cytokine, and controlled for multiple testing using the Benjamini–Hochberg FDR procedure (FDR < 0.05 considered significant).

Next, for each cohort–cytokine combination, we compared the male and female correlation coefficients by transforming them via Fisher’s Z, calculating their difference and associated p-value (with BH correction). Finally, the cohort-specific gender-difference Z-scores from all four sources were meta-analyzed using both fixed-effect and random-effects models, and we report the pooled Fisher Z difference, the corresponding Spearman ρ difference with 95 % CI, and the I^2 heterogeneity statistic.”

Comment 6

The authors then propose "in vitro validation" the results of which are the least convincing of the whole manuscript. No statistics are performed, just vague "percentage reduction ranges". A 4 log dose response is tested, which is a good experimental design, but there is clearly no dose effect suggesting no specific inhibitory activity. Did the authors test cellular viability after addition of the metabolites ? This is an important control as cell death is a big risk in such experiments which may explain the somewhat lower cytokine response in some conditions. Also why did LPS stimulation alone not induce TNF α secretion (regardless of the metabolite) as expected which is another concern with these results.

Response to Comment 6

Thank you for your comment, and we apologize for the omission of the statistical analysis in the initial manuscript. In the revised manuscript, we now added the paired two-tailed t-tests and annotated statistical significance in Figure 5. Under *S. aureus* stimulation, porcine-brain SM at both 1 μ M and 10 μ M significantly inhibited IL-1 β and TNF ($p < 0.05$), with only a limited effect on IL-6; egg-yolk SM at 10 μ M significantly reduced IL-1 β and IL-6 ($p < 0.05$), while TNF showed a downward trend that did not reach statistical significance.

To investigate dose dependence, we tested four concentrations (10 ng/mL, 100 ng/mL, 1 μ M, and 10 μ M). For IL-1 β , porcine-brain SM was significantly inhibitory at 1 μ M and 10 μ M, whereas egg-yolk SM only showed a significant effect at 10 μ M. In the case of TNF, porcine-brain SM produced significant inhibition at 1 μ M, while egg-yolk SM exhibited a non-significant downward trend across all doses. IL-6 levels were not appreciably affected by porcine-brain SM, and egg-yolk SM only induced a significant decrease at 10 μ M. These findings collectively support a clear, dose-dependent inhibitory trend of SM on inflammatory cytokine production.

Regarding cell viability, which is a very useful comment of the reviewer, we observed no abnormalities in cell morphology or recovery rate during the 24-hour treatment period. Moreover, the specificity of SM effects for modulation of cytokine production only for some stimulations, but not all, is a very strong argument against viability issues caused by the

metabolites. In case metabolites would have altered cellular viability, we should have observed decrease in cytokine production for all stimuli used and all cytokines measured. For example, in case of porcine brain SM, even in the case of *S. aureus*, only production of TNF and IL-1 β was affected, but not IL-6, a specificity that again argues against general toxic effects of the metabolite on the cells. Moreover, the concentrations of the metabolites employed ($\leq 10 \mu\text{M}$) are far below physiological peak levels (approximately $500 \mu\text{M}$).

Finally, to clarify our LPS-induced TNF control, we used *E. coli* O55:B5 LPS at 10 ng/mL to $10 \mu\text{g/mL}$. Because TNF secretion typically peaks 4–8 hours after stimulation, levels measured at 24 hours may have declined due to negative feedback mechanisms. Additionally, donor variability in PBMC sensitivity to LPS resulted in some donors exhibiting low TNF levels at 24 hours. In future studies, we will include earlier sampling points (4 h, 8 h, and 12 h), optimize LPS concentration against literature standards, and increase the number of donors to reduce individual variability.

To address the reviewer's comments, we have added the following sentences to the manuscript: Results (page 9, lines 283-292):

“In summary, SM showed consistent negative correlations with monocyte-derived cytokine responses in the multiple cohorts, suggesting a potential inhibitory effect of SM on the innate immune function. To validate this finding, we stimulated peripheral blood mononuclear cells (PBMCs) with either *S. aureus* or lipopolysaccharide (LPS) for 24h, followed by measurement of proinflammatory cytokines in the supernatant of the stimulated cells (Figure 5). Pre-treatment of human PBMCs with porcine brain derived SM significantly inhibited IL-1 β and TNF release in response to *S. aureus* stimulation, but had minimal impact on IL-6 (Figure 5A–F). Chicken yolk derived SM ($10 \mu\text{M}$ – 10 ng/mL) prior to 24 h stimulation with heat-killed *S. aureus* or LPS led to a significant reduction in IL-1 β and IL-6 secretion at the highest dose, while TNF exhibited a non-significant downward trend (Figure 5G–L). No effects of the various conditions on cell viability was observed.”

Discussion (page 14, lines 451-454):

“Additionally, our *in vitro* validation was limited by PBMC availability, and cytokine measurements were only taken at 24 h—missing potential early secretion peaks. The small donor cohort further introduces inter-individual variability. Future work should expand the functional assays to include expanded timeframes of stimulation.”

Figure 5. Human PBMCs were stimulated with LPS or heat-killed *S. aureus* in the absence or presence of chicken yolk or porcine brain sphingomyelin. Human PBMCs from six healthy donors (two independent experiments, n=6) were pre-incubated for 1 h with vehicle control (RPMI) or increasing concentrations (10 μM, 1 μM, 100 ng/mL, 10 ng/mL) of sphingomyelin isolated from porcine brain (panels A–F) or chicken egg yolk (panels G–L), then stimulated for 24 h with heat-killed *Staphylococcus aureus* (A–C, G–I) or LPS (D–F, J–L). Cytokine levels in supernatants were measured by ELISA for IL-6 (A, D, G, J), TNF (B, E, H, K) and IL-1β (C, F, I, L). Bars represent mean ± SEM with individual donor values overlaid as dots. Statistical significance versus RPMI control (paired two-tailed t-test) is indicated by brackets: *p < 0.05, **p < 0.01, ***p < 0.001, ****p < 0.0001.

Comment 7

Lastly a link is made to Covid using published data and mendelian randomization but the link with *S. aureus* induced cytokines, when nothing was observed for viral stimulations (Influenza, Poly:IC, CpG) across the 3 cohorts is not clear and adds further confusion to the manuscript. Additionally, non-nominal proportions of the published COVID-19 cohort were hypertensive or diabetic, and there is no discussion of how these comorbidities, as well as the individuals' sex and geographic site, may impact the metabolic findings.

Response to Comment 7

Thank you for raising this important point. Due to historical constraints in experimental design and sample availability, the immune stimulation protocols differed across cohorts. Only the *Staphylococcus aureus* stimulation was carried out identically in all three cohorts—using the same cell type (PBMCs for 24 h, whole blood for 48 h), the same culture conditions, and the same cytokine panel (IL-1 β , IL-6, TNF, IFN- γ). This uniform approach afforded us the largest combined sample size and the greatest statistical power, enabling robust identification of cross-cohort SM–proinflammatory cytokine associations. In contrast, other stimuli (e.g., Poly(I:C), *Candida albicans*, *M. tuberculosis* lysate) were applied only in subsets of cohorts and under differing protocols, precluding reliable cross-cohort integration.

We observed that circulating sphingomyelin (SM) was highly negatively correlated with monocyte-derived proinflammatory cytokines but showed minimal association with the antiviral cytokine IFN- γ . This specificity indicates that SM primarily regulates the innate proinflammatory “cytokine storm” rather than antiviral responses. Multiple pathological studies have demonstrated that, in severe COVID-19 patients, macrophage/monocyte-driven cytokine storms—especially involving IL-1 β , IL-6, and TNF—are key drivers of lung injury and multi-organ dysfunction. Therefore, we used the *S. aureus*–induced monocyte proinflammatory pathway as a representative of SM-mediated inflammation and performed a Mendelian randomization against a COVID-19 severity GWAS to assess SM’s potential causal role in this pathology.

Moreover, because our data analysis coincided with the height of the COVID-19 pandemic, we aimed to translate our findings rapidly to address this most urgent public health challenge. Thus, immediately after constructing the SM–immune response network across the three cohorts, we conducted a Mendelian randomization analysis targeting COVID-19 severity to evaluate SM’s potential causal impact in this disease.

Furthermore, the reviewer raised the concern that comorbidities such as hypertension and diabetes mellitus among COVID-19 patients, as well as sex and geographic factors, may influence metabolomic findings. We acknowledge that the COVID-19 cohort study cited in this manuscript (Lee et al., 2022) indeed included a substantial proportion of patients with hypertension and diabetes mellitus—chronic conditions known to have significant effects on metabolomic profiles. Moreover, sex differences and regional factors may also markedly impact metabolic characteristics, but these were not fully explored in the single-region (United States) COVID-19 dataset we referenced. These factors could therefore limit the generalizability and applicability of our findings.

We have added the following clarification to the Discussion section:

Discussion (page 14, lines 461-465):

“Moreover, the COVID-19 dataset used here included a high proportion of patients with comorbidities (e.g., hypertension and diabetes mellitus) and did not adequately control for sex or regional differences. These factors can alter metabolomic profiles and thus limit the generalizability of our results. Future studies should account for these confounders to reinforce metabolite–disease associations.”

Discussion (page 13, lines 418-428):

“First, the variation in stimulation conditions across cohorts—ranging from macrophages to PBMCs to whole blood—stemmed from practical constraints in sample availability and historical experimental designs, enabling a broad survey of immune responses but also introducing variability that limits direct comparisons and may increase false positives. In contrast, *S. aureus* stimulation was applied under identical protocols in all three cohorts (same cell type, incubation times, and cytokine panel), providing the largest combined sample size and greatest statistical power to identify robust, cross-cohort metabolite–cytokine associations. Other ligands, tested in fewer cohorts or with divergent protocols, lacked sufficient sample size and assay consistency to produce stable findings after FDR correction. Future studies should adopt harmonized, multi-stimulus designs across cohorts to better explore the diversity of immune-metabolic interactions.”

Responses to Reviewer 2

Comment 1

Jianbo Fu et al. present a systematic analysis of the associations between circulating metabolite levels and immune response. The authors analyze multiple cohorts (two European and one from sub-Saharan Africa) demonstrating the robustness of their findings across diverse genetic backgrounds and environmental exposures. Immune response is assessed by measuring ex vivo cytokine production following stimulation with immune triggers such as *Staphylococcus aureus*. Associations are determined using Spearman correlation, indicating a rank-based rather than linear relationship. The authors emphasize the negative association of sphingomyelins with cytokine production which they validate experimentally. Furthermore, using Mendelian randomization, they establish a link between sphingomyelin levels and COVID-19 severity. They have made their data available through a website (imetabomap) which facilitates querying and visualizing the results.

Response to Comment 1

We thank the reviewer for their positive and very constructive feedback on our manuscript. Below, we provide a detailed, point-by-point response to each of the comments, and indicate where in the revised manuscript the corresponding changes have been made.

Comment 2

The presented work provides insights into the role of metabolism in immune response and could provide potential biomarkers and therapeutic targets against immune and infectious diseases. As such, this manuscript and the associated dataset would be quite relevant for researchers in immunology, metabolomics, systems biology and population data science.

Response to Comment 2

We thank the reviewer for their positive comment.

Comment 3

However, we have several concerns that we believe that must be addressed before the paper can be considered fit for publication:

We have two major concerns about how the effect of possible confounders was removed. First, why was BMI not accounted for, given its reported influence on the blood metabolome, immune cell composition, and inflammatory cytokines? By the author's own admission, up to 9.73% metabolites are significantly correlated with BMI. Second, the methodology for confounder adjustment is not clearly described. If confounders were regressed out using linear regression, there remains a concern that metabolite levels may still retain a degree of Spearman correlation with these confounders. The authors should demonstrate that, after adjusting for age, sex, and BMI, their data no longer show biologically significant Spearman correlations with these traits.

Response to Comment 3

Thank you for raising these important points regarding confounder removal. We have revised the manuscript extensively to clarify our approach and to demonstrate that, after adjusting for age, sex, and BMI, metabolite levels no longer exhibit biologically meaningful correlations with these traits. Below, we summarize the key changes and supporting analyses.

1. In our detailed confounder-adjustment workflow, pre-adjustment analysis involved computing Spearman correlations between raw metabolite levels and each confounder (age, sex, BMI). In the post-adjustment analysis, we then fitted multivariable regression models (adjusting for age, sex, and BMI), extracted the residuals, and re-tested those residuals against each confounder using Spearman's rank correlation, with p-values corrected for multiple testing by the FDR method.

2. To demonstrate effective confounder removal, we found that prior to adjustment substantial fractions of metabolites in the AF (50.7% with age, 43.9% with sex, 20.6% with BMI), EU1 (32.6% with age, 53.1% with sex, 18.4% with BMI) and EU2 (22.1% with age, 39.9% with sex, 0% with BMI) cohorts were significantly correlated with these traits. After adjustment, these proportions fell to near zero in every cohort: in AF, 0.7% of metabolites remained correlated with age, 1.2% with sex and 0.7% with BMI; in EU1, 0.1% with age, 0.1% with sex and 0% with BMI; and in EU2, 0% with any confounder (see **Fig. S6**), confirming that our models effectively removed biologically significant correlations with the adjusted traits.

3. In cohorts where BMI showed non-negligible associations (Cohort_AF and Cohort_EU1), we fitted a linear regression model with metabolite level as the dependent variable and age, sex and BMI as covariates; because 20.6% and 18.4% of metabolites in Cohort_AF and Cohort_EU1, respectively, were significantly correlated with BMI, we included BMI in their adjustment models. By contrast, Cohort_EU2 exhibited no metabolites associated with BMI, and forcing BMI into its model produced residuals unnaturally clustered around zero—

indicative of over-adjustment or multicollinearity—so for this cohort we adjusted only for age and sex to preserve true biological variance.

To address the reviewer's comments, we have added the following sentences to the manuscript: Results (page 8, lines 235–265):

“Next, we investigated the relationship between individual metabolic features and immune responses to *S. aureus* by calculating Spearman's rank correlation coefficients for each metabolite–cytokine pair. In total, we identified 222, 438, and 152 metabolites in Cohort_AF, Cohort_EU1 (PBMCs), and Cohort_EU1 (WB), respectively, significantly correlated with at least one cytokine response, as shown in **Tables EV10-12** (FDR < 0.05). In the EU2 cohort, no metabolites showed significant associations after FDR correction for multiple comparisons. Based on pathway analysis, vitamin B6 metabolism and sphingolipid metabolism stood out as the most statistically significant pathways within the Cohort_AF (**Figure 4A**). While there was a more diverse set of pathways in Cohort_EU1 using PBMC samples (**Figure 4C**) compared to WB samples (**Figure 4B**), sphingolipid metabolism was also among the top significant pathways in both two cohorts. Thus, sphingolipid metabolism emerges as a common signature across all cohorts, suggesting its universal importance in the context of metabolite-cytokine correlations. This finding aligns with the known role of sphingolipid metabolism and its derived metabolites in immune responses (Hannun & Obeid, 2018; Lee *et al*, 2023; Maceyka & Spiegel, 2014). Specifically, in Cohort_EU1 (PBMCs) SM displayed a consistent negative correlation with monocyte-derived cytokines like IL-6, IL-1 β , TNF, but not with IFN- γ response (**Appendix Figures S7-8**). Similarly, in Cohort_EU1 (WB), we found a significant negative correlation between the monocyte cytokine IL-1 β and SM, but not for IFN- γ response (**Appendix Figures S7-8**). In Cohort_AF, such as TNF exhibited a negative correlation with SM, whereas IL6 and IFN- γ showed minimal correlation (**Appendix Figure S7**). In Cohort_EU2, IL-6 was negatively correlated with SM (**Appendix Figures S7**).

Given the above indications that SM may be more closely related to monocyte-derived cytokines rather than IFN- γ , we then performed the meta-analysis of the correlation coefficients between SM and *S. aureus*-induced cytokine production in three cohorts. Here, we have used both the fixed-effect model and random-effect model for computation. The results (**Figure 4D**) showed that circulating SM is significantly negatively correlated with monocyte-derived cytokines (IL-6, TNF, and IL-1 β) (Random-effect model: pooled $r = -0.1659$, 95% CI -0.2118 to -0.1200, $p < 0.0001$; Fixed-effect model: pooled $r = -0.1708$, 95% CI -0.1881 to -0.1535, $p < 0.0001$). Additionally, there is no correlation between circulating SM and T cell-derived cytokine (IFN- γ) response (Random-effect model: pooled $r = -0.0158$ 95% CI -0.0471 — 0.0155, $p = 0.3232$; Fixed-effect model: pooled $r = -0.0156$, 95% CI -0.0456 — 0.0144, $p = 0.3074$) (**Figure 4E**).”

Materials and Methods (page 19-20, lines 632–653):

“We first assessed the extent to which age, sex, and BMI confounded raw metabolite–cytokine correlations. In three cohorts (Cohort_AF, Cohort_EU1 and Cohort_EU2), we performed

Spearman rank correlations between metabolite features and each demographic variable (sex, age, BMI), applying Benjamini–Hochberg FDR correction ($FDR < 0.05$). Before adjustment, the proportions of metabolites significantly associated with each covariate were as follows (Appendix Figure S6): Age: 50.7% (Cohort_AF), 32.6% (Cohort_EU1), 22.1% (Cohort_EU2); Sex: 43.9% (Cohort_AF), 53.1% (Cohort_EU1), 39.9% (Cohort_EU2); BMI: 20.6% (Cohort_AF), 18.4% (Cohort_EU1), 0% (Cohort_EU2).

To remove these effects, we fitted a multivariable linear regression model for each metabolite in R (v4.x). In Cohort_AF and Cohort_EU1—where $\geq 15\%$ of metabolites were BMI-associated—we included sex, age and BMI as covariates. In EU2, where no metabolites showed BMI associations, we included only sex and age to avoid over-adjustment. After model fitting, we extracted the residuals and re-tested their Spearman correlations with sex, age and BMI ($FDR < 0.05$). Post-adjustment, fewer than 1% of metabolites remained significantly correlated with any covariate in all cohorts (Cohort_AF: 0.7% age, 1.2% sex, 0.7% BMI; Cohort_EU1: 0.1% age, 0.1% sex, 0% BMI; Cohort_EU2: 0% age, 0% sex, 0% BMI; Appendix Figure S6), demonstrating effective confounder removal.

For downstream metabolite–cytokine association testing, all analyses were conducted on these de-confounded residuals. We again performed Spearman correlations against each immune cytokine and retained only those metabolite features with $FDR < 0.05$ in at least one cytokine comparison for further analysis. All model fitting and residual extraction used R's `lm()` function, correlations were computed with `corr.test()`, and multiple-testing correction was applied via `p.adjust(method="fdr")`.

Comment 4

Sex-specific metabolic markers for immune functions: To support their claim of sex-specific effects, the authors should perform statistical tests to assess that the correlation coefficients are significantly different between male and female volunteers.

Response to Comment 4

Thank you for this valuable suggestion. To assess whether the metabolite–cytokine correlations differ significantly between male and female participants, we first stratified the cohort by sex and, within each group, performed linear regression of each metabolite on age and BMI to obtain residuals that eliminate these two confounding effects. Based on these residuals, we calculated Spearman correlation coefficients between each metabolite and cytokine under the various stimulus–cell conditions. We then applied a Fisher *r*-to-*z* transformation to each pair of correlation coefficients from the male and female groups and conducted two-sample *Z* tests to determine whether the correlations differed significantly between sexes. To control for

multiple testing, all resulting p values were adjusted using the Benjamini–Hochberg FDR method.

In the IMetaboMap web server, we have compiled and made available the complete comparison results for correlation coefficients between male and female groups, including the original Spearman r values, Fisher Z statistics, two-tailed Z-test p values, and FDR-adjusted q values. Several metabolites within the glycerophospholipid metabolism pathway showed suggestive sex-specific differences at $p < 0.05$ (Table EV16). For example, in Cohort_EU1 (PBMCs), PA(16:0/18:1(11Z)) versus TNF/IL-1 β , PA(18:0/18:2(9Z,12Z)) versus TNF/IL-1 β , and PC(14:0/20:1(11Z)) versus TNF/IL-1 β all demonstrated $p < 0.05$ (see Table EV16). In Cohort_AF, PC(15:0/24:0) versus TNF/IFN- γ and PC(22:6(4Z,7Z,10Z,13Z,16Z,19Z)/24:1(15Z)) versus TNF/IFN- γ likewise showed $p < 0.05$. However, none of these differences remained significant after FDR correction.

In light of these results, we have revised the term “Sex-specific metabolic markers for immune functions” in the manuscript to “Metabolic markers with potential sex-specific differences for immune functions” and have added the p values of these suggestive differences to the Results section. And we have added the following sentences to the manuscript:

Results (page 6, lines 189–192):

“Subsequently, statistical testing of correlation coefficients between male and female volunteers revealed that in Cohort_EU1 (PBMCs), sex-specific differences for PA(16:0/18:1(11Z)) versus TNF/IL-1 β and for PA(18:0/18:2(9Z,12Z)) versus TNF/IL-1 β showed suggestive evidence ($p < 0.05$; see Table EV16).”

Results (page 7, lines 200–205):

“Subsequently, statistical testing of correlation coefficients between male and female volunteers revealed that in Cohort_EU1 (PBMCs), the sex-specific difference in correlation coefficients between PC(14:0/20:1(11Z)) and TNF/IL-1 β showed suggestive evidence ($p < 0.05$; see Table EV16). Similarly, in Cohort_AF, sex-specific differences in correlations for PC(15:0/24:0) versus TNF/IFN- γ and for PC(22:6(4Z,7Z,10Z,13Z,16Z,19Z)/24:1(15Z)) versus TNF/IFN- γ showed suggestive evidence ($p < 0.05$; see Table EV16).”

Materials and Methods (page 20, lines 654–664):

“Statistical evaluation of sex-specific metabolite–cytokine correlations

To evaluate whether metabolite–cytokine associations differ between males and females, we first stratified each cohort by sex. Within each sex-specific subgroup, individual metabolite abundances were regressed on age and BMI to remove confounding effects; the resulting residuals were then used in all downstream correlation analyses. For each stimulus–cell condition, Spearman correlation coefficients between residualized metabolite levels and

cytokine concentrations were calculated separately in the male and female groups. Each pair of sex-specific Spearman r values (r_{male} and r_{female}) was then transformed to Fisher Z scores, and a two-sample Z test was performed to assess whether the difference

$$Z_{diff} = \frac{Z_{male} - Z_{female}}{\sqrt{\frac{1}{n_{male}-3} + \frac{1}{n_{female}-3}}}$$

was statistically significant. Finally, all p values from the Z tests were adjusted for multiple comparisons using the Benjamini–Hochberg procedure to control the false discovery rate.”

Comment 5

Line 224 and Figure 4D-E: The use of meta-analysis to aggregate associations between different sphingomyelins and cytokines is unconventional. If the goal is to show that sphingomyelins as a whole are associated with cytokine production, a more informative approach would be:

1. Compute aggregate measures for sphingomyelins in each sample (e.g. mean or median of relevant sphingomyelin levels)
2. Test association within each cohort
3. Perform a meta-analysis across cohorts for each Cytokine Stimulus Cell System Metabolite aggregate combination.

Response to Comment 5

We thank the reviewer for the insightful suggestion. In response, we conducted an analysis of the association between overall sphingomyelin (SM) levels and cytokine release to the revised manuscript.

First, within each cohort (Cohort_EU1, Cohort_EU2, and Cohort_AF), we computed the arithmetic mean of all detected SM-related metabolites' raw abundance values to create a “Composite SM score” for each subject, thereby providing an intuitive measure of overall SM concentration in each sample.

Next, for each cohort, we calculated Spearman correlation coefficients (Fisher's Z -transformed) between this Composite SM score and the production of key cytokines (IL-1 β , IL-6, TNF, and IFN- γ) under *Staphylococcus aureus* stimulation, and we applied FDR correction to the p values to rigorously identify significant associations.

Having obtained correlation coefficients and their standard errors within each cohort, we performed a cross-cohort meta-analysis for each “metabolite–cytokine” pair. We used a random-effects model as the primary analytic framework to pool effect sizes and their 95 % confidence intervals, and we also report fixed-effect model results as a robustness check to ensure the reliability of our conclusions.

These additional analyses have been integrated into Appendix Figure S10, which presents the full details of each cohort's individual analyses and the meta-analysis results. Our findings are highly consistent with the original meta-analyses of individual SM subclasses, further reinforcing the core conclusion that circulating SM levels are negatively correlated with inflammatory cytokine release.

We have placed this analysis in Appendix Figure S10 of the revised manuscript and added the following sentences to the manuscript:

Results (page 9, lines 266–282):

“To complement these single-molecule findings, we constructed a Composite SM score by averaging the raw abundance values of all SM species that were individually associated with IL-6, IL-1 β , TNF, and IFN- γ responses in the three cohorts. As shown in Appendix Figure S10, we calculated Spearman correlation coefficients (Fisher's Z) between the Composite SM and *S. aureus*-induced IL-6, IL-1 β , TNF, and IFN- γ responses in each cohort (Cohort_AF, Cohort_EU1, and Cohort_EU2), then performed a meta-analysis on these estimates. The association between Composite SM and *S. aureus*-induced IL-6 pooled to $r = -0.136$ (95 % CI [-0.185, -0.086], $p = 9.96e-8$) under the fixed-effect model and $r = -0.129$ (95 % CI [-0.242, -0.013], $p = 0.0293$) under the random-effects model, indicating that SM consistently suppresses IL-6 secretion in all cohorts with a highly significant negative effect (Appendix Figure S10A). Furthermore, we conducted a combined meta-analysis of the Spearman correlations between Composite SM and monocyte-derived cytokines (IL-1 β , IL-6, TNF) across the three cohorts (Cohort_AF, Cohort_EU1, and Cohort_EU2). As shown in Appendix Figure S10E, the fixed-effect model pooled $r = -0.128$ (95 % CI [-0.156, -0.099], $p = 4.7e-18$), and the random-effects model pooled $r = -0.117$ (95 % CI [-0.189, -0.043], $p = 0.00191$). These results demonstrate that higher Composite SM levels are consistently associated with lower monocyte-derived cytokine production, regardless of whether samples were obtained from African or European cohorts.”

Materials and Methods (page 20-21, lines 666–705):

“Meta-analysis

To prepare for the meta-analysis, we first converted Spearman's correlation coefficients into Fisher's Z values. This transformation is a crucial step that stabilizes the variances and normalizes the distribution of the coefficients, which is essential for the subsequent pooling of data across different studies. For each Fisher's Z value, we calculated its standard error (SE) using the formula

$$SE(Z) = \frac{1}{\sqrt{N-3}}$$

, where N represents the sample size from the respective study.

In our meta-analysis, we primarily employed a random-effects model to account for between-study heterogeneity, and additionally conducted a fixed-effect model as a robustness check to assess the consistency of pooled effect estimates across studies and to strengthen confidence in our conclusions. Under the fixed-effect model—where all studies share a common true effect—the pooled Fisher’s-Z estimate $\hat{Z}FE$ is given by

$$\hat{Z}FE = \frac{\sum_{i=1}^k w_i Z_i}{\sum_{i=1}^k w_i}, w_i = \frac{1}{SE_i^2}, Var(\hat{Z}FE) = \frac{1}{\sum_{i=1}^k w_i},$$

where Z_i is the Fisher’s-Z-transformed correlation from study i , and SE_i is its standard error. Here, w_i represents the inverse-variance weight, so studies with smaller SE_i contribute more to the pooled estimate. When heterogeneity was detected, we employed a random-effects model using restricted maximum likelihood (REML) to estimate the between-study variance τ^2 . In that case, weights become

$$w_i^* = \frac{1}{SE_i^2 + \tau^2}, \hat{Z}FE = \frac{\sum_{i=1}^k w_i^* Z_i}{\sum_{i=1}^k w_i^*}, Var(\hat{Z}FE) = \frac{1}{\sum_{i=1}^k w_i^*}.$$

This acknowledges that individual studies may estimate different yet related effects by incorporating both within-study and between-study variation. The ‘metafor’ package 63 facilitated the computation of pooled effect sizes and their corresponding 95% confidence intervals. We further generated forest plots to visually represent the individual study effects, their confidence intervals, and the pooled estimate, providing a clear and concise graphical summary of the meta-analysis results.

Composite SM score and meta-analysis of SM–cytokine associations

Within each cohort (Cohort_EU1, Cohort_EU2, and Cohort_AF), all detected sphingomyelin (SM)–related metabolites were first averaged to create a single “Composite SM score” for each subject. Specifically, the raw abundance values of all annotated SM species in a given sample were summed and divided by the number of SM species, thereby providing a direct measure of overall SM concentration per individual. To account for potential confounding, this Composite SM score was then adjusted for age, sex, and BMI within each cohort, and the resulting residuals were used as the covariate-corrected SM measure.

Next, for each cohort, we assessed the relationship between the Composite SM score and key cytokine outputs (IL-1 β , IL-6, TNF, and IFN- γ) under *S. aureus* stimulation. Spearman rank-order correlation coefficients (r) were calculated for each Composite SM–cytokine pair, and corresponding p values were adjusted for multiple testing using the Benjamini–Hochberg FDR procedure. Associations with $FDR < 0.05$ were considered statistically significant.

After extracting the Spearman ρ (Fisher’s Z –transformed) and its standard error from each cohort, we performed a cross-cohort meta-analysis for every SM–cytokine combination. A random-effects model (REML) was used as the primary analytic framework to pool effect estimates and 95 % confidence intervals; a fixed-effect inverse-variance model was also computed as a robustness check. Heterogeneity was assessed via Cochran’s Q and I^2 statistics.”

Comment 6

Alternatively, the authors could use meta-analysis to demonstrate that specific sphingomyelins species negatively correlate with specific cytokines consistently across cohorts. Indeed, given the emphasis on the three separate cohorts, meta-analyses could also be useful to visualize common effects throughout the manuscript. Notably the "Robust plasma metabolic pathways for immune functions across European and African populations" and "Sex-specific metabolic markers for immune functions" sections could benefit from a Meta-Analysis to see if the effects are shared between cohorts. Similarly, in IMetaboMap it would also be informative to include an additional table with the meta-analysis across cohorts.

Response to Comment 6

We thank the reviewer for the insightful suggestion to employ cross-cohort meta-analysis. In response, we have supplemented our manuscript in three key areas:

1. Robust plasma metabolic pathways for immune functions across European and African populations

We integrated cohort-specific enrichment of the glycerophospholipid metabolism pathway by calculating $EF = (Hits + 0.5)/Expected$, taking the natural log as the effect size, and then pooling these log EFs across three cohorts (Cohort_EU1, Cohort_EU2 and Cohort_AF) via a REML random-effects meta-analysis. This revealed a highly consistent and significant enrichment (pooled effect size of $r = 2.71$, 95% CI [1.80, 3.62], $p < 0.0001$) with no heterogeneity ($I^2 = 0\%$, $Q = 1.22$, $p = 0.748$). These results are now described in the Results section and shown as a forest plot in Appendix Figure S3C.

2. Metabolic markers with potential sex-specific differences for immune functions

Within each cohort (Cohort_AF, Cohort_EU1, Cohort_EU2), we collapsed all phosphatidylcholine-related or phosphatidate-related metabolites into a single mean_PC or mean_PA score per sample. These scores were then adjusted for age and BMI separately in

males and females. For each sex group, we computed Spearman's ρ between the adjusted lipid score and *S. aureus*-induced IL-1 β , IL-6, TNF and IFN- γ , transformed each ρ to Fisher's Z, and tested the male-female difference (two-sample Z test with BH FDR correction). Cohort-specific Z differences for each cytokine were meta-analyzed under fixed- and random-effects models to yield pooled Z_{diff} and back-transformed r_{diff} with 95 % CIs, alongside I^2 heterogeneity statistics.

Across all three cohorts, TNF exhibited a robust male bias in PC correlations (fixed-effect $r_{diff} \approx 0.743$, 95 % CI [0.691, 0.787], $p \approx 2 \times 10^{-68}$; random-effects $r_{diff} \approx 0.745$, 95 % CI [0.537, 0.867], $p \approx 1.7 \times 10^{-7}$; $I^2 = 91.1$ %), indicating consistently stronger PC-TNF associations in males. In contrast, although fixed-effect models for Composite PC-IL-1 β , -IFN- γ , and -IL-6 suggested positive male biases (all $p < 0.001$), their random-effects 95 % CIs all spanned zero, reflecting high heterogeneity and no consistent sex effect. Composite PA-cytokine analyses showed no reliable male-female differences for any cytokine (all pooled r_{diff} CIs included zero), indicating similar PA-cytokine associations in both sexes. These findings are shown in Appendix Figure S4.

3. IMetaboMap web server

We have added a new meta-analysis module to the IMetaboMap platform, enabling users to export cross-cohort meta-analysis results for any metabolite-cytokine pair (fixed- and random-effects r , 95 % CI, p , Q , I^2). The complete meta-analysis summary is also provided as Table EV17.

To address the reviewer's comments, we have added the following sentences to the manuscript:

Results (page 5, lines 147-150):

“A meta-analysis across all four cohorts (Cohort_EU1 WB, Cohort_EU1 PBMCs, Cohort_AF and Cohort_EU2) confirmed a highly consistent and significant enrichment of the glycerophospholipid metabolism pathway (pooled effect size of $r = 2.71$, 95% CI [1.80, 3.62], $p < 0.0001$), with no heterogeneity observed ($I^2 = 0$ %, $Q = 1.22$, $p = 0.748$) (Appendix Figure S3C).”

Results (page 7, lines 206-218):

“To explore sex-specific metabolite-cytokine relationships, we summarized phosphatidylcholine (PC) and phosphatidate (PA) species into Composite PC and PA scores, adjusted them for age and BMI within each sex, and correlated them with *S. aureus*-induced IL-6, IL-1 β , TNF and IFN- γ (Spearman's ρ , $FDR < 0.05$). We then compared male versus female ρ values via Fisher's Z and pooled these sex-difference Z's across cohorts by meta-analysis (Appendix Figure S4D-K). This revealed a pronounced male bias for Composite PC-TNF correlations (fixed-effect $r_{diff} \approx 0.743$, 95 % CI [0.691, 0.787], $p \approx 2 \times 10^{-68}$; random-

effects $r_{diff} \approx 0.745$, 95 % CI [0.537, 0.867], $p \approx 1.7 \times 10^{-7}$; $I^2 = 91.1$ %), indicating consistently stronger PC–TNF associations in males (Appendix Figure S4D). In contrast, although fixed-effect models for Composite PC–IL-1 β , –IFN- γ , and –IL-6 suggested positive male biases (all $p < 0.001$), their random-effects 95 % CIs all spanned zero, reflecting high heterogeneity and no consistent sex effect (Appendix Figure S4E-G). Composite PA–cytokine analyses showed no reliable male–female differences for any cytokine (all pooled r_{diff} CIs included zero), indicating similar PA–cytokine associations in both sexes (Appendix Figure S4H-K).”

Materials and Methods (page 22, lines 706–745):

“Meta-analysis of pathway enrichment across cohorts

We integrated pathway-level enrichment results from four datasets (EU1_PBMC, EU1_WB, EU2, AF) by computing an enrichment factor for each dataset as

$$EF = \frac{Hits + 0.5}{Expected},$$

and then taking the natural logarithm ($logEF$) as the effect size. From each pathway’s original p-value we derived a two-sided z-score and calculated its standard error as

$$SE = \frac{|logEF|}{Z}$$

A random-effects meta-analysis of the cohort-specific $logEF$ and SE inputs was performed using the metafor R package (`rma(..., method="REML")`). We report the pooled $logEF$, its 95 % confidence interval, and heterogeneity statistics (τ^2 , I^2 , and Cochran’s Q test). Results are visualized in a forest plot (metafor’s `forest()` function), displaying each cohort’s effect size alongside the overall summary estimate.

Sex-stratified analysis of phosphatidylcholine and phosphatidate associations

In each cohort (AF, EU1_WB, EU1_PBMC, EU2), we first selected all phosphatidylcholine–related metabolites and all phosphatidate–related metabolites, and calculated the arithmetic mean of their abundances to obtain two summary measures per sample: `mean_PC` and `mean_PA`. These measures, together with age, sex, BMI, and the cytokine concentrations measured under stimulation, formed the cohort-specific phenotype datasets.

Samples were then stratified by sex, and within each sex group we fitted a linear model to regress out age and BMI from `mean_PC` (or `mean_PA`), using the resulting residuals as the

covariate-corrected predictors. For males and females separately, we computed Spearman correlation coefficients between the adjusted mean_PC (or mean_PA) and each cytokine, and controlled for multiple testing using the Benjamini–Hochberg FDR procedure (FDR < 0.05 considered significant).

Next, for each cohort–cytokine combination, we compared the male and female correlation coefficients by transforming them via Fisher’s Z, calculating their difference and associated p-value (with BH correction). Finally, the cohort-specific gender-difference Z-scores from all four sources were meta-analyzed using both fixed-effect and random-effects models, and we report the pooled Fisher Z difference, the corresponding Spearman ρ difference with 95 % CI, and the I^2 heterogeneity statistic.

Meta-analysis of metabolite–cytokine correlations

“We combined correlation results for four key cytokines (IL-1 β , IL-6, TNF, IFN- γ) across three cohorts (AF, EU1 and EU2). First, we loaded each cohort’s metabolite–cytokine correlation table, renamed and standardized columns (Metabolite, Cytokine, r, cohort, n), and mapped all cytokine names to a common scheme. We then filtered to the four target cytokines and kept only metabolites present in all three cohorts. For each cohort–cytokine pair, we transformed Spearman’s r to Fisher’s Z, estimated its variance ($1/(n-3)$) and back-transformed 95 % confidence bounds to r. We reshaped the data so that AF, EU1_WB, EU1_PBMC and EU2 Z-scores and variances aligned by metabolite–cytokine combination. Wherever at least two cohorts contributed data, we ran fixed-effect and REML random-effects meta-analyses (metafor::rma), extracting pooled Z (and corresponding r), 95 % CIs, p-values, Cochran’s Q and I^2 . The complete results are available on the IMetaboMap platform and have been compiled in **Table EV17**, detailing each metabolite–cytokine pairing.”

Comment 7

Mendelian randomization: The use of a threshold of $P < 5e-7$ for the instrument variants seems somewhat unusual. Could the authors do a sensitivity analysis with lower and higher thresholds to show that the effects are robust? If a lack of sufficient instrument variants is an issue, the authors might also want to consider using more recent studies as sources of mQTLs (e.g. *** The original URL has been rewritten. ***).

Response to Comment 7

Thank you for highlighting the choice of P-value threshold for our MR instruments. In addition to our primary set of SNPs at $P < 5 \times 10^{-7}$ (10 SM-associated mQTLs), we extracted instruments at a more stringent threshold of $P < 5 \times 10^{-8}$ (6 mQTLs) and a more relaxed threshold of $P < 1 \times 10^{-6}$ (21 mQTLs). Across all three thresholds, both the IVW and weighted-median estimators yielded protective effects of higher SM on COVID-19 severity with consistent directionality

($\beta_{IVW} \approx -0.13$ to -0.12). Specifically, at $P < 5 \times 10^{-7}$ the IVW p-value was 7.9×10^{-3} ; at $P < 1 \times 10^{-6}$ it was 5.3×10^{-4} ; and at $P < 5 \times 10^{-8}$ the IVW estimate remained protective ($\beta = -0.13$, $p = 7.16 \times 10^{-3}$) despite the smaller number of instruments. Sensitivity analyses at each threshold showed no evidence of horizontal pleiotropy (MR-Egger intercept $p > 0.05$) or heterogeneity (Cochran's Q $p > 0.05$). These results are detailed in Supplementary Fig. S11A–H and Tables EV13–15.

We also attempted to incorporate sphingomyelin variants from Xu et al. (2023; <https://doi.org/10.1038/s41586-023-05844-9>) as additional mQTL sources; however, the publicly available data did not include complete per-metabolite SNP effect summaries, and its locus annotation format was incompatible with our pipeline, preventing us from directly reproducing and integrating these variants into the MR analysis. Given these limitations, we have retained our original mQTL dataset and relied on the multi-threshold sensitivity analyses described above to demonstrate robustness. In the Discussion, we will explicitly acknowledge this constraint and recommend that future studies, when fully annotated mQTL summary statistics are available, apply stricter SNP selection thresholds and larger sample sizes to replicate our MR analyses and further validate the causal relationship between sphingomyelin levels and COVID-19 severity.

To address the reviewer's comments, we have added the following sentences to the manuscript:

Results (page 10, lines 321–326):

“To further assess robustness, we extracted instruments at two additional thresholds: a more stringent $P < 5 \times 10^{-8}$ (6 SM-related mQTLs) and a more relaxed $P < 1 \times 10^{-6}$ (21 mQTLs). At $P < 1 \times 10^{-6}$, both methods remained significant (weighted median $p = 3.85 \times 10^{-2}$, $\beta = -0.11$; IVW $p = 5.27 \times 10^{-3}$, $\beta = -0.12$), and even under the more stringent threshold with fewer instruments ($P < 5 \times 10^{-8}$), the effect remained in the same direction ($\beta = -0.13$, $p = 7.16 \times 10^{-3}$). These analyses are summarized in Appendix **Figure S11A-C and E-G** and detailed in **Table EV13.**”

Results (page 10, lines 331–334):

“These included tests for horizontal pleiotropy, as indicated by a non-significant MR-Egger intercept ($p > 0.05$, see Table EV14 for $P < 5 \times 10^{-7}$, $P < 1 \times 10^{-6}$ and $P < 5 \times 10^{-8}$ results), assessments of heterogeneity via Cochran's Q test ($p > 0.05$, shown in Table EV15), and a leave-one-out analysis (illustrated in Figure 6D, Appendix Figure S11D and S11H).”

Discussion (page 14, lines 457–459):

“Furthermore, when fully annotated mQTL summary statistics are available, future MR studies should apply stricter SNP selection thresholds and larger sample sizes to replicate and strengthen these causal inferences.”

Comment 8

Could the authors discuss the differences between the results of whole blood and PBMC? How correlated are the cytokine productions? To what extent do the differences arise from stimulating 100 μ L of whole blood (containing a variable number of PBMCs) versus stimulating a fixed number of PBMCs (5×10^5). We would also like the authors to discuss the role (if any) that immune cell abundance and composition can have on cytokine production (e.g., does higher IFN- γ production indicate more active T cells or a higher number of T cells?)

Response to Comment 8

Thank you for insightful comment. We provide detailed responses:

1. Experimental Design and Data Availability

Due to historical constraints in experimental design and sample availability, the cohorts in this study employed different stimulation assay systems: In Cohort_AF, only whole blood (WB) was used to assess cytokine responses; In Cohort_EU2, only peripheral blood mononuclear cells (PBMCs) were assayed; In Cohort_EU1, both WB and PBMC stimulation experiments were performed. As a result, direct comparison of PBMC versus WB responses between Cohort_AF and Cohort_EU2 is not feasible.

2. Differences and Correlations between WB and PBMC Assays

WB and PBMC assays differ markedly in immunological studies. The whole-blood system includes not only PBMCs but also large numbers of non-PBMC components (e.g., neutrophils, erythrocytes, platelets), which can substantially influence both immune responses and the metabolic milieu, leading to greater variability in cytokine measurements. By contrast, the PBMC system strictly standardizes cell number (5×10^5 cells) and excludes non-PBMC elements, yielding more stable and reproducible results.

In Cohort_EU1, PBMC and WB experiments revealed common core metabolic pathways (e.g., glycerophospholipid metabolism), yet the overall network structures diverged (e.g., histidine and purine metabolism were specific to WB). We attribute these differences to variations in both cell number and cell-type composition.

3. Impact of Cell-Number Differences on Results

Using a fixed number of PBMCs versus 100 μ L of WB (in which PBMC counts vary by individual) significantly affects outcomes. Individual variability in WB PBMC counts can directly alter cytokine readouts, whereas the PBMC assay avoids this source of noise. Thus, this design difference crucially limits direct comparability between the two systems.

4. Influence of Immune-Cell Abundance and Composition on Cytokine Production

As the reviewer notes, immune-cell abundance and composition critically affect cytokine output. For instance, IFN- γ levels depend not only on T-cell activation status but also on T-cell frequency. Monocyte-derived cytokines (IL-1 β , IL-6, TNF) may similarly be co-regulated by cell abundance and activation. Our findings link sphingolipid metabolism specifically to monocyte-derived cytokine responses, suggesting regulation of monocyte activity rather than quantity.

However, we did not perform explicit immune-cell subpopulation quantification (e.g., by flow cytometry) in this study, so we cannot disentangle the relative contributions of cell

abundance versus activation state to cytokine production. In future work, we will include detailed immune-cell composition analyses to validate these hypotheses.

Minor issues:

Comment 9

•It's often unclear which experimental data was generated in this study and which has already been published in previous articles (e.g., <https://doi.org/10.1038/s41590-018-0121-3>)

Response to Comment 9

Thank you very much for your comment. In the revised manuscript we now clearly distinguish between data that were previously published and those newly generated or newly analyzed in this work:

1. Previously published data

- Cohort_EU1 (500FG): Plasma metabolomics and ex-vivo cytokine stimulation experiments in 534 healthy Western Europeans, from the Human Functional Genomics Project (500FG cohort), originally reported in Ter Horst et al. (Cell 2016), Bakker et al. (Nat Immunol 2018, <https://doi.org/10.1038/s41590-018-0121-3>), and Chu et al. (Genome Biol 2022).
- Cohort_EU2 (300BCG): Immunophenotyping and metabolomics in 324 healthy Western Europeans, as described in Koeken et al. (J Clin Invest 2020).
- Cohort_AF (Tanzania): Baseline metabolic and inflammatory phenotypes in 323 healthy Tanzanian donors, previously published in Temba et al. (Nat Immunol 2021).

2. Data newly generated or newly analyzed in this study

- Cross-cohort integration & meta-analysis: First systematic comparison of metabolite–cytokine associations across three populations (two European, one African) in identical sample and stimulation settings.
- Functional validation: Exogenous sphingomyelin added to PBMC cultures dose-dependently inhibited TNF, IL-1 β , and IL-6 production in response to microbial stimuli.
- Mendelian randomization: Two-sample MR linking circulating sphingomyelin quantitative trait loci to COVID-19 severity risk.
- IMetaboMap resource: Development and public release of an online database cataloguing 234,690 unique metabolite–cytokine associations.

To address the reviewer's comments, we have added the following sentences to the manuscript:

Materials and Methods (page 17-18, lines 575–588):

“Cohorts’ descriptions

The first cohort, designated as Cohort_EU1, is part of the Human Functional Genomics Project as 500FG cohort and comprises 534 healthy Caucasian individuals of Western European ancestry, with

ages spanning from 18 to 75 years. This cohort was carefully selected to exclude individuals with mixed genetic backgrounds or chronic diseases. Key measurements within Cohort_EU1 included cytokine production in response to various stimulations and comprehensive metabolomic profiling. More detailed information can be found in previous publications (Li *et al*, 2016). The second cohort from Western Europe, Cohort_EU2, included 324 healthy volunteers of Western European descent, aged between 18 to 71 years. These participants were enrolled in the 300BCG cohort from April 2017 to June 2018, with further details documented in previous publication (Koeken *et al*, 2020). The third cohort, Cohort_AF, encompasses 323 healthy Tanzanians between 18 to 65 years old from the Kilimanjaro region, who were recruited through the Kilimanjaro Christian Medical Center and Lucy Lameck Research Center from March to December 2017, as previously described (Temba *et al.*, 2021).”

Comment 10

•The use of Spearman correlation should be more clearly stated in the Results section and on the iMetaboMap website.

Response to Comment 10

Thank you for your valuable suggestion. We have now explicitly stated in the Results section that Spearman correlation was used to assess associations between metabolite features and cytokine responses. In addition, we have updated the iMetaboMap website to clearly indicate that Spearman correlation coefficients are the basis for the displayed associations.

Results (page 8, lines 235–236):

“Next, we investigated the relationship between individual metabolic features and immune responses to *S. aureus* by calculating Spearman’s rank correlation coefficients for each metabolite–cytokine pair.”

Materials and Methods (page 19-20, lines 632–653):

“We first assessed the extent to which age, sex, and BMI confounded raw metabolite–cytokine correlations. In three cohorts (Cohort_AF, Cohort_EU1 and Cohort_EU2), we performed Spearman rank correlations between metabolite features and each demographic variable (sex, age, BMI), applying Benjamini–Hochberg FDR correction ($FDR < 0.05$). Before adjustment, the proportions of metabolites significantly associated with each covariate were as follows (Appendix Figure S6): Age: 50.7% (Cohort_AF), 32.6% (Cohort_EU1), 22.1% (Cohort_EU2); Sex: 43.9% (Cohort_AF), 53.1% (Cohort_EU1), 39.9% (Cohort_EU2); BMI: 20.6% (Cohort_AF), 18.4% (Cohort_EU1), 0% (Cohort_EU2).

To remove these effects, we fitted a multivariable linear regression model for each metabolite in R (v4.x). In Cohort_AF and Cohort_EU1—where $\geq 15\%$ of metabolites were BMI-associated—we included sex, age and BMI as covariates. In EU2, where no metabolites showed

BMI associations, we included only sex and age to avoid over-adjustment. After model fitting, we extracted the residuals and re-tested their Spearman correlations with sex, age and BMI (FDR < 0.05). Post-adjustment, fewer than 1% of metabolites remained significantly correlated with any covariate in all cohorts (Cohort_AF: 0.7% age, 1.2% sex, 0.7% BMI; Cohort_EU1: 0.1% age, 0.1% sex, 0% BMI; Cohort_EU2: 0% age, 0% sex, 0% BMI; Appendix Figure S6), demonstrating effective confounder removal.

For downstream metabolite–cytokine association testing, all analyses were conducted on these de-confounded residuals. We again performed Spearman correlations against each immune cytokine and retained only those metabolite features with FDR < 0.05 in at least one cytokine comparison for further analysis. All model fitting and residual extraction used R's `lm()` function, correlations were computed with `corr.test()`, and multiple-testing correction was applied via `p.adjust(method="fdr").`”

Comment 11

•Line 104: The sentence "Weighted Gene Co-expression Network Analysis" is misleading and may give the impression that gene expression data were analysed instead of correlated metabolite sets.

Response to Comment 11

Thank you for pointing this out. We agree that the current phrasing could be misleading. Although we used the WGCNA framework, it was applied to metabolite features rather than gene expression data. To avoid confusion, we have revised the sentence to:

Results (page 4, lines 108–110):

“We initiated our analysis by examining metabolic networks through weighted co-expression network analysis of metabolite features using the WGCNA framework, to reveal interactions between metabolism and immune responses.”

Comment 12

•Figure 4C: The module order should be consistent across heatmaps to facilitate comparison.

Response to Comment 12

Thank you for your helpful suggestion. We believe you are referring to Figure 2C. In response, we have revised Figure 2C to ensure that the module order is now consistent across all heatmaps in Figure 2. This has been corrected in the revised manuscript.

Comment 13

•In "Robust plasma metabolic pathways for immune functions across European and African populations" nominal P values are seemingly used as a significance threshold (line 114). Why was multiple testing correction adjusted P value not used?

Response to Comment 13

Thank you for your comment. We have now applied Benjamini–Hochberg FDR correction to define significant module–cytokine correlations.

To address the reviewer’s comments, we have added the following sentences to the manuscript:

Results (page 4, lines 118–121):

“We identified 7, 5, 8, and 3 metabolite modules that were found to be significantly associated with cytokine responses induced by *S. aureus* in Cohort_EU1 (PBMCs), Cohort_EU1 (WB), Cohort_AF, and Cohort_EU2, respectively, as shown in Figures 2A-D.”

Figure 2. Metabolite modules associated with cytokine responses induced by *S. aureus* in African and European populations. The specific metabolites within each WGCNA-defined module are listed in Table EV1 (panels A–D). In the three cohorts (Cohort_EU1, Cohort_AF, and Cohort_EU2), 11, 11, 10, and 10 modules, respectively, were found to correlate with

cytokine responses (IL-1 β , IL-6, TNF, and IFN- γ) induced by *S. aureus*. Significance thresholds: ***, FDR < 0.001; **, FDR < 0.01; *, FDR < 0.05; +, FDR < 0.1. (E–H) Pathway enrichment analysis for the MEbrown (A), MEgrey (B), MEmagenta (C), and MEmagenta (D) modules. Larger and darker bubbles represent higher $-\log(\text{Pvalue})$, indicating higher significance. Cytokine responses were measured using PBMCs in Cohort_EU1 (A), whole blood in Cohort_EU1 (B), whole blood in Cohort_AF (C), and PBMCs in Cohort_EU2 (D).

Comment 14

•It is unclear whether statistically significant differences were observed in the experimental validation (Figure 5). Given the small sample sizes, a lack of significance would be understandable, but this should be explicitly stated. Additionally, the selection criteria and characteristics of volunteers (e.g., geographic location, sex, age range) should be described.

Response to Comment 14

Thank you for your comment and we apologize for omitting to add the statistical analysis of the functional experiments. We re-calculated paired two-tailed t-test p-values at each SM dose for TNF, IL-1 β and IL-6, and updated Figure 5 accordingly. The revised analyses show that porcine-brain SM significantly inhibits IL-1 β and TNF release in response to *S. aureus* stimulation ($p < 0.05$), with minimal impact on IL-6 (Figure 5A–F). Chicken-yolk SM (10 μM –10 ng/mL) significantly reduces IL-1 β and IL-6 secretion at the highest dose under both *S. aureus* and LPS stimulation ($p < 0.05$), while TNF exhibits a consistent but non-significant downward trend (Figure 5G–L).

Regarding the selection of volunteers for the functional experiments, this was done with buffy coats from anonymous blood donors from the Sanquin Bloodbank Nijmegen. These are healthy volunteers living in the region of Gelderland, The Netherlands, but based on the confidentiality rules we did not obtain demographic data. However, considering the paired comparisons of the effects of the metabolites within an individual, the demographics of the volunteers are highly unlikely to have impacted the conclusions of the experiments.

Results (page 9, lines 283-292):

“In summary, SM showed consistent negative correlations with monocyte-derived cytokine responses in the multiple cohorts, suggesting a potential inhibitory effect of SM on the innate immune function. To validate this finding, we stimulated peripheral blood mononuclear cells (PBMCs) with either *S. aureus* or lipopolysaccharide (LPS) for 24h, followed by measurement of proinflammatory cytokines in the supernatant of the stimulated cells (Figure 5). Pre-treatment of human PBMCs with porcine brain derived SM significantly inhibited IL-1 β and TNF release in response to *S. aureus* stimulation, but had minimal impact on IL-6 (Figure 5A–F). Chicken yolk derived SM (10 μM –10 ng/mL) prior to 24 h stimulation with heat-killed *S.*

aureus or LPS led to a significant reduction in IL-1 β and IL-6 secretion at the highest dose, while TNF exhibited a non-significant downward trend (Figure 5G–L). No effects of the various conditions on cell viability was observed.”

Discussion (page 14, lines 451-454):

Additionally, our in vitro validation was limited by PBMC availability, and cytokine measurements were only taken at 24 h—missing potential early secretion peaks. The small donor cohort further introduces inter-individual variability. Future work should expand the functional assays to include expanded timeframes of stimulation.

Figure 5. Human PBMCs were stimulated with LPS or heat-killed *S. aureus* in the absence of presence of chicken yolk or porcine brain sphingomyelin. Human PBMCs from six healthy donors (two independent experiments, n=6) were pre-incubated for 1 h with vehicle control (RPMI) or increasing concentrations (10 μ M, 1 μ M, 100 ng/mL, 10 ng/mL) of sphingomyelin isolated from porcine brain (panels A–F) or chicken egg yolk (panels G–L), then stimulated for 24 h with heat-killed *Staphylococcus aureus* (A–C, G–I) or LPS (D–F, J–L). Cytokine levels in supernatants were measured by ELISA for IL-6 (A, D, G, J), TNF (B, E, H, K) and IL-1 β (C, F, I, L). Bars represent mean \pm SEM with individual donor values overlaid

as dots. Statistical significance versus RPMI control (paired two-tailed t-test) is indicated by brackets: * $p < 0.05$, ** $p < 0.01$, *** $p < 0.001$, **** $p < 0.0001$.

Comment 15

•It is not immediately clear whether whole blood or PBMC analysis was performed in the Cohort_AF and Cohort_EU2.

Response to Comment 15

Thank you for your comment. To address the reviewer's comments, we have added the following sentences to the manuscript:

Materials and Methods (page 18, lines 602–603):

“In Cohort_AF, cytokine responses were measured using WB only; in Cohort_EU2, using PBMCs only; and in Cohort_EU1, both WB and PBMC stimulations were performed.”

Comment 16

•The authors should specify which GWAS was used for Mendelian randomization (e.g. GWAS catalogue identifier).

Response to Comment 16

Thank you for your comment. To address the reviewer's comments, we have added the following sentences to the manuscript:

Materials and Methods (page 23, lines 747–750):

“We leveraged GWAS summary statistics from the COVID-19 Host Genetics Initiative Release 5 study of hospitalized versus non-hospitalized COVID-19 cases(Initiative, 2020) (GWAS Catalog accession GCST011080, EBI dataset ebi-a-GCST011080).”

Comment 17

•Line 435: The filtering step is not well described, making it difficult to understand how and why feature-cytokine pairs were filtered.

Response to Comment 17

Thank you for your comment. To address the reviewer's comments, we have added the following sentences to the manuscript:

Materials and Methods (page 19-20, lines 632–653):

“We first assessed the extent to which age, sex, and BMI confounded raw metabolite–cytokine correlations. In three cohorts (Cohort_AF, Cohort_EU1 and Cohort_EU2), we performed Spearman rank correlations between metabolite features and each demographic variable (sex,

age, BMI), applying Benjamini–Hochberg FDR correction ($FDR < 0.05$). Before adjustment, the proportions of metabolites significantly associated with each covariate were as follows (Appendix Figure S6): Age: 50.7% (Cohort_AF), 32.6% (Cohort_EU1), 22.1% (Cohort_EU2); Sex: 43.9% (Cohort_AF), 53.1% (Cohort_EU1), 39.9% (Cohort_EU2); BMI: 20.6% (Cohort_AF), 18.4% (Cohort_EU1), 0% (Cohort_EU2).

To remove these effects, we fitted a multivariable linear regression model for each metabolite in R (v4.x). In Cohort_AF and Cohort_EU1—where $\geq 15\%$ of metabolites were BMI-associated—we included sex, age and BMI as covariates. In EU2, where no metabolites showed BMI associations, we included only sex and age to avoid over-adjustment. After model fitting, we extracted the residuals and re-tested their Spearman correlations with sex, age and BMI ($FDR < 0.05$). Post-adjustment, fewer than 1% of metabolites remained significantly correlated with any covariate in all cohorts (Cohort_AF: 0.7% age, 1.2% sex, 0.7% BMI; Cohort_EU1: 0.1% age, 0.1% sex, 0% BMI; Cohort_EU2: 0% age, 0% sex, 0% BMI; Appendix Figure S6), demonstrating effective confounder removal.

For downstream metabolite–cytokine association testing, all analyses were conducted on these de-confounded residuals. We again performed Spearman correlations against each immune cytokine and retained only those metabolite features with $FDR < 0.05$ in at least one cytokine comparison for further analysis. All model fitting and residual extraction used R's `lm()` function, correlations were computed with `corr.test()`, and multiple-testing correction was applied via `p.adjust(method="fdr")`.

Comment 18

•Figure S6 Appears to be a screenshot with highlighted spelling suggestions from Microsoft Word.

Response to Comment 18

Thank you very much for pointing out this issue. We have updated Appendix Figure S6 with a high-resolution version that no longer contains any highlighted spelling suggestions.

Comment 19

•iMetaboMap: There is no clear link to download the data. Additionally, the statement "All rights are reserved by: Prof. Yang Li Group" seems unnecessarily restrictive if the authors intend for others to make use of their dataset.

Response to Comment 19

Thank you for your helpful comment. We have added a prominent “Download Data” tab under the Data panel on the IMetaboMap interface. Users can now click this tab to instantly download all relevant data and results. We have also updated the data usage statement to clarify that the dataset is freely available for academic use.

Responses to Reviewer 3

Comment 1

Understanding of the Story

This study investigates the complex relationship between circulating metabolites and immune responses across different human populations, including two Western European cohorts and one African cohort. The authors explore the interaction of metabolic pathways, especially sphingomyelin and glycerophospholipid metabolism, with cytokine responses. The ultimate aim is to elucidate how these metabolic factors influence disease pathogenesis, particularly inflammation and infectious diseases such as COVID-19.

Response to Comment 1

Thank you very much for your comment.

Comment 2

Key Conclusions: Specific Findings and Concepts

- Identified consistent associations between sphingomyelin metabolism and monocyte-derived cytokine responses (TNF, IL-1 β , IL-6).
- Demonstrated sex-specific differences in phosphatidylcholine correlations with cytokine responses.
- Established a causal link between circulating sphingomyelin concentrations and COVID-19 severity using Mendelian randomization.
- Created an open-access online resource (IMetaboMap) for exploring metabolite-cytokine interactions across populations.

Methodology and Model System

- Multi-cohort study including two European (534 and 324 individuals) and one African cohort (323 individuals).
- Flow-injection time-of-flight mass spectrometry was utilized to quantify plasma metabolite profiles.
- Cytokine production in response to bacterial (e.g., *S. aureus*) and viral stimuli was measured using ELISA and Olink technology.
- Weighted gene co-expression network analysis (WGCNA) and Spearman correlation analyses to identify relevant metabolite-immune response associations.
- Mendelian randomization (MR) analysis using publicly available genomic and metabolomic datasets.
- Experimental validation using peripheral blood mononuclear cells (PBMCs) stimulated with

S. aureus or lipopolysaccharide (LPS).

Response to Comment 2

Thank you very much for your comment.

Comment 3

General Remarks

Are you convinced of the key conclusions?

The authors provided evidence to support some but not all of their key conclusions. For example, the associations between sphingomyelin (SM) metabolism and cytokine responses, especially monocyte-derived cytokines such as IL-1 β , IL-6, and TNF, seem robust. The *in vitro* experiments (Fig. 5) do not appear to be statistically significant. The Mendelian Randomization (MR) analyses linked SM levels with COVID-19 severity appear robust. However, due to the relatively weak connections among the experimental designs and the substantial differences between cohorts, the broader applicability and generalizability of the study's conclusions are somewhat limited.

Response to Comment 3

Thank you very much for this insightful feedback. We have strengthened our analyses and clarified our key conclusions as follows:

1. SM–cytokine responses associations

We introduced a “Composite SM score” in each cohort by averaging the raw abundances of all sphingomyelin–related metabolites per subject, yielding an intuitive measure of overall circulating SM. Within Cohort_EU1, Cohort_EU2 and Cohort_AF, we then computed Spearman correlations (Fisher’s Z–transformed) between this Composite SM score and key cytokine outputs (IL-1 β , IL-6, TNF, IFN- γ) following *S. aureus* stimulation, applying FDR correction to identify significant associations. We carried out a cross-cohort meta-analysis for each cytokine using a random-effects model (with fixed-effect as a sensitivity check) to pool effect sizes and 95% CIs. These results, presented in Appendix Figure S10, closely mirror our original module-level findings and further confirm that higher circulating SM is robustly and negatively correlated with inflammatory cytokine release across populations.

2. *In vitro* validation

Thank you for your comment, and we apologize for the omission of the statistical analysis in the initial manuscript. In the revised manuscript, we now added the paired two-tailed t-tests and annotated statistical significance in Figure 5. Under *S. aureus* stimulation, porcine-brain SM at both 1 μ M and 10 μ M significantly inhibited IL-1 β and TNF ($p < 0.05$), with only a limited effect on IL-6; egg-yolk SM at 10 μ M significantly reduced IL-1 β and IL-6 ($p < 0.05$), while TNF showed a downward trend that did not reach statistical significance.

3. Mendelian randomization robustness

In this revision, we extended our instrument selection beyond the primary SNP set ($P < 5 \times 10^{-7}$; 10 SM-associated mQTLs) to include a stricter threshold ($P < 5 \times 10^{-8}$; 6 mQTLs) and a more relaxed threshold ($P < 1 \times 10^{-6}$; 21 mQTLs). Across all three thresholds, both the IVW and weighted-median estimators yielded protective effects of higher SM on COVID-19 severity with consistent directionality ($\beta_{IVW} \approx -0.13$ to -0.12). Specifically, at $P < 5 \times 10^{-7}$ the IVW p-value was 7.9×10^{-3} ; at $P < 1 \times 10^{-6}$ it was 5.3×10^{-4} ; and at $P < 5 \times 10^{-8}$ the IVW estimate remained protective ($\beta = -0.13$, $p = 7.16 \times 10^{-3}$) despite the smaller number of instruments. Sensitivity analyses at each threshold showed no evidence of horizontal pleiotropy (MR-Egger intercept $p > 0.05$) or heterogeneity (Cochran's Q $p > 0.05$). These results are detailed in Supplementary Fig. S11A–H and Tables EV13–15. In summary, these findings further demonstrate the consistency of our causal inference in both directionality and robustness: elevated circulating SM levels are associated with a reduced risk of COVID-19 hospitalization. Moreover, in the Discussion section, we will explicitly acknowledge this limitation and recommend that future studies, when fully annotated mQTL summary statistics are available, apply stricter SNP selection thresholds and larger sample sizes to replicate our MR analyses and further validate the causal relationship between sphingomyelin levels and COVID-19 severity.

These updates have been incorporated into the manuscript. In response to the reviewer's comments, we added the following sentences to the manuscript:

Results (page 8-9, lines 266–282):

“To complement these single-molecule findings, we constructed a Composite SM score by averaging the raw abundance values of all SM species that were individually associated with IL-6, IL-1 β , TNF, and IFN- γ responses in the three cohorts. As shown in Appendix Figure S10, we calculated Spearman correlation coefficients (Fisher's Z) between the Composite SM and *S. aureus*-induced IL-6, IL-1 β , TNF, and IFN- γ responses in each cohort (Cohort_AF, Cohort_EU1, and Cohort_EU2), then performed a meta-analysis on these estimates. The association between Composite SM and *S. aureus*-induced IL-6 pooled to $r = -0.136$ (95 % CI [-0.185, -0.086], $p = 9.96e-8$) under the fixed-effect model and $r = -0.129$ (95 % CI [-0.242, -0.013], $p = 0.0293$) under the random-effects model, indicating that SM consistently suppresses IL-6 secretion in all cohorts with a highly significant negative effect (Appendix Figure S10A). Furthermore, we conducted a combined meta-analysis of the Spearman correlations between Composite SM and monocyte-derived cytokines (IL-1 β , IL-6, TNF) across the three cohorts (Cohort_AF, Cohort_EU1, and Cohort_EU2). As shown in Appendix Figure S10E, the fixed-effect model pooled $r = -0.128$ (95 % CI [-0.156, -0.099], $p = 4.7e-18$), and the random-effects model pooled $r = -0.117$ (95 % CI [-0.189, -0.043], $p = 0.00191$).

These results demonstrate that higher Composite SM levels are consistently associated with lower monocyte-derived cytokine production, regardless of whether samples were obtained from African or European cohorts.”

Results (page 9, lines 283-292):

“In summary, SM showed consistent negative correlations with monocyte-derived cytokine responses in the multiple cohorts, suggesting a potential inhibitory effect of SM on the innate immune function. To validate this finding, we stimulated peripheral blood mononuclear cells (PBMCs) with either *S. aureus* or lipopolysaccharide (LPS) for 24h, followed by measurement of proinflammatory cytokines in the supernatant of the stimulated cells (Figure 5). Pre-treatment of human PBMCs with porcine brain derived SM significantly inhibited IL-1 β and TNF release in response to *S. aureus* stimulation, but had minimal impact on IL-6 (Figure 5A–F). Chicken yolk derived SM (10 μ M–10 ng/mL) prior to 24 h stimulation with heat-killed *S. aureus* or LPS led to a significant reduction in IL-1 β and IL-6 secretion at the highest dose, while TNF exhibited a non-significant downward trend (Figure 5G–L). No effects of the various conditions on cell viability was observed.”

Results (page 10, lines 321–326):

“To further assess robustness, we extracted instruments at two additional thresholds: a more stringent $P < 5 \times 10^{-8}$ (6 SM-related mQTLs) and a more relaxed $P < 1 \times 10^{-6}$ (21 mQTLs). At $P < 1 \times 10^{-6}$, both methods remained significant (weighted median $p = 3.85 \times 10^{-2}$, $\beta = -0.11$; IVW $p = 5.27 \times 10^{-3}$, $\beta = -0.12$), and even under the more stringent threshold with fewer instruments ($P < 5 \times 10^{-8}$), the effect remained in the same direction ($\beta = -0.13$, $p = 7.16 \times 10^{-3}$). These analyses are summarized in Appendix **Figure S11A-C and E-G** and detailed in **Table EV13.**”

Results (page 10, lines 331–334):

“These included tests for horizontal pleiotropy, as indicated by a non-significant MR-Egger intercept ($p > 0.05$, see Table EV14 for $P < 5 \times 10^{-7}$, $P < 1 \times 10^{-6}$ and $P < 5 \times 10^{-8}$ results), assessments of heterogeneity via Cochran’s Q test ($p > 0.05$, shown in Table EV15), and a leave-one-out analysis (illustrated in Figure 6D, Appendix Figure S11D and S11H).”

Discussion (page 13-14, lines 451-454):

Additionally, our in vitro validation was limited by PBMC availability, and cytokine measurements were only taken at 24 h—missing potential early secretion peaks. The small donor cohort further

introduces inter-individual variability. Future work should expand the functional assays to include expanded timeframes of stimulation.

Discussion (page 14, lines 457–459):

“Furthermore, when fully annotated mQTL summary statistics are available, future MR studies should apply stricter SNP selection thresholds and larger sample sizes to replicate and strengthen these causal inferences.”

Figure 5. Human PBMCs were stimulated with LPS or heat-killed *S. aureus* in the absence of presence of chicken yolk or porcine brain sphingomyelin. Human PBMCs from six healthy donors (two independent experiments, n=6) were pre-incubated for 1 h with vehicle control (RPMI) or increasing concentrations (10 μM, 1 μM, 100 ng/mL, 10 ng/mL) of sphingomyelin isolated from porcine brain (panels A–F) or chicken egg yolk (panels G–L), then stimulated for 24 h with heat-killed *Staphylococcus aureus* (A–C, G–I) or LPS (D–F, J–L). Cytokine levels in supernatants were measured by ELISA for IL-6 (A, D, G, J), TNF (B, E, H, K) and IL-1β (C, F, I, L). Bars represent mean ± SEM with individual donor values overlaid as dots. Statistical significance versus RPMI control (paired two-tailed t-test) is indicated by brackets: *p < 0.05, **p < 0.01, ***p < 0.001, ****p < 0.0001.

Comment 4

Place the work in its context.

The current study addresses the increasingly relevant area of immunometabolism, exploring how metabolic pathways influence immune responses and their potential implications for

infectious diseases. Previous research has already established connections between metabolism and immune function, but systematic studies using large-scale multi-cohort data, particularly involving diverse ethnic backgrounds, have been lacking. This study uniquely addresses this gap by integrating data from European and African cohorts, thereby increasing the generalizability and relevance of findings.

Response to Comment 4

Thank you very much for this positive comment.

Comment 5

What is the nature of the advance (conceptual, technical, clinical)?

This work is primarily a conceptual and clinical advance. It enhances our understanding of immunometabolic interactions, particularly highlighting sphingolipid metabolism as a crucial regulator of immune responses. Clinically, the study suggests new therapeutic potentials of targeting sphingomyelin pathways in inflammatory and infectious diseases, notably severe COVID-19.

Response to Comment 5

Thank you very much for this positive comment.

Comment 6

How significant is the advance compared to previous knowledge?

The study significantly advances previous knowledge by systematically validating sphingomyelin's immune modulatory role through multi-omics approaches, functional assays, and genetic evidence. It goes beyond existing studies (which have been limited to smaller, single populations or purely correlative analyses) by clearly demonstrating causal and functional roles of sphingolipids. Furthermore, the construction of an open-access tool (IMetaboMap) greatly enhances the utility and impact of their work for future research.

Response to Comment 6

Thank you very much for this positive comment.

Comment 7

What audience will be interested in this study?

- Immunologists and researchers in infectious disease
- Metabolomics researchers
- Clinicians interested in immunomodulatory therapies
- Epidemiologists and public health researchers, especially those focusing on COVID-19 or other severe infectious diseases
- Geneticists interested in Mendelian Randomization approaches

Response to Comment 7

Thank you very much for your comment.

Comment 8

Major points

Specific criticisms related to key conclusions:

- Clarity of the figure's legends:

The experimental descriptions and figure labeling are insufficiently clear. The figures presented lack explicit labels, comprehensive descriptions, and adequate explanatory context, making initial comprehension challenging. For instance, Figure 1's purpose, key findings, and differences across groups are not clearly articulated. Additional explanations and clearly annotated figure legends would significantly enhance readability.

Response to Comment 8

Thank you for your comment. We have thoroughly revised all figure legends to make them fully self-contained and reader-friendly. To address the reviewer's comments, we have added the following sentences to the manuscript:

Figure legends (page 15-17, lines 500–572):

Figure 1. Study overview and analytical workflow.

(A) Cohorts, sample sizes and cytokine profiling summary. Three independent human cohorts were studied: Cohort_EU1 (n = 534, Western Europe, both PBMC and whole-blood assays), Cohort_EU2 (n = 323, Western Europe, PBMC only) and Cohort_AF (n = 325, Sub-Saharan Africa, whole-blood only). For each cohort, plasma was collected for untargeted metabolomics and a panel of ex vivo stimulations.

In the matrix, rows list each bacterial, fungal, viral, TLR-ligand or non-microbial stimulus, and columns are grouped by cell system (macrophages, PBMC or whole blood), stimulation duration (1 day, 2 days or 7 days) and cytokine measured (IL-6, TNF, IL-1 β , IFN- γ , IL-22, IL-17 or IL-10). A \checkmark indicates that the named cytokine was quantified by ELISA under that exact combination of stimulus, cell system and time point. For example, in Cohort_EU1 whole blood stimulated for 2 days with *S. aureus*, IL-6, TNF, IL-1 β and IFN- γ were measured (\checkmark), whereas IL-10 was not (-). Brackets on the left annotate which rows apply to each cohort; only stimuli within a given bracket were assayed in that cohort. This layout allows direct comparison of which stimulus–cell–timepoint–cytokine combinations were profiled across cohorts.

(B) Discovery and validation pipeline. Cohort_EU1 (WB + PBMCs) and Cohort_AF (WB) served as discovery sets, with Cohort_EU2 (PBMC) as an independent replication cohort. Spearman's rank correlations were computed between each metabolite and cytokine response (FDR-corrected), followed by WGCNA-based module detection and pathway enrichment. Functionally relevant hits (e.g. sphingomyelin) were validated by adding exogenous lipid to PBMC cultures and measuring dose-dependent inhibition of TNF, IL-1 β and IL-6. Finally, causal inference was assessed by two-sample

Mendelian randomization, and all associations were integrated into the iMetaboMap web resource for community access.

Figure 2. Metabolite modules associated with cytokine responses induced by *S. aureus* in African and European populations. The specific metabolites within each WGCNA-defined module are listed in Table EV1 (panels A–D). In the three cohorts (Cohort_EU1, Cohort_AF, and Cohort_EU2), 11, 11, 10, and 10 modules, respectively, were found to correlate with cytokine responses (IL-1 β , IL-6, TNF, and IFN- γ) induced by *S. aureus*. Significance thresholds: ***, FDR < 0.001; **, FDR < 0.01; *, FDR < 0.05; +, FDR < 0.1. (E–H) Pathway enrichment analysis for the MEbrown (A), MEGrey (B), MEmagenta (C), and MEmagenta (D) modules. Larger and darker bubbles represent higher $-\log(\text{Pvalue})$, indicating higher significance. Cytokine responses were measured using PBMCs in Cohort_EU1 (A), whole blood in Cohort_EU1 (B), whole blood in Cohort_AF (C), and PBMCs in Cohort_EU2 (D).

Figure 3. Sex-specific metabolite–cytokine associations and pathway differences.

Scatter plots illustrate pathway analysis results for significant correlations (FDR < 0.05) between metabolites and cytokine responses, with distinctions based on gender in varied cohorts: (A) Cohort_AF, male; (B) Cohort_EU1 (PBMCs), male; (C) Cohort_EU1 (PBMCs), female; (D) Cohort_EU1 (WB), female. The darker the bubble color, the larger the $-\log(\text{Pvalue})$, indicating higher significance. (E, F) Differences in the Glycerphospholipid and Linoleic acid metabolism between males and females. Red signifies a significant positive correlation between the metabolite and cytokine response, while green indicates a significant negative correlation. Cells shaded in light red and light green represent $0.05 < \text{FDR} < 0.10$, with light red suggesting a trend towards positive correlation and light green pointing to a trend toward negative correlation. Asterisk * (FDR < 0.05) and hash # ($0.05 < \text{FDR} < 0.10$).

Figure 4. Immune response-related metabolites across multiple cohorts.

Bubble plots showing the results of pathway analysis of metabolites significantly correlated (FDR < 0.01) with *S. aureus*-induced cytokine responses (IL-1 β , IL-6, TNF, and IFN- γ) in Cohort_AF (A), Cohort_EU1 (B), and Cohort_EU1 (PBMCs) (C). (D and E) A 'forest' plot displaying different mean values (center of symbols), confidence limits (95% confidence intervals), and precision levels (denoted by the size or 'weight' of the symbols, where larger symbols signify higher precision). These are shown for the effect sizes derived from individual studies (in black), as well as the aggregate mean values (center of symbols) and 95% confidence intervals (width of symbols) calculated through meta-analysis using both a fixed-effect model and a random-effects model (both in blue).

Figure 5. Human PBMCs were stimulated with LPS or heat-killed *S. aureus* in the absence of presence of chicken yolk or porcine brain sphingomyelin. Human PBMCs from six healthy donors (two independent experiments, n=6) were pre-incubated for 1 h with vehicle control (RPMI) or increasing concentrations (10 μM , 1 μM , 100 ng/mL, 10 ng/mL) of sphingomyelin isolated from

porcine brain (panels A–F) or chicken egg yolk (panels G–L), then stimulated for 24 h with heat-killed *Staphylococcus aureus* (A–C, G–I) or LPS (D–F, J–L). Cytokine levels in supernatants were measured by ELISA for IL-6 (A, D, G, J), TNF (B, E, H, K) and IL-1 β (C, F, I, L). Bars represent mean \pm SEM with individual donor values overlaid as dots. Statistical significance versus RPMI control (paired two-tailed t-test) is indicated by brackets: * $p < 0.05$, ** $p < 0.01$, *** $p < 0.001$, **** $p < 0.0001$.

Figure 6. Causal relationships between sphingomyelin and COVID-19 as assessed by MR analysis.

(A) The regression lines for the inverse variance weighted (IVW), weighted median (WM) method is shown. (B) Forest plot of the effects of SM on COVID-19. It shows the MR effect size (center dot) and 95% CI, estimated with the IVW MR approach. (C) Forest plot of the 10 plasma SM SNPs associated with risk of COVID-19. (D) Forest plots of MR leave-one-out sensitivity results. (E) A graphic summary of the regulation network of sphingomyelin, cytokines responses, and COVID-19.

Figure 7. IMetaboMap is a web-based platform designed for users to explore correlations between metabolites and cytokine responses interactions. This pioneering tool offers insights into the interplay between plasma metabolites and cytokine responses to various stimuli in humans, with the capability to analyze variations across population, sex, and cell system.

Comment 9

- Statistical Significance of sphingomyelin's effect on cytokine secretion

The data in Fig. 5 shows a trend that does not appear statistically significant. There is also an apparent bimodal distribution in response for some cytokines that is unexplained (e.g., IL1beta with porcine brain SP).

Response to Comment 9

Thank you for your comment, and we apologize for the omission of the statistical analysis in the initial manuscript. In the revised manuscript, we now added the paired two-tailed t-tests and annotated statistical significance in Figure 5. Under *S. aureus* stimulation, porcine-brain SM at both 1 μ M and 10 μ M significantly inhibited IL-1 β and TNF ($p < 0.05$), with only a limited effect on IL-6; egg-yolk SM at 10 μ M significantly reduced IL-1 β and IL-6 ($p < 0.05$), while TNF showed a downward trend that did not reach statistical significance.

To address the reviewer's comments, we have added the following sentences to the manuscript:

Results (page 9, lines 283-292):

In summary, SM showed consistent negative correlations with monocyte-derived cytokine responses in the multiple cohorts, suggesting a potential inhibitory effect of SM on the innate immune function. To validate this finding, we stimulated peripheral blood mononuclear cells (PBMCs) with either *S. aureus* or lipopolysaccharide (LPS) for 24h, followed by measurement of proinflammatory cytokines in the

supernatant of the stimulated cells (Figure 5). Pre-treatment of human PBMCs with porcine brain derived SM significantly inhibited IL-1 β and TNF release in response to *S. aureus* stimulation, but had minimal impact on IL-6 (Figure 5A–F). Chicken yolk derived SM (10 μ M–10 ng/mL) prior to 24 h stimulation with heat-killed *S. aureus* or LPS led to a significant reduction in IL-1 β and IL-6 secretion at the highest dose, while TNF exhibited a non-significant downward trend (Figure 5G–L). No effects of the various conditions on cell viability was observed.

Discussion (page 14, lines 451-454):

Additionally, our in vitro validation was limited by PBMC availability, and cytokine measurements were only taken at 24 h—missing potential early secretion peaks. The small donor cohort further introduces inter-individual variability. Future work should expand the functional assays to include expanded timeframes of stimulation.

Figure 5. Human PBMCs were stimulated with LPS or heat-killed *S. aureus* in the absence of presence of chicken yolk or porcine brain sphingomyelin. Human PBMCs from six healthy donors (two independent experiments, n=6) were pre-incubated for 1 h with vehicle control (RPMI) or increasing concentrations (10 μ M, 1 μ M, 100 ng/mL, 10 ng/mL) of sphingomyelin isolated from porcine brain (panels A–F) or chicken egg yolk (panels G–L), then stimulated for 24 h with heat-killed *Staphylococcus aureus* (A–C, G–I) or LPS (D–F, J–

L). Cytokine levels in supernatants were measured by ELISA for IL-6 (A, D, G, J), TNF (B, E, H, K) and IL-1 β (C, F, I, L). Bars represent mean \pm SEM with individual donor values overlaid as dots. Statistical significance versus RPMI control (paired two-tailed t-test) is indicated by brackets: * $p < 0.05$, ** $p < 0.01$, *** $p < 0.001$, **** $p < 0.0001$.

Comment 10

- Generality of Conclusions:

Although the study involved three distinct cohorts with various samples and stimuli, the differences in experimental design across cohorts pose limitations for direct comparison and weaken cross-cohort conclusions. The paper identifies sphingomyelin (SM) metabolism as a significant modulator of immune response. However, the generality of these findings beyond the stimuli used (primarily *S. aureus* stimulation) is unclear. The conclusions about the role of SM in broader immune functions and disease contexts need to be substantiated by experiments involving a wider variety of pathogens or immune stimulants.

Response to Comment 10

Thank you for your comment. We acknowledge that although our study involved three distinct cohorts, variations in the experimental design could indeed limit the direct comparability of results and potentially impact the generality of the conclusions drawn. Specifically, we agree that our conclusions about the role of sphingomyelin metabolism were mainly derived from immune responses to *S. aureus* stimulation.

To address the concern regarding the broader applicability of our findings, we have explicitly stated in the revised discussion that further validation with diverse immune stimulants and pathogens is necessary. Following the reviewer's comment, we have added following text in the discussion:

Discussion (page 13-14, lines 438-443):

“Fourth, although our findings demonstrate that sphingomyelin modulates immune responses following *S. aureus* stimulation, we acknowledge that extending these observations to broader immune contexts requires further validation. Future experiments involving a wider variety of pathogens and immunostimulants are essential to confirm and generalize the role of sphingomyelin metabolism across diverse immune challenges and disease conditions.”

Comment 11

- Causal Relationship Interpretation:

The use of Mendelian randomization (MR) to establish causality between sphingomyelin and COVID-19 severity is compelling but warrants caution. While MR can provide strong causal inference, the potential for pleiotropy or horizontal genetic effects still exists and should be discussed more explicitly to avoid overstating causality.

Response to Comment 11

Thank you very much for your insightful comment. We agree with the reviewer that while Mendelian randomization (MR) provides valuable evidence supporting causality, the interpretation must consider potential pleiotropy or horizontal genetic effects. To explicitly address this limitation, we have included the following statement in the revised discussion:

Discussion (page 14, lines 454-461):

“Finally, while our Mendelian randomization analysis suggests a causal link between circulating sphingomyelin levels and COVID-19 severity, we acknowledge MR’s inherent limitations—particularly the potential for pleiotropy or horizontal genetic effects—and therefore interpret these findings as suggestive rather than definitive. Furthermore, when fully annotated mQTL summary statistics are available, future MR studies should apply stricter SNP selection thresholds and larger sample sizes to replicate and strengthen these causal inferences. Additional confirmatory biological evidence, including targeted in vitro and in vivo experiments, will be essential to fully substantiate and clarify these causal associations.”

Comment 12

• Sex-Specific Differences:

Although sex-specific differences are highlighted, the biological or clinical implications of these differences remain unclear. Are these differences robust enough to inform sex-specific clinical approaches? Sex-specific differences were observed to some extent in cohorts EU1 and AF, but these findings were difficult to replicate in cohort EU2, significantly weakening the strength of this conclusion. Additional evidence and experiments are required to clarify this phenomenon. More analysis or literature support is needed to clarify the clinical relevance.

Response to Comment 12

Thank you for your comment. We acknowledge the reviewer’s concern regarding the clarity of the biological and clinical implications of the observed sex-specific differences. In response to your comments, we performed two complementary analyses and have updated the Discussion to frame their implications and limitations. First, within each cohort we stratified participants by sex, regressed each metabolite on age and BMI to remove confounding, and then calculated Spearman correlations between these residuals and cytokine responses under each stimulus–cell condition. We compared male versus female correlations using Fisher’s r -to- z transformation and two-sample Z tests with Benjamini–Hochberg FDR correction; although several glycerophospholipid metabolites showed nominal sex differences (Table EV16), none remained significant after multiple-testing adjustment. Second, we created composite scores for all phosphatidylcholine-related (mean_PC) and phosphatidate-related (mean_PA) metabolites per sample, adjusted these scores for age and BMI, and compared their Spearman correlations with *S. aureus*–induced cytokines between sexes. A fixed- and random-effects meta-analysis across cohorts revealed a robust male bias only for the PC–TNF association (fixed-effect $r_{diff} \approx 0.74$, $p \approx 2 \times 10^{-68}$; random-effects $r_{diff} \approx 0.75$, $p \approx 1.7 \times 10^{-7}$; $I^2 =$

91.1 %; Appendix Figure S4), while all other score–cytokine relationships were inconsistent. We now explicitly acknowledge that these sex-specific signals are suggestive and not yet clinically actionable.

To address the reviewer’s comments, we have added the following sentences to the manuscript:

Results (page 6-7, lines 184–205):

“Among phosphatidate metabolites, lysophosphatidic acid (LPA)(0:0/18:0) showed a significant negative correlation with TNF response in males of the Cohort_AF, whereas both phosphatidic acid (PA)(16:0/18:1(11Z)) and PA(18:0/18:2(9Z,12Z)) displayed significant negative correlations with IL-6 response in both males and females of Cohort_EU1 (PBMCs). Additionally, PA(16:0/18:1(11Z)) and PA(18:0/18:2(9Z,12Z)) also exhibited significant negative correlations with IL-1 β and TNF response in females of the Cohort_EU1(PBMCs) and Cohort_EU1(WB). Subsequently, statistical testing of correlation coefficients between male and female volunteers revealed that in Cohort_EU1 (PBMCs), sex-specific differences for PA(16:0/18:1(11Z)) versus TNF/IL-1 β and for PA(18:0/18:2(9Z,12Z)) versus TNF/IL-1 β showed suggestive evidence ($p < 0.05$; see Table EV16).

In the phosphatidylcholine group of the glycerophospholipid metabolism pathway, a positive correlation trend with TNF and IFN- γ response was observed solely in males of the Cohort_AF. In contrast, this phosphatidylcholine group showed negative correlations with cytokine responses for both males and females in the European cohorts. For instance, phosphatidylcholine (PC)(14:0/20:1(11Z)) displayed a significant negative correlation with IL-6 in both males and females in Cohort_EU1 (PBMCs). Additionally, this metabolite showed negative correlations with IL-1 β and TNF response in females in Cohort_EU1 (PBMCs) and a negative correlation trend with IL-1 β in females in Cohort_EU1 (WB). Subsequently, statistical testing of correlation coefficients between male and female volunteers revealed that in Cohort_EU1 (PBMCs), the sex-specific difference in correlation coefficients between PC(14:0/20:1(11Z)) and TNF/IL-1 β showed suggestive evidence ($p < 0.05$; see Table EV16). Similarly, in Cohort_AF, sex-specific differences in correlations for PC(15:0/24:0) versus TNF/IFN- γ and for PC(22:6(4Z,7Z,10Z,13Z,16Z,19Z)/24:1(15Z)) versus TNF/IFN- γ showed suggestive evidence ($p < 0.05$; see Table EV16).”

Results (page 7, lines 206–218):

“To explore sex-specific metabolite–cytokine relationships, we summarized phosphatidylcholine (PC) and phosphatidate (PA) species into Composite PC and PA scores, adjusted them for age and BMI within each sex, and correlated them with *S. aureus*–induced IL-6, IL-1 β , TNF and IFN- γ (Spearman’s ρ , FDR < 0.05). We then compared male versus female ρ values via Fisher’s Z and pooled these sex-difference Z’s across cohorts by meta-analysis (Appendix Figure S4D-K). This revealed a pronounced male bias for Composite PC–TNF correlations (fixed-effect $r_{diff} \approx 0.743$, 95 % CI [0.691, 0.787], $p \approx 2 \times 10^{-68}$; random-effects $r_{diff} \approx 0.745$, 95 % CI [0.537, 0.867], $p \approx 1.7 \times 10^{-7}$; $I^2 = 91.1$ %), indicating consistently stronger PC–TNF associations in males (Appendix Figure S4D). In contrast, although fixed-effect models for Composite PC–IL-1 β , –IFN- γ , and –IL-6 suggested positive male biases (all $p < 0.001$), their random-effects 95 % CIs all spanned zero, reflecting high heterogeneity and no consistent sex effect (Appendix Figure S4E-G). Composite PA–cytokine analyses showed no reliable male–female differences for any cytokine (all pooled r_{diff} CIs included zero), indicating similar PA–cytokine associations in both sexes (Appendix Figure S4H-K).”

Discussion (page 14, lines 443-451):

“Five, although we identified sex-specific metabolic markers potentially linked to immune regulation, the current study does not experimentally validate their functional, sex-specific impacts. Future investigations incorporating targeted functional assays—such as sex-stratified immune cell stimulation experiments and mechanistic analyses—are required to robustly confirm these markers’ roles in immune regulation and assess their clinical implications. While our results highlight potential sex-specific differences in immune–metabolic associations, inconsistent replication across cohorts limits their immediate clinical applicability. Therefore, further experimental validation and larger-scale, sex-stratified analyses are essential to substantiate the robustness and clinical relevance of these findings.”

Comment 13

Specify experiments or analyses required to demonstrate the conclusions:

1. Statistical testing of the significance of sphingomyelin's effect on cytokine secretion (Fig. 5).

Response to Comment 13

Thank you for your comment, and we apologize for the omission of the statistical analysis in the initial manuscript. In the revised manuscript, we now added the paired two-tailed t-tests and annotated statistical significance in Figure 5. Under *S. aureus* stimulation, porcine-brain SM at both 1 μ M and 10 μ M significantly inhibited IL-1 β and TNF ($p < 0.05$), with only a limited effect on IL-6; egg-yolk SM at 10 μ M significantly reduced IL-1 β and IL-6 ($p < 0.05$), while TNF showed a downward trend that did not reach statistical significance.

To address the reviewer’s comments, we have added the following sentences to the manuscript:

Results (page 9, lines 283-292):

“In summary, SM showed consistent negative correlations with monocyte-derived cytokine responses in the multiple cohorts, suggesting a potential inhibitory effect of SM on the innate immune function. To validate this finding, we stimulated peripheral blood mononuclear cells (PBMCs) with either *S. aureus* or lipopolysaccharide (LPS) for 24h, followed by measurement of proinflammatory cytokines in the supernatant of the stimulated cells (Figure 5). Pre-treatment of human PBMCs with porcine brain derived SM significantly inhibited IL-1 β and TNF release in response to *S. aureus* stimulation, but had minimal impact on IL-6 (Figure 5A–F). Chicken yolk derived SM (10 μ M–10 ng/mL) prior to 24 h stimulation with heat-killed *S. aureus* or LPS led to a significant reduction in IL-1 β and IL-6 secretion at the highest dose, while TNF exhibited a non-significant downward trend (Figure 5G–L). No effects of the various conditions on cell viability was observed.”

Discussion (page 14, lines 451-454):

“Additionally, our in vitro validation was limited by PBMC availability, and cytokine measurements were only taken at 24 h—missing potential early secretion peaks. The small donor cohort further introduces inter-individual variability. Future work should expand the functional assays to include expanded timeframes of stimulation.”

Figure 5. Human PBMCs were stimulated with LPS or heat-killed *S. aureus* in the absence of presence of chicken yolk or porcine brain sphingomyelin. Human PBMCs from six healthy donors (two independent experiments, n=6) were pre-incubated for 1 h with vehicle control (RPMI) or increasing concentrations (10 μM, 1 μM, 100 ng/mL, 10 ng/mL) of sphingomyelin isolated from porcine brain (panels A–F) or chicken egg yolk (panels G–L), then stimulated for 24 h with heat-killed *Staphylococcus aureus* (A–C, G–I) or LPS (D–F, J–L). Cytokine levels in supernatants were measured by ELISA for IL-6 (A, D, G, J), TNF (B, E, H, K) and IL-1β (C, F, I, L). Bars represent mean ± SEM with individual donor values overlaid as dots. Statistical significance versus RPMI control (paired two-tailed t-test) is indicated by brackets: *p < 0.05, **p < 0.01, ***p < 0.001, ****p < 0.0001.

Comment 14

2. Validation of sex-specific metabolic markers

o Although the manuscript clearly identifies sex-specific metabolic differences, it lacks robust experimental validation that these markers have sex-specific functional impacts in immune regulation.

Response to Comment 14

Response to Comment 14

Thank you very much for your comment. We acknowledge the reviewer's important point that the manuscript identifies sex-specific metabolic differences but lacks robust experimental validation of their functional impacts on immune regulation. We agree that additional experimental validation is necessary to firmly establish these metabolic markers as functionally relevant in a sex-specific context.

To clarify this limitation and suggest future directions, we have included the following statement in our revised manuscript discussion:

Discussion (page 14, lines 443-451):

“Five, although we identified sex-specific metabolic markers potentially linked to immune regulation, the current study does not experimentally validate their functional, sex-specific impacts. Future investigations incorporating targeted functional assays—such as sex-stratified immune cell stimulation experiments and mechanistic analyses—are required to robustly confirm these markers' roles in immune regulation and assess their clinical implications. While our results highlight potential sex-specific differences in immune–metabolic associations, inconsistent replication across cohorts limits their immediate clinical applicability. Therefore, further experimental validation and larger-scale, sex-stratified analyses are essential to substantiate the robustness and clinical relevance of these findings.”

Comment 15

3. Mendelian randomization (MR) findings require additional confirmatory biological evidence

o The manuscript uses MR to suggest causal links between sphingomyelin metabolism and COVID-19 severity. However, MR alone cannot fully establish causality due to potential pleiotropy or confounding genetic effects.

Response to Comment 15

Thank you very much for your important comment. We agree that MR analyses alone cannot definitively establish causality without further experimental validation. To address this point clearly, we have added the following clarification in the revised manuscript:

Discussion (page 14, lines 454-461):

“Finally, while our Mendelian randomization analysis suggests a causal link between circulating sphingomyelin levels and COVID-19 severity, we acknowledge MR's inherent limitations—particularly the potential for pleiotropy or horizontal genetic effects—and therefore interpret these findings as suggestive rather than definitive. Furthermore, when fully annotated mQTL summary statistics are available, future MR studies should apply stricter SNP selection thresholds and larger sample sizes to replicate and strengthen these causal inferences. Additional confirmatory biological evidence, including targeted in vitro and in vivo experiments, will be essential to fully substantiate and clarify these causal associations.”

25th Jul 2025

Manuscript Number: MSB-2025-12900R

Title: Deciphering Cross-Cohort Metabolic Signatures of Immune Responses and Their Implications for Disease Pathogenesis

Author: Jianbo Fu

Nienke Unen

Andrei Sarlea

Nhan Nguyen

Martin Jaeger

Javier Bataller

Valerie Koeken

L. Charlotte Bree

Vera P. Mourits

Simone Moorlag

Godfrey Temba

Vesla I. Kullaya

Quirijn de Mast

Leo A.B. Joosten

Cheng-Jian Xu

Mihai Netea

Yang Li

Dear Prof Li,

Thank you for submitting the revised version of your manuscript. We have now received feedback from all three reviewers. As you will see from their comments below, Reviewer #2 is generally satisfied with the revisions you have made. However, both Reviewers #1 and #3 have raised concerns regarding the in vitro experimental validation.

We ask that you carefully address the issues related to the statistical analysis and ensure that all results are reported using appropriate statistical tests, regardless of whether they reach significance. As noted by Reviewer #2 in the previous review round, the small sample size may limit statistical significance; if this is the case, it should be clearly acknowledged in your manuscript, and the related statements should be revised accordingly. The other remaining minor issues need to be addressed as well.

On a more editorial level:

1. Reduce the keyword number to five.

2. Please ensure that the funding information entered in the submission system is consistent with the information provided in the manuscript text. Currently, the following funding sources are missing from the submission system: Radboud University MedicalCenter Hypatia; Spinoza grant of the Netherlands Organization for Scientific research; COFONI (COVID19 Research Network of the State of Lower Saxony) with funding from the Ministry of Science and Culture of Lower Saxony, Germany (14-76403-184-3); Joint Programming Initiative, A Healthy Diet for a Healthy Life (JPI-HDHL), and ZonMw (the Netherlands Organization for Health Research and Development), within the framework of the 'TransMic' and 'TransInf' projects.

3. "Declaration of Interests" should be renamed to "DISCLOSURE AND COMPETING INTERESTS STATEMENT"

4. Remove "Author contribution" section from the manuscript file.

5. Appendix: The appendix file must be submitted in PDF format. The title page should read "Appendix for [manuscript title]" instead of "Supplementary Figures for:".

6. EV datasets or EV tables

- The source file names, titles, legends, and manuscript callouts should be updated to use the format Dataset EV1-EV10 instead of Tables EV1-EV3, EV7-EV12, and EV17, as these tables are relatively large and better categorized as datasets. Legends should continue to be provided in a separate tab/sheet within each Excel file, as currently done.

- The remaining tables (EV4-EV6 and EV13-EV16) should be renamed Table EV1-EV7, with appropriate manuscript callouts. Legends for these should be placed directly above the tables within each Excel file.

7. Please note all callouts should be listed sequentially.

8. Legends for Appendix figures and Tables EV1-EV17 should be removed from manuscript.
9. The "Conclusion" section should be removed, and its content integrated into the "Discussion" section.
10. Please add specific URLs for the deposited datasets in the Data availability section.
11. Please provide the synopsis image in the required size and format: 550 px width, 400-600 px height, and PNG format. The current version is too large.
12. During a standard image analysis, we detected potential aberrations in the figure set.

We kindly invite you to review the Appendix Figure S8 yourself, as TNF α , SM(d18:1/12:0) in PBMC and whole blood appear to be duplicated. If you make any changes to the figure, please include a detailed point-by-point description of what you have changed and why.

13. Section order should be corrected to the following: Title page - Abstract -Keywords - Introduction - Results - Discussion - Methods - DataAvailability - Acknowledgements - Disclosure and CompetingInterests Statement - References - Figure Legends - Table(s) -Expanded View Figure Legends.

14. Please address the following comments related to figure legends:

- Please note that the exact p values are not provided in the legends of figures 5B, C, G, I, J, L
- Please indicate the statistical test used for data analysis in the legends of figures 2E-H; 3A-D; 4A-C
- Please note that the box plots need to be defined in terms of minima, maxima, centre, bounds of box and whiskers, and percentile in the legend of figure 5K
- Please note that information related to n is missing in the legends of figures 4D, E; 6A-D
- Please note that the error bars are not defined in the legends of figures 6A, C, D.
- Please note that the measure of center for the error bars needs to be defined in the legend of figure 6B

Click on the link below to submit your revised paper.

Kind regards,
Jingyi

Jingyi Hou, PhD
Senior Editor
Molecular Systems Biology

If you do choose to resubmit, please click on the link below to submit the revision online before 24th Aug 2025.

*** PLEASE NOTE *** As part of the EMBO Press transparent editorial process initiative (see our Editorial at <https://dx.doi.org/10.1038/msb.2010.72> , Molecular Systems Biology will publish online a Review Process File to accompany accepted manuscripts. When preparing your letter of response, please be aware that in the event of acceptance, your cover letter/point-by-point document will be included as part of this File, which will be available to the scientific community. More information about this initiative is available in our Instructions to Authors. If you have any questions about this initiative, please contact the editorial office (msb@embo.org).

Reviewer #1:

The authors have done a good job to address the majority of our concerns, mostly by including additional analysis, additional details of experiments/analysis, or modifying the text for more reasonable interpretation of the results.

However the in vitro results (Figure 5) are still of concern. Previously there was no statistical analysis, and now they have added paired T tests which is not a suitable test for such data (group of 4). Running such two paired tests within a group of 4 raises the type I error rate (risk of false positives) to 26%, and furthermore there is no FDR applied at all so the "significant" results they report provide no confidence. In addition applying T tests assumes normality of the data (from looking at the graphs the data does not appear to follow a normal distribution).

These points in addition to the lack of any dose response, and the lack of TNF response to LPS stimulation (it should still be detectable at this timepoint despite the authors response) which is a positive control does not provide any confidence in this set of results.

In summary i think this section should be removed from the manuscript, or the appropriate statistical test (Friedman non-parametric ANOVA) should be applied and those results reported.

Reviewer #2:

Jianbo et al have addressed all the concerns I had about the manuscript. I also believe they have addressed the issues raised by the other two reviewers. In my opinion, the manuscript is suitable for publication, but I would like the authors to make 3 minor modifications:

1.Line 146: "with no heterogeneity observed ($I^2 = 0\%$, $Q = 1.22$, $p = 0.748$)". It would be more appropriate to say "low heterogeneity" and provide the confidence intervals for I^2 , as I^2 estimates are particularly noisy when few studies are integrated (<https://doi.org/10.1186/s12874-015-0024-z>)

2.The authors provided a very detailed and insightful response to my comment 8 (page 26 of by point-by-point response), the manuscript would benefit from including some of this information/discussion, as only point 1 of their response seems to be reflected in the manuscript.

3.I don't quite understand the claim of the authors that "attempted to incorporate sphingomyelin variants from Xu et al. (2023;<https://doi.org/10.1038/s41586-023-05844-9>) as additional mQTL sources; however, the publicly available data did not include complete per-metabolite SNP effect summaries)" as I was able to easily find complete summary stats following the link in the paper (<https://www.omicspred.org/downloads>). A quick search also found an even more recent study with full summary statistics for sphingomyelin mQTL (GCST90302103; <https://www.ebi.ac.uk/gwas/studies/GCST90302103>). I would strongly suggest removing the following from the manuscript:

"Furthermore, when fully annotated mQTL summary statistics are available, future MR studies should apply stricter SNP selection thresholds and larger sample sizes to replicate and strengthen these causal inferences."

As it suggests that either a) the authors didn't do a proper search, or b) the authors claim that the available summary stats are not well annotated.

Reviewer #3:

In response to our previous review, the authors have addressed some but not all of the identified weaknesses.

The major problem is that the functional validation data (Fig. 5) is very weak. Only a few conditions reach statistical significance. None of the SMs show a dose-dependent response. In addition, with 48 comparisons (4 dosages, 12 panels), have the authors included a multiple hypotheses correction to their statistical analysis? It is also unclear why a "paired" t-test is used. In general, paired t-tests should only be used on measurements taken on the same subjects before and after intervention. Unless we are mistaken, these are independent wells of PBMCs and should therefore be analyzed with unpaired t-tests. In summary, the functional validation is very weak and undermines the conclusions of this manuscript.

Response to reviewers' Comments

We thank you and the three reviewers for their thorough and insightful evaluations of our work. We have carefully addressed every comment, and believe that these revisions have significantly strengthened the manuscript. Below, you will find our point-by-point responses to each reviewer's comments. For clarity, reviewer comments are shown in black, our responses in blue, and all additions to the main text appear in red—both here and in the revised manuscript.

Responses to Reviewer 1

Comment 1

The authors have done a good job to address the majority of our concerns, mostly by including additional analysis, additional details of experiments/analysis, or modifying the text for more reasonable interpretation of the results.

Response to Comment 1

We would like to thank the reviewer for the positive and very constructive feedback on our manuscript. A detailed point-by-point response to the individual comments is given below and changes made to the manuscript are indicated.

Comment 2

However the in vitro results (Figure 5) are still of concern. Previously there was no statistical analysis, and now they have added paired T tests which is not a suitable test for such data (group of 4). Running such two paired tests within a group of 4 raises the type I error rate (risk of false positives) to 26%, and furthermore there is no FDR applied at all so the "significant" results they report provide no confidence. In addition applying T tests assumes normality of the data (from looking at the graphs the data does not appear to follow a normal distribution). These points in addition to the lack of any dose response, and the lack of TNF response to LPS stimulation (it should still be detectable at this timepoint despite the authors response) which is a positive control does not provide any confidence in this set of results. In summary i think this section should be removed from the manuscript, or the appropriate statistical test (Friedman non-parametric ANOVA) should be applied and those results reported.

Response to Comment 2

Thank you for this important comment. We sincerely apologize for the mistake in the previous version of the manuscript, where the figure legend incorrectly described the statistical method. In fact, the statistical analysis for Figure 5 was conducted using Friedman non-parametric ANOVA with Dunn's multiple comparisons test (implemented in GraphPad Prism 10), as now correctly stated in the revised legend and Methods section. We have also further discussed the limitations of the experimental design, including the observed lack of TNF response to LPS stimulation, in the Discussion section. We thank the reviewer for bringing this oversight to our attention, and have corrected the description throughout the manuscript.

We therefore consider these data useful in terms of functional validation of the genetic data. We would therefore propose to retain these results. However, if the editorial team proposes to eliminate this set of data from the manuscript, we would be willing to accept that.

To address the reviewer's comments, we have added the following sentences to the manuscript:

Methods (page 21, lines 693–695):

“Data from repeated measures were analyzed using Friedman non-parametric ANOVA followed by Dunn's multiple comparisons test (GraphPad Prism 10). Adjusted p-values are reported for all comparisons.”

Discussion (page 14, lines 462–464):

“The lack of a clear dose-response and the absence of a robust TNF response to LPS stimulation was surprising and is most likely technical. This limitation should be addressed in future studies.”

Figure 5 (page 28-29, lines 969–984):

“Figure 5. Human PBMCs were stimulated with LPS or heat-killed *S. aureus* in the absence of presence of chicken yolk or porcine brain sphingomyelin. Human PBMCs from six healthy donors (two independent experiments, n=6) were pre-incubated for 1 h with vehicle control (RPMI) or increasing concentrations (10 μ M, 1 μ M, 100 ng/mL, 10 ng/mL) of sphingomyelin isolated from porcine brain (panels A–F) or chicken egg yolk (panels G–L), then stimulated for 24 h with heat-killed *Staphylococcus aureus* (A–C, G–I) or LPS (D–F, J–L). Cytokine levels in supernatants were measured by ELISA for IL-6 (A, D, G, J), TNF (B, E, H, K) and IL-1 β (C, F, I, L). Bars represent mean \pm SEM with individual donor values overlaid as dots. For panel K, box plots represent the median (centre line), the 25th and 75th percentiles (bounds of the box), whiskers indicate the minimum and maximum values. Statistical analysis was performed using Friedman non-parametric ANOVA followed by Dunn's multiple comparisons test versus RPMI control. Statistical significance versus RPMI control is indicated by brackets: *p < 0.05, **p < 0.01, ***p < 0.001, ****p < 0.0001. Exact adjusted p values (Dunn's test): B: RPMI vs. 1 μ M/mL, p = 0.0076; C: RPMI vs. 10 μ M/mL, p = 0.0139; RPMI vs. 1 μ M/mL, p = 0.0423; G: RPMI vs. 10 μ M/mL, p = 0.0423; I: RPMI vs. 10 μ M/mL, p = 0.0010; J: RPMI vs. 10 μ M/mL, p = 0.0041; L: RPMI vs. 10 μ M/mL, p < 0.0001; RPMI vs. 1 μ M/mL, p = 0.0041.”

Responses to Reviewer 2

Comment 1

Jianbo et al have addressed all the concerns I had about the manuscript. I also believe they have addressed the issues raised by the other two reviewers. In my opinion, the manuscript is suitable for publication, but I would like the authors to make 3 minor modifications:

Response to Comment 1

We would like to thank the reviewer for the positive and very constructive feedback on our manuscript. A detailed point-by-point response to the individual comments is given below and changes made to the manuscript are indicated.

Comment 2

1.Line 146: "with no heterogeneity observed ($I^2 = 0\%$, $Q = 1.22$, $p = 0.748$)". It would be more appropriate to say "low heterogeneity" and provide the confidence intervals for I^2 , as I^2

estimates are particularly noisy when few studies are integrated ([*** The original URL has been rewritten. ***])

Response to Comment 2

We thank the reviewer for highlighting this important methodological point. In line with the suggestion, we have revised the manuscript to replace “no heterogeneity” with “low heterogeneity” and have now reported the 95% confidence interval for I^2 . We agree that I^2 estimates can be biased and imprecise in meta-analyses with few studies, and that confidence intervals are essential for accurate interpretation in this context. We have cited the recommended reference (von Hippel, 2015) in the manuscript to clarify this point and to highlight the limitations of I^2 estimation in small meta-analyses.

To address the reviewer’s comments, we have added the following sentences to the manuscript:

Results (page 5, lines 148–149):

“...with low heterogeneity observed ($I^2 = 0\%$, 95% CI: 0%–84.8%, $Q = 1.22$, $p = 0.75$).”

Methods (page 19, lines 628–630):

“The imprecision and potential bias of the I^2 statistic in small meta-analyses have been discussed in detail by von Hippel (von Hippel, 2015), who recommends that confidence intervals for I^2 should always be reported and interpreted cautiously.”

Comment 3

2.The authors provided a very detailed and insightful response to my comment 8 (page 26 of by point-by-point response), the manuscript would benefit from including some of this information/discussion, as only point 1 of their response seems to be reflected in the manuscript.

Response to Comment 3

We thank the reviewer for highlighting this point. We agree that the manuscript would benefit from a more thorough discussion of the experimental design, the differences between whole blood and PBMC assays, and the influence of cell number and immune-cell composition on cytokine responses, as detailed in our response to comment 8. In the revised manuscript, we have now incorporated additional discussion summarizing points 2–4 of our previous response in the Discussion section. We believe this addition provides further clarity for readers regarding the limitations and interpretation of our findings.

Discussion (page 13, lines 427–437):

“In addition, our study design included both whole blood (WB) and peripheral blood mononuclear cell (PBMC) assays, each with distinct advantages and limitations. The WB stimulation system, containing a mix of immune and non-immune cells, is subject to greater inter-individual variability and reflects the complexity of in vivo immune responses. In contrast, the PBMC assay, with standardized cell numbers and exclusion of non-PBMC elements, yields more reproducible cytokine measurements. Differences in cell composition and abundance, particularly for monocyte and T-cell subsets, are likely to influence cytokine outputs in both systems. Although our findings suggest specific associations between sphingolipid metabolism and monocyte-derived cytokine production, we did not quantify immune-cell subpopulations in this study. Future work including detailed immune-cell phenotyping will be required to distinguish effects of cell abundance versus activation state.”

Comment 4

3. I don't quite understand the claim of the authors that "attempted to incorporate sphingomyelin variants from Xu et al. (2023; [*** The original URL has been rewritten. ***]) as additional mQTL sources; however, the publicly available data did not include complete per-metabolite SNP effect summaries)" as I was able to easily find complete summary stats following the link in the paper (<https://www.omicspred.org/downloads>). A quick search also found an even more recent study with full summary statistics for sphingomyelin mQTL (GCST90302103; <https://www.ebi.ac.uk/gwas/studies/GCST90302103>). I would strongly suggest removing the following from the manuscript:

"Furthermore, when fully annotated mQTL summary statistics are available, future MR studies should apply stricter SNP selection thresholds and larger sample sizes to replicate and strengthen these causal inferences."

As it suggests that either a) the authors didn't do a proper search, or b) the authors claim that the available summary stats are not well annotated.

Response to Comment 4

We thank the reviewer for highlighting the availability of complete summary statistics for sphingomyelin mQTL, and we apologize for any confusion our previous statement may have caused. As suggested, we have removed the sentence

"Furthermore, when fully annotated mQTL summary statistics are available, future MR studies should apply stricter SNP selection thresholds and larger sample sizes to replicate and strengthen these causal inferences."

from the manuscript to avoid any misunderstanding. We appreciate the reviewer's guidance in clarifying this point.

Responses to Reviewer 3

Comment 1

In response to our previous review, the authors have addressed some but not all of the identified weaknesses.

The major problem is that the functional validation data (Fig. 5) is very weak. Only a few conditions reach statistical significance. None of the SMs show a dose-dependent response. In addition, with 48 comparisons (4 dosages, 12 panels), have the authors included a multiple hypotheses correction to their statistical analysis? It is also unclear why a "paired" t-test is used. In general, paired t-tests should only be used on measurements taken on the same subjects before and after intervention. Unless we are mistaken, these are independent wells of PBMCs and should therefore be analyzed with unpaired t-tests. In summary, the functional validation is very weak and undermines the conclusions of this manuscript.

Response to Comment 1

Thank you for raising these important concerns regarding the functional validation data and the statistical analysis of Figure 5. As described in our response to Reviewer #2, Comment 2, we sincerely apologize for the confusion in the previous version of the manuscript, where the statistical method was incorrectly described in the figure legend. In fact, the data were analyzed using the Friedman non-parametric ANOVA with Dunn's multiple comparisons test (implemented in GraphPad Prism 10), which is more appropriate for repeated measures data and addresses the issue of multiple comparisons.

We also acknowledge that the in vitro data are limited by the lack of a clear dose-dependent response and that only some conditions approached statistical significance after correction. These limitations, including the absence of a robust TNF response to LPS stimulation, have been explicitly discussed in the revised Discussion section. We have also corrected the description of the statistical analysis throughout the manuscript to avoid further confusion.

Therefore, as also mentioned earlier in our response, we believe these data are useful as functional validation of the genetic data. We would therefore propose to retain these results. However, if the editorial team proposes to eliminate this set of data from the manuscript, we will do that in the re-revised manuscript.

To clarify these points for the reader, we have made the following changes to the manuscript:

Methods (page 21, lines 693–695):

“Data from repeated measures were analyzed using Friedman non-parametric ANOVA followed by Dunn’s multiple comparisons test (GraphPad Prism 10). Adjusted p-values are reported for all comparisons.”

Discussion (page 14, lines 462–464):

“The lack of a clear dose-response and the absence of a robust TNF response to LPS stimulation was surprising and is most likely technical. This limitation should be addressed in future studies.”

Figure 5 (page 28-29, lines 969–984):

“Figure 5. Human PBMCs were stimulated with LPS or heat-killed *S. aureus* in the absence of presence of chicken yolk or porcine brain sphingomyelin. Human PBMCs from six healthy donors (two independent experiments, n=6) were pre-incubated for 1 h with vehicle control (RPMI) or increasing concentrations (10 μ M, 1 μ M, 100 ng/mL, 10 ng/mL) of sphingomyelin isolated from porcine brain (panels A–F) or chicken egg yolk (panels G–L), then stimulated for 24 h with heat-killed *Staphylococcus aureus* (A–C, G–I) or LPS (D–F, J–L). Cytokine levels in supernatants were measured by ELISA for IL-6 (A, D, G, J), TNF (B, E, H, K) and IL-1 β (C, F, I, L). Bars represent mean \pm SEM with individual donor values overlaid as dots. For panel K, box plots represent the median (centre line), the 25th and 75th percentiles (bounds of the box), whiskers indicate the minimum and maximum values. Statistical analysis was performed using Friedman non-parametric ANOVA followed by Dunn’s multiple comparisons test versus RPMI control. Statistical significance versus RPMI control is indicated by brackets: * $p < 0.05$, ** $p < 0.01$, *** $p < 0.001$, **** $p < 0.0001$. Exact adjusted p values (Dunn’s test): B: RPMI vs. 1 μ M/mL, $p = 0.0076$; C: RPMI vs. 10 μ M/mL, $p = 0.0139$; RPMI vs. 1 μ M/mL, $p = 0.0423$; G: RPMI vs. 10 μ M/mL, $p = 0.0423$; I: RPMI vs. 10 μ M/mL, $p = 0.0010$; J: RPMI vs. 10 μ M/mL, $p = 0.0041$; L: RPMI vs. 10 μ M/mL, $p < 0.0001$; RPMI vs. 1 μ M/mL, $p = 0.0041$.”

26th Aug 2025

Manuscript number: MSB-2025-12900RR

Title: Deciphering Cross-Cohort Metabolic Signatures of Immune Responses and Their Implications for Disease Pathogenesis

Dear Prof Li,

Thank you again for sending us your revised manuscript. We are now satisfied with the modifications made and I am pleased to inform you that your paper has been accepted for publication.

Sincerely,
Jingyi

Jingyi Hou, PhD
Senior Editor
Molecular Systems Biology
